# Complexity of Decentralized Optimization with Mixed Affine Constraints

**Demyan Yarmoshik** [1]   **Nhat Trung Nguyen** [1]   **Alexander Rogozin** [1]   **Alexander Gasnikov** [1 2 3]

## Abstract

This paper considers decentralized optimization of convex functions with mixed affine equality constraints involving both local and global variables. Constraints on global variables may vary across different nodes in the network, while local variables are subject to coupled and node-specific constraints. Such problem formulations arise in machine learning applications, including federated learning and multi-task learning, as well as in resource allocation and distributed control. We analyze this problem under smooth and non-smooth assumptions, considering both strongly convex and general convex objective functions. Our main contribution is an optimal algorithm for the smooth, strongly convex regime, whose convergence rate matches established lower complexity bounds. We further provide optimal and near-optimal methods for the remaining cases.

## 1. Introduction

We consider distributed optimization problems in which the objective function is decomposed across multiple computational nodes and the decision variables are subject to both local and global affine constraints. Specifically, we consider problems of the form

$$\min_{x_1,\ldots,x_n,\tilde{x}} \sum_{i=1}^{n} f_i(x_i, \tilde{x}) \tag{P}$$

$$\text{s.t.} \sum_{i=1}^{n} (A_i x_i - b_i) = 0, \; C_i x_i = c_i, \; \widetilde{C}_i \tilde{x} = \tilde{c}_i \quad \forall i.$$

There are two groups of variables: $x_i \in \mathbb{R}^{d_i}$ are individual for each node while $\tilde{x} \in \mathbb{R}^{\tilde{d}}$ is common for all the nodes. Each node locally owns $x_i, A_i, b_i, C_i, c_i, \widetilde{C}_i, \tilde{c}_i$ and $f_i$, where $A_i \in \mathbb{R}^{m \times d_i}$, $b_i \in \mathbb{R}^m$, $C_i \in \mathbb{R}^{p_i \times d_i}$, $c_i \in \mathbb{R}^{p_i}$,

---

[1]MIRAI, Moscow, Russia [2]Innopolis University, Innopolis, Russia [3]HDI Lab, HSE University, Moscow, Russia. Correspondence to: Demyan Yarmoshik <yarmoshik.d@miriai.org>.

*Proceedings of the 43^{rd} International Conference on Machine Learning*, Seoul, South Korea. PMLR 306, 2026. Copyright 2026 by the author(s).

$\widetilde{C}_i \in \mathbb{R}^{\tilde{p}_i \times \tilde{d}}$, $\tilde{c}_i \in \mathbb{R}^{\tilde{p}_i}$ and $f_i$ is a convex function. Each node can perform matrix-vector multiplications with its matrices and compute function values and gradients of $f_i$. Throughout the paper, we assume that the feasible set of (P) is nonempty.

The nodes (agents) are organized into a computational network $\mathcal{G} = (\mathcal{V}, \mathcal{E})$, where $\mathcal{V} = \{1, \ldots, n\}$ is the set of vertices and $\mathcal{E}$ is the set of edges. Each agent can exchange information with those nodes to which it is connected by an edge. The constraints in (P) can be divided into three types:

**Coupled Constraints.** These constraints link the variables across multiple nodes and require coordination among nodes to satisfy. The main coupled constraints are

$$\sum_{i=1}^{n} (A_i x_i - b_i) = 0, \tag{1}$$

which jointly restrict the variables $x_i$.

**Local Constraints.** These constraints depend only on the local variables and data at each node $i$. Specifically,

$$C_i x_i = c_i, \quad 1, \ldots, n, \tag{2}$$

which can be enforced independently by node $i$ without coordination with others. They typically represent local restrictions or operational limits specific to each node.

**Shared Variable Constraints.** This type of constraint is imposed on the shared variable $\tilde{x}$. These constraints take the form

$$\widetilde{C}_i \tilde{x} = \tilde{c}_i, \quad 1, \ldots, n. \tag{3}$$

The variable $\tilde{x}$ is the same at all the nodes, but the constraint matrices $\widetilde{C}_i$ are individual.

If the problem includes a combination of coupled constraints, local and shared variable constraints, we call it a problem with *mixed constraints*. Our study is largely motivated by classical work on distributed yet centralized algorithms (Boyd et al., 2011), which shows how playing with affine-constraint reformulations enable decomposition and distributed optimization of various problems. Our main goal is to bring this level of flexibility to decentralized optimization, together with development of theoretical tools for comparing the complexity of different reformulations.

Problems of type (P) have applications in several machine learning problems, including horizontal and vertical federated learning, distributed multi-task learning, and control of distributed energy systems.

**Horizontal Federated Learning (HFL, or Consensus Optimization).** Let the variables locally held by the nodes be connected by consensus constraints, i.e., $x_1 = \ldots = x_n$. We come to the problem of consensus optimization.

$$\min_{x_1,\ldots,x_n} \sum_{i=1}^{n} f_i(x_i) \qquad (4)$$
$$\text{s.t.} \quad x_1 = \ldots = x_n.$$

This is a standard decentralized optimization (or horizontal federated learning) problem statement (Boyd et al., 2011; Yang et al., 2019; Kairouz et al., 2021; Nedić & Ozdaglar, 2009; Scaman et al., 2017; Kovalev et al., 2021b;a; Li & Lin, 2021). The training data is distributed between computational entities by samples, i.e., each node holds a part of the dataset. During the optimization process, the agents aim to maintain equal model weights $x_i$.

Coupled constraints are capable of expressing consensus constraints: for instance, cyclic equalities $x_1 - x_2 = 0$, $x_2 - x_3 = 0, \ldots, x_n - x_1 = 0$ can be written as $\sum_{i=1}^{n} A_i x_i = 0$, where $x_i \in \mathbb{R}^d$, $A_1 = \begin{pmatrix} I_d & 0_d & \cdots & 0_d & -I_d \end{pmatrix}^\top$ and so on. It is interesting, though, that all black-box reductions of consensus constraints to coupled constraints are provably ineffective for first-order decentralized algorithms, as any choice of matrices $A_i$ increases communication complexity by at least a factor of $\sqrt{n}$ compared with optimal specialized consensus optimization algorithms (Yarmoshik et al., 2024b, Appendix A). This fact motivates the explicit differentiation between consensus and coupled-constrained variables in (P).

**Vertical Federated Learning (VFL)** Unlike consensus optimization, where the data is distributed sample-wise between the nodes, in VFL the data is distributed feature-wise (Liu et al., 2024; Chen et al., 2020). Each node corresponds to a party possessing its local subset of weights $X_i$, and submatrix of features $F_i$.

In deep VFL each party transforms its local features into an intermediate representation, which is then consumed by a shared "top" model. A simple instance keeps the party-side mapping linear, $H_i := F_i X_i \in \mathbb{R}^{N \times m}$ with $X_i \in \mathbb{R}^{d_i \times m}$, aggregates $Z := \sum_{i=1}^{n} H_i$, and predicts $\widehat{y} = g(Z; \widetilde{X})$ with top-model parameters $\widetilde{X}$.

This yields the mixed-constraints formulation

$$\min_{Z,\widetilde{X},X_1,\ldots,X_n} \ell(g(Z,\widetilde{X})) + \sum_{i=1}^{n} r_i(X_i) + r(\widetilde{X})$$
$$\text{s.t.} \quad \sum_{i=1}^{n} F_i X_i = Z,$$

where $r_i$, $r$ are regularizers. Note that the affine constraint $\sum_{i=1}^{n} F_i X_i - IZ = 0$ is a special case of coupled constraints. Papers (Vepakomma et al., 2018; Xie et al., 2024) considered similar setup, but used nonlinear mappings for intermediate representations.

A fully decentralized implementation can avoid a dedicated "top" node by replicating the top-model parameters and splitting the sample dimension for computational parallelism. Partition each party feature matrix into vertical blocks $F_i = \mathrm{col}\,(F_{i,1}, \ldots, F_{i,n})$, which induces the corresponding split $H_i = \mathrm{col}\,(F_{i,1}X_i, \ldots, F_{i,p}X_i)$. Introduce local aggregated representations $Z_i$ and local copies of the top-model parameters $\widetilde{X}_i$. Then VFL can be written as a problem with mixed coupled and consensus constraints that fits directly into (P):

$$\min_{\substack{Z_1,\ldots,Z_n \\ \widetilde{X}_1,\ldots,\widetilde{X}_n \\ X_1,\ldots,X_n}} \sum_{i=1}^{n} \ell_i\left(g(Z_i,\widetilde{X}_i)\right) + \sum_{i=1}^{n} r_i(X_i) + \frac{1}{n}\sum_{i=1}^{n} r(\widetilde{X}_i)$$
$$\text{s.t.} \quad \sum_{i=1}^{n} F_{i,j} X_i = Z_j, \quad j = 1,\ldots,n,$$
$$\widetilde{X}_1 = \ldots = \widetilde{X}_n,$$

where $\ell_i$ aggregates the losses over the $i$-th sample block and the consensus constraint enforces a shared top model.

Other problem formulations that reduce to (P) include a mixture of global and local models (Zhang et al., 2015; Hanzely & Richtárik, 2020; Hanzely et al., 2020), distributed multi-task learning (Wang et al., 2016) and federated self-supervised learning (Makhija et al., 2022). We discuss these problems in Appendix B.

**Related Work.** Most popular special cases of problem (P) are consensus constraints and coupled constraints. Both of these scenarios have been largely studied in the literature.

**Consensus constraints** have form $x_1 = \ldots = x_n$ and are a special case of (P) when only the shared variable $\tilde{x}$ is present. Originating from the works (Nedić & Ozdaglar, 2009; Boyd et al., 2011), the work on decentralized consensus optimization came to building lower complexity bounds and optimal first-order algorithms (Scaman et al., 2017; Kovalev et al., 2020) and generalization to time-varying graphs (Nedic et al., 2017; Kovalev et al., 2021b;a; Li & Lin, 2021). Generalizations on randomly varying networks

were also done in (Koloskova et al., 2020; 2021). Algorithms for nonsmooth problems were proposed in (Scaman et al., 2018; Dvinskikh & Gasnikov, 2021a; Gorbunov et al., 2019; Kovalev et al., 2024).

Consensus optimization arises in large scale model training, i.e. in federated learning (Kairouz et al., 2019; McMahan et al., 2017; Lian et al., 2017).

**Coupled constraints** problem statements initially arised in control of distributed power systems. Different variants of method with corresponding engineering problem statements were studied in (Necoara et al., 2011; Necoara & Nedelcu, 2014; 2015). A special case of coupled constraints is resource allocation problem (Doan & Olshevsky, 2017; Li et al., 2018; Nedić et al., 2018). Aside from coupled equality constraints, nonlinear inequality constraints (Liang et al., 2019; Gong & Zhang, 2023; Wu et al., 2022), local constraints (Nedic et al., 2010; Zhu & Martinez, 2011), restricted domains of local functions (Wang & Hu, 2022; Liang et al., 2019; Nedić et al., 2018; Gong & Zhang, 2023; Zhang et al., 2021; Wu et al., 2022) were studied. Generalizations on time-varying networks were presented in (Zhang et al., 2021; Nedić et al., 2018).

The two main approaches to coupled constraints optimization include proximal ADMM-type algorithms and gradient methods. Proximal methods were studied in (Boyd et al., 2011; Chang, 2016; Falsone et al., 2020; Wu et al., 2022), and (Gong & Zhang, 2023) studied an algorithm with inexact prox. Computing the proximal mapping is computationally tractable when the objective structure is prox-friendly. For this reason, proximal methods are mostly applied in power systems control, where the loss functions are simple enough. The situation is different in machine learning, where first-order methods are needed (Doan & Olshevsky, 2017; Nedić et al., 2018; Yarmoshik et al., 2024b). Lower complexity bounds and first-order optimal methods are presented in (Yarmoshik et al., 2024b).

The value of coupled constraints for machine learning research is mostly its application to vertical federated learning (Chen et al., 2020; Liu et al., 2024; Stanko et al., 2026).

**Mixed constraints**. To the best of our knowledge, mixed constraints are little studied in the literature. The seminal work (Boyd et al., 2011) proposes general form consensus, when each of the local variables $x_i$ is a subvector of the global variable $x$. However, if local functions do not depend on $x$, such constraint may be written in a coupled (not mixed) constraint form. Papers (Du & Meng, 2023; 2025) proposes aggregative optimization, where each local function depends on the global decision vector $x$ along with global vector part $x_i$, and the optimization is performed w.r.t. coupled inequality constraints. They use a first-order method and achieve a linear convergence rate. Their prob-

lem can be written in form (P), but with local constraints that tie local individual variables to shared variables. In our paper, we study a slightly different setting.

**Our Contributions.**

- A new class of decentralized problems with mixed affine constraints that generalizes most popular consensus-constrained and coupled-constrained setups, allowing for their efficient combination.
- Tight complexity analysis of first-order algorithms (with new corresponding lower bounds) in the smooth and strongly convex case, and supposedly optimal and near-optimal upper bounds for nonsmooth and non-strongly convex setups.
- A quantification of how joint structure of distributed affine constraints and communication network affects optimization performance.

**Notation.** For vectors $x_1, \ldots, x_n$ we denote a column-stacked vector $\mathbf{x} = \mathrm{col}\,(x_1, \ldots, x_n)$. We denote by $\|\cdot\|$ the Euclidean norm, and by $\langle \cdot, \cdot \rangle$ the standard inner product. Sign $\otimes$ denotes the Kronecker product. We use bold letters for matrices stacked from different agents. These are block matrices and matrices obtained by Kronecker product, e.g. $\mathbf{A} = \mathrm{diag}(A_1, \ldots, A_n)$, $\mathbf{A}' = (A_1 \ldots A_n)$. Maximal and minimal positive singular values of matrix $A$ are denoted $\sigma_{\max}(A)$ and $\sigma_{\min+}(A)$, respectively. Maximal and minimal positive eigenvalues of a symmetric matrix $A$ are denoted $\lambda_{\max}(A)$ and $\lambda_{\min+}(A)$, respectively. For any matrix $A$, we introduce $\kappa_A = \frac{\sigma_{\max}^2(A)}{\sigma_{\min+}^2(A)}$. When referring to a group of matrices $A$, we interpret this parameter as defined for the block-diagonal matrix, i.e. $\kappa_A = \frac{\max_i \sigma_{\max}^2(A_i)}{\min_i \sigma_{\min+}^2(A_i)}$. For a given linear subspace $\mathcal{L}$ we also introduce a corresponding projection operator $\mathbf{P}_{\mathcal{L}}$. Additional notation and summary tables are provided in Appendix A.

**Paper Organization.** We begin with overview of optimization methods for convex optimization with affine constraints in Section 2. The algorithms described in Section 2 are then applied to specific reformulations of corresponding special cases of problem (P) in the following sections. Section 3 gives an overview of existing results in the field, and Section 4 contains complexity results (optimal algorithms with corresponding lower bounds) for smooth strongly convex setup which are summarized in Table 1. Analogously, the results for nonsmooth and non-strongly convex settings are described in Section 5. The concluding remarks are given in Section 6.

| Prob. | Oracle | Compl. |
|---|---|---|
| Consensus constr. (4) | Grad. Comm. Paper | $\dfrac{\sqrt{\kappa_f}}{\sqrt{\kappa_f}\sqrt{\kappa_W}}$ (Scaman et al., 2017) |
| Coupled constr. (1) | Grad. Mat. $A$ Comm. Paper | $\dfrac{\sqrt{\kappa_f}}{\sqrt{\kappa_f}\sqrt{\widehat{\kappa}_A}}$ $\sqrt{\kappa_f}\sqrt{\widehat{\kappa}_A}\sqrt{\kappa_W}$ (Yarmoshik et al., 2024b) |
| Shared var. constr. (S) | Grad. Mat. $\widetilde{C}$ Comm. Paper | $\dfrac{\sqrt{\kappa_f}}{\sqrt{\kappa_f}\sqrt{\widehat{\kappa}_{\widetilde{C}^\top}}}$ $\sqrt{\kappa_f}\sqrt{\widehat{\kappa}_{\widetilde{C}^\top}}\sqrt{\kappa_W}$ This paper, Th. 4.1 |
| Local var. constr. (C&L) | Grad. Mat. $A$ Mat. $C$ Comm. Paper | $\dfrac{\sqrt{\kappa_f}}{\sqrt{\kappa_f}\sqrt{\widetilde{\kappa}_{AC}}}$ $\sqrt{\kappa_f}\sqrt{\widetilde{\kappa}_{AC}}\sqrt{\kappa_C}$ $\sqrt{\kappa_f}\sqrt{\widetilde{\kappa}_{AC}}\sqrt{\kappa_W}$ This paper, Th. 4.5 |
| Mixed constr. (P) | Grad. Mat. $A$ Mat. $C$ Mat. $\widetilde{C}$ Comm. Paper | $\dfrac{\sqrt{\kappa_f}}{\sqrt{\kappa_f}\sqrt{\widetilde{\kappa}_{AC}}}$ $\sqrt{\kappa_f}\sqrt{\widetilde{\kappa}_{AC}}\sqrt{\kappa_C}$ $\sqrt{\kappa_f}\sqrt{\widehat{\kappa}_{\widetilde{C}^\top}}$ $\sqrt{\kappa_f}\left(\sqrt{\widetilde{\kappa}_{AC}}+\sqrt{\widehat{\kappa}_{\widetilde{C}^\top}}\right)\sqrt{\kappa_W}$ This paper, Th. 4.6 |

*Table 1.* Convergence rates for decentralized smooth strongly convex optimization with affine constraints. Mixed condition numbers are denoted $\widehat{\kappa}_A$, $\widehat{\kappa}_{C^\top}$ (see Definition 3.4). Condition number $\widetilde{\kappa}_{AC}$ is defined in Definition 4.2. The term $\log\left(\frac{1}{\varepsilon}\right)$ is omitted.

## 2. Optimization with Affine Constraints

All problems considered in this work can be formulated as convex optimization problems with affine equality constraints:

$$\min_{u \in \mathcal{U}} G(u) \quad \text{s.t.} \quad Bu = b, \tag{5}$$

where $G : \mathcal{U} \to \mathbb{R}$ is a convex function defined on a set $\mathcal{U} \subseteq \mathbb{R}^d$, the matrix $B \in \mathbb{R}^{s \times d}$ is nonzero, and $b$ belongs to the column space of $B$. Depending on the structural properties of the objective function $G$, we employ different optimization techniques to solve problem (5).

In the algorithms below the affine constraint $Bu = b$ enters only through the operator

$$\mathcal{R}_{B,b}(u) := B^\top(Bu - b). \tag{6}$$

### 2.1. Assumptions on the Objective Function

We introduce a set of assumptions that enable the derivation of convergence guarantees and complexity bounds for solving problem (5) over several classes of convex functions.

**Assumption 2.1** (Strong convexity). The function $G(u)$ is $\mu$-strongly convex on $\mathcal{U}$ for $\mu \geq 0$, i.e. for any $u, u' \in \mathcal{U}$,

$$G(u') \geq G(u) + \langle \nabla G(u), u' - u \rangle + \frac{\mu}{2}\|u' - u\|^2.$$

When $\mu = 0$, the function $G$ is said to be (non-strongly) convex.

**Assumption 2.2** (Smoothness). The function $G(u)$ is $L$-smooth on $\mathcal{U}$, i.e. it is differentiable on $\mathcal{U}$ and for any $u, u' \in \mathcal{U}$ we have

$$G(u') - G(u) - \langle \nabla G(u), u' - u \rangle \leq \frac{L}{2}\|u' - u\|^2.$$

**Assumption 2.3** (Bounded subgradients). The function $G$ has bounded subgradients on $\mathcal{U}$, i.e. there exists a constant $M > 0$ such that for any $u \in \mathcal{U}$ and any $G'(u) \in \partial G(u)$,

$$\|G'(u)\| \leq M.$$

### 2.2. Smooth Objectives

Problems of type (5) with smooth objectives (i.e. satisfying Assumption 2.2) can be solved using the Accelerated Proximal Alternating Predictor-Corrector algorithm (APAPC), proposed in (Salim et al., 2022) for strongly convex problems. This is an optimal method for convex optimization with affine constraints when $\mathcal{U} = \mathbb{R}^d$. To apply the method for non-strongly convex objectives, we introduce a reduction technique via regularization which analysis is non-standard due to the presence of affine constraints. The theoretical complexity results for APAPC are summarized in Theorem 2.4. Although during paper review we found an alternative algorithmic approach, the composite Penalty Similar Triangles Method (PSTM) (Dvinskikh & Gasnikov, 2021b, Section 3), with better gradient complexity, we still present the discussion of the regularization technique in Section D.

**Theorem 2.4.** *Let $G(u)$ satisfy Assumptions 2.1 and 2.2 with $0 < \mu < L$. Let $\left\|u^0 - u^*\right\|^2 \leq R^2$ and introduce accuracy $\varepsilon > 0$.*

*1. **Strongly convex case** $\mu > 0$ (Salim et al., 2022, Proposition 1). There exists a set of parameters for Algorithm 1 such that after $N = O\left(\kappa_B\sqrt{L/\mu}\log(1/\varepsilon)\right)$ iterations the algorithm yields $u^N$ satisfying $\left\|u^N - u^*\right\|_2^2 \leq \varepsilon$.*

*2. **Convex case** $\mu = 0$ (**new**, Appendix D). Applying Algorithm 1 to regularized function $G^\nu(u) = G(u) + \nu/2\|u^0 - u\|^2$ with $\nu = \varepsilon/R^2$, after $N = O\left(\kappa_B\sqrt{LR^2/\varepsilon}\log(1/\varepsilon)\right)$ iterations we obtain $u^N$, for which $G(u^N) - G(u^*) \leq \varepsilon$ and $\|Bu^N - b\|^2 \leq O(\sigma_{\max}^2(B)\varepsilon^2)$.*

### 2.3. Non-Smooth Objectives

In this section, we present an approach for solving Problem (5) when the objective function $G$ is non-smooth. Following (Dvinskikh & Gasnikov, 2021b; Gorbunov et al., 2019), we handle the affine equality constraints by introduc-

**Algorithm 1** APAPC

1: **Input:** $u^0 \in \mathbb{R}^d$
2: **Parameters:** $\eta, \theta, \alpha > 0, \tau \in (0,1), N \in \mathbb{N}$
3: Set $u_f^0 = u^0, z^0 = 0 \in \mathbb{R}^d$
4: **for** $k = 0, 1, 2, \ldots, N-1$ **do**
5: $\quad u_g^k := \tau u^k + (1-\tau)u_f^k$
6: $\quad u^{k+\frac{1}{2}} := (1+\eta\alpha)^{-1}(u^k - \eta(\nabla G(u_g^k) - \alpha u_g^k + z^k))$
7: $\quad z^{k+1} := z^k + \theta \mathcal{R}_{B,b}(u^{k+\frac{1}{2}})$
8: $\quad u^{k+1} := (1+\eta\alpha)^{-1}(u^k - \eta(\nabla G(u_g^k) - \alpha u_g^k + z^{k+1}))$
9: $\quad u_f^{k+1} := u_g^k + \frac{2\tau}{2-\tau}(u^{k+1} - u^k)$
10: **end for**
11: **Output:** $u^N$

ing the penalized objective function:

$$\min_{u \in \mathcal{U}} H_r(u) := G(u) + \frac{r}{2}\|Bu - b\|^2, \quad (7)$$

where $r > 0$ is the penalty coefficient.

With a appropriate choice of $r$, solving the original problem reduces to minimizing $H_r$ (see Appendix E for details). Since $H_r$ consists of a non-smooth term $G(u)$ and a smooth quadratic penalty, we can apply the sliding technique to obtain separate complexity bounds for each component. Specifically, we employ the GRADIENT SLIDING algorithm from (Lan, 2019). To describe the algorithm, we define the following quadratic model of $H_r$:

$$\Phi_{H_r}(w, u_1, u_2, u_3, \beta, \eta) = \langle G'(u_1) + r\mathcal{R}_{B,b}(u_2), w \rangle$$
$$+ \frac{\beta}{2}\|u_3 - w\|^2 + \frac{\beta\eta}{2}\|u_1 - w\|^2, \quad (8)$$

where $\beta, \eta$ are parameters and $w, u_1, u_2, u_3 \in U$. We then derive the Algorithm 2 for solving Problem (5) in non-smooth setting. At each iteration, the algorithm finds a solution of the subproblem of type (8).

**Algorithm 2** GRADIENT SLIDING

1: **Input:** $u^0 \in U$
2: **Parameters:** $\{\gamma_k\}_{k=1}^\infty, \{\eta_k\}_{t=1}^\infty, \{\theta_t\}_{t=1}^\infty \subseteq \mathbb{R}_{++}$, $N \in \mathbb{N}, \{T_k\}_{k=1}^N \subseteq \mathbb{N}^N$
3: $\overline{u}^0 := u^0$
4: **for** $k = 1, 2, \ldots, N$ **do**
5: $\quad \underline{u}^k := \gamma_k u^{k-1} + (1-\gamma_k)\overline{u}^{k-1}$
6: $\quad u_0^k := \tilde{u}_0^k, \tilde{u}_0^k := u^{k-1}$
7: $\quad$ **for** $t = 1, 2, \ldots, T_k$ **do**
8: $\quad\quad u_t^k = \arg\min_{w \in \mathcal{U}} \Phi_{H_r}(w, u_{t-1}^k, \underline{u}^k, u^{k-1}, \beta_k, \eta_t)$
9: $\quad\quad \tilde{u}_t^k = \theta_t u_t^k + (1-\theta_t)\tilde{u}_{t-1}^k$
10: $\quad$ **end for**
11: $\quad u^k := u_{T_k}^k, \tilde{u}^k := \tilde{u}_{T_k}^k$
12: $\quad \overline{u}^k := \gamma_k \tilde{u}^k + (1-\gamma_k)\tilde{u}^{k-1}$
13: **end for**
14: **Output:** $\overline{u}^N$

Theorem 2.5 provides upper bounds on the number of gradient computations and matrix multiplications required for Algorithm 2 to reach $(\varepsilon, \delta)$ accuracy, which means $G(u) - G(u^*) \leq \varepsilon$ and $\|Bu - b\| \leq \delta$. For details refer to Appendix E.

**Theorem 2.5** (Lan et al. (2020))**.** *Let $G$ satisfy Assumption 2.1 with $\mu \geq 0$, Assumption 2.3 with $M > 0$, and suppose $\|u^0 - u^*\|^2 \leq R^2$. Given $\varepsilon > 0$, consider solving the penalized problem* (7) *using Algorithm 2.*
*1.* ***Convex case*** *($\mu = 0$). The algorithm requires $N_B = O\left(\frac{MR}{\varepsilon}\sqrt{\kappa_B}\right)$ multiplications by $B, B^\top$ and $N_{G'} = O\left(\frac{M^2 R^2}{\varepsilon^2} + N_B\right)$ evaluations of subgradient of $G$ to generate an output $\overline{u}^N$, for which $G(\overline{u}^N) - G(u^*) \leq \varepsilon$, and $\|B\overline{u}^N - b\| \leq O(\varepsilon)$.*
*2.* ***Strongly convex case*** *($\mu > 0$). With restarting, the algorithm returns $\overline{u}^N$ satisfying the same accuracy guarantees using $N_B = O\left(\frac{M}{\sqrt{\mu\varepsilon}}\sqrt{\kappa_B}\right)$ multiplications by $B, B^\top$ and $N_{G'} = O\left(\frac{M^2}{\mu\varepsilon} + N_B\right)$ evaluations of subgradient of $G$.*

### 2.4. Chebyshev Acceleration

Chebyshev acceleration (Salim et al., 2022; Scaman et al., 2017; Auzinger & Melenk, 2011), initially applied as a theoretical tool to reduce communication complexity in smooth consensus optimization, effectively improves condition number of affine constraint matrix. The idea of Chebyshev acceleration is to equivalently reformulate an affine constraint $Bu = b$ as $Kx = b'$, where $K = P_B(B^\top B)$, $b' = \frac{P_B(B^\top B)}{B^\top B}B^\top b$, and $P_B$ is a polynomial such that $P_B(\sigma_i^2(B)) = $ if and only if $\sigma_i(B) = 0$. With appropriately scaled and shifted Chebyshev polynomials, the reformulation would have condition number $\kappa_K = O(1)$ and matrix-vector multiplications $Kx$ could be implemented via $O(\sqrt{\kappa_B})$ matrix-vector multiplications by $B^\top u$ and $Bu$, so there is no need to explicitly compute and store matrix polynomial $K$.

In the sequel, when we say that Chebyshev acceleration is applied to the constraint matrix $B$, we mean the following operational replacement: every occurrence of the operation $\mathcal{R}_{B,b}(u)$ in the base algorithm is replaced by a Chebyshev-accelerated residual oracle $\mathsf{Ch}_{B,b}(u)$.

The oracle $\mathsf{Ch}_{B,b}$ denotes the output of a Chebyshev recurrence corresponding to the particular structure of $B$. Thus, the notation is abstract: for different affine matrices $B$ used in the paper, the oracle is implemented by different routines described in Appendix C. We use the shorthand notation $B \rightarrow P_B(B^\top B)$ for this replacement.

# 3. Existing Results in Decentralized Optimization with Affine Constraints

This section discusses about known results in decentralized optimization and their convergence properties. We review related formulations of the decentralized optimization problem from prior work and their convergence properties. These formulations are special cases of the problem (P).

## 3.1. Notations and Assumptions

Decentralized communication is typically represented as a matrix-vector multiplication. In particular, we assume the existence of a *gossip matrix* $W \in \mathbb{R}^{n \times n}$, associated with the communication network $\mathcal{G} = (\mathcal{V}, \mathcal{E})$, with the following properties.

**Assumption 3.1.** The *gossip matrix* $W$ satisfies:
1. $W$ is symmetric and positive semidefinite.
2. $W_{ij} \neq 0$ if and only if $i = j$ or $(i,j) \in \mathcal{E}$.
3. $Wx = 0$ if and only if $x_1 = \ldots = x_n$.

An example of such matrix is the Laplacian matrix $L = D - A$, where $A$ is the adjacency matrix and $D$ is the degree matrix of the network $\mathcal{G}$. Throughout this paper, we denote $\mathbf{W} = W \otimes I$, where the dimension of identity matrix $I$ is determined by the context.

We consider the following standard assumptions on the objective functions in decentralized optimization.

**Assumption 3.2.** Each function $f_i$ is $L_f$-smooth and $\mu_f$-strongly convex for some $L_f \geq \mu_f \geq 0$.

When $\mu_f > 0$, we denote the condition number of the local functions by $\kappa_f = L_f / \mu_f$.

**Assumption 3.3.** Each function $f_i$ has bounded subgradients with constant $M_f > 0$ and is $\mu_f$-strongly convex for some $\mu_f \geq 0$.

## 3.2. Consensus Optimization

This case corresponds to scenarios without affine equality constraints, involving only common variables. When Assumptions 3.1 and 3.2 hold with $\mu_f > 0$, the optimal convergence rates are $O(\sqrt{\kappa_f} \log(1/\varepsilon))$ for gradient calls and $O(\sqrt{\kappa_f}\sqrt{\kappa_W} \log(1/\varepsilon))$ for communication rounds, as established in (Kovalev et al., 2020).

In the nonsmooth and non-strongly convex setting, that is, when Assumptions 3.1 and 3.3 hold with $\mu_f = 0$, the number of subgradient evaluations is upper bounded by $O(M_f^2 R^2 / \varepsilon)$, while the number of communication rounds is bounded by $O(\sqrt{\kappa_W} M_f R / \varepsilon)$, where $R$ denotes the radius of the feasible region. These bounds are achieved and shown to be optimal in (Scaman et al., 2018).

## 3.3. Identical Local Constraints

Consider the case with only the common variable $\tilde{x}$ and identical local constraints, i.e., $\widetilde{C}_i = \widetilde{C}$ and $\tilde{c}_i = \tilde{c}$ for all $i = 1, \ldots, n$. In (Rogozin et al., 2022), author obtained upper bounds for this formulation when Assumptions 3.1 and 3.2 hold: $O(\sqrt{\kappa_f} \log(1/\varepsilon))$ gradient calls, $N_{\widetilde{C}} = O(\sqrt{\kappa_f}\sqrt{\kappa_{\widetilde{C}}} \log(1/\varepsilon))$ multiplications by $\widetilde{C}$ and $\widetilde{C}^\top$, and $N_W = O\left(\sqrt{\kappa_f}\sqrt{\kappa_W} \log(1/\varepsilon)\right)$ communications. As we will show in Section 4.1, the complexity bounds change substantially when local constraints are nonidentical.

## 3.4. Coupled Constraints

We now consider the case when each node locally holds a part of affine constraints $A_i$.

$$\min_{x_1 \in \mathbb{R}^{d_1}, \ldots, x_n \in \mathbb{R}^{d_n}} \sum_{i=1}^n f_i(x_i) \quad \text{s.t.} \quad \sum_{i=1}^n (A_i x_i - b_i) = 0. \tag{9}$$

This problem was studied in (Yarmoshik et al., 2024b) for smooth and strongly convex objective functions. Authors introduced a new kind of condition number that captured the convergence rates for coupled constraints.

**Definition 3.4.** For a set of matrices $(B_1, \ldots, B_n)$, introduce an interaction matrix

$$S_B = \frac{1}{n} \sum_{i=1}^n B_i B_i^\top, \tag{10}$$

and a mixed condition number

$$\widehat{\kappa}_B = \frac{\max\limits_{i=1,\ldots,n} \lambda_{\max}(B_i B_i^\top)}{\lambda_{\min^+}(S_B)} = \frac{\max\limits_{i=1,\ldots,n} \lambda_{\max}(B_i B_i^\top)}{\lambda_{\min^+}\left(\frac{1}{n}\sum_{i=1}^n B_i B_i^\top\right)}.$$

*Remark* 3.5. Note that the transposition order in definition of $\widehat{\kappa}_B$ is important. Let

$$\widehat{\kappa}_{B^\top} = \frac{\max\limits_{i=1,\ldots,n} \lambda_{\max}(B_i^\top B_i)}{\lambda_{\min^+}(S_{B^\top})} = \frac{\max\limits_{i=1,\ldots,n} \lambda_{\max}(B_i^\top B_i)}{\lambda_{\min^+}\left(\frac{1}{n}\sum_{i=1}^n B_i^\top B_i\right)}.$$

In general $\widehat{\kappa}_{B^\top} \neq \widehat{\kappa}_B$. For example, let $B_i = e_i$, where $e_i$ is the $i$-th coordinate vector. Then $\widehat{\kappa}_B = n$ and $\widehat{\kappa}_{B^\top} = 1$.

*Remark* 3.6. Note that in the case of equal matrices $B_1 = \ldots = B_n = B$ mixed condition number $\widehat{\kappa}_B$ naturally reduces to usual condition number $\kappa_B$ so that $\widehat{\kappa}_B = \widehat{\kappa}_{B^\top} = \kappa_B$.

When Assumptions 3.1 and 3.2 hold with $\mu_f > 0$, the optimal convergence rates achieved in (Yarmoshik et al., 2024b) are $O(\sqrt{\kappa_f} \log(1/\varepsilon))$ for gradient calls, $O(\sqrt{\kappa_f}\sqrt{\widehat{\kappa}_A} \log(1/\varepsilon))$ matrix-vector multiplications $\mathbf{A}$, $\mathbf{A}^\top$ and $O(\sqrt{\kappa_f}\sqrt{\widehat{\kappa}_A}\sqrt{\kappa_W} \log(1/\varepsilon))$ for communication rounds.

# 4. Optimal Algorithms for Smooth Strongly Convex Problems

Before deriving the algorithm and complexity bound for the problem (P), we consider two special cases. The first is when only local constraints are imposed on the common variable, and the second is when both coupled and local constraints are present.

## 4.1. Shared Variable Constraints

We consider the case when each node is subject to its own local affine constraint on the global variable. In particular, we study the generation of formulation in (Rogozin et al., 2022) from identical to non-identical constraints. The optimization problem can be written as

$$\min_{\tilde{x}\in\mathbb{R}^d} \sum_{i=1}^n f_i(\tilde{x}) \quad \text{s.t.} \quad \widetilde{C}_i\tilde{x} = \tilde{c}_i \quad \forall i \in \{1,\ldots,n\}. \quad \text{(S)}$$

Introducing local copies of the variable allows us to reformulate the problem in a block-matrix format

$$\min_{\tilde{\mathbf{x}}\in(\mathbb{R}^{\tilde{d}})^n} F(\tilde{\mathbf{x}}) := \sum_{i=1}^n f_i(\tilde{x}_i) \text{ s.t. } \mathbf{W}\tilde{\mathbf{x}}=0,\ \widetilde{\mathbf{C}}\tilde{\mathbf{x}}=\tilde{\mathbf{c}}, \quad (11)$$

where $\tilde{\mathbf{x}} = \text{col}\{\tilde{x}_1,\ldots,\tilde{x}_n\}$, $\tilde{\mathbf{c}} = \text{col}\{\tilde{c}_1,\ldots,\tilde{c}_n\}$, $\mathbf{W} = W \otimes I_d$ encodes consensus, and matrix $\widetilde{\mathbf{C}} = \text{diag}(\widetilde{C}_1,\ldots,\widetilde{C}_n) \in \mathbb{R}^{\tilde{p}\times\tilde{d}n}$ collects the local constraints, where $\tilde{p} = \sum_{i=1}^n \tilde{p}_i$.

The constraint matrix has a block structure: $\widetilde{\mathbf{B}}^\top = (\widetilde{\mathbf{C}}^\top\ \ \gamma\mathbf{W})$. Applying Lemma 2 from (Yarmoshik et al., 2024b), we select an appropriate scaling parameter $\gamma$ and derive the upper bounds for decentralized optimization with affine equality constraints on shared variable in Theorem 4.1.

**Theorem 4.1** (**new,** Appendix G). *Applying Algorithm 1 to a reformulation of problem (11) with $\mathbf{W} \to P_W(\mathbf{W})$ and then $\widetilde{\mathbf{B}} \to P_C(\widetilde{\mathbf{B}}^\top\widetilde{\mathbf{B}})$, we obtain a method with complexity specified in Table 1. This bound is optimal in a naturally defined class of decentralized first-order algorithms for problems with local constraints.*

The Chebyshev acceleration used for this shared-variable constraint is implemented by Algorithm 4.

A notable distinction from the coupled case is that regularization is not needed in order to guarantee strong convexity, and there is more freedom in the design of preconditioners. In particular, $\widetilde{C}$ can be preconditioned independently of $\mathbf{W}$, whereas for coupled constraints $\mathbf{A}$ and $\mathbf{W}$ are inherently entangled. Indeed, $(\widetilde{\mathbf{C}}^\top\ \mathbf{W})^\top x = 0$ if and only if $(P(\widetilde{\mathbf{C}}^\top\widetilde{\mathbf{C}})^\top\ \mathbf{W})^\top x = 0$ for any polynomial $P$ without a constant term, provided that $P(\lambda) \neq 0$ for all nonzero

eigenvalues $\lambda$ of $\widetilde{\mathbf{C}}^\top\widetilde{\mathbf{C}}$. This property enables richer preconditioning strategies. Nevertheless, despite this additional flexibility, the upper bounds obtained above turn out to be tight. Optimality can be proved in a manner analogous to the lower bound arguments for coupled constraints (see Appendix G).

## 4.2. Coupled and Local Constraints

Individual variables are related by coupled constraints and locally tied by local constraints.

$$\min_{x_1\in\mathbb{R}^{d_1},\ldots,x_n\in\mathbb{R}^{d_n}} \sum_{i=1}^n f_i(x_i) \quad \text{(C\&L)}$$
$$\text{s.t.} \sum_{i=1}^n (A_ix_i - b_i) = 0,\ C_ix_i = c_i.$$

For this formulation the decentralized-friendly matrix of the affine constraints is $\mathbf{B} = \begin{pmatrix} \mathbf{A} & \alpha\mathbf{W} \\ \beta\mathbf{C} & 0 \end{pmatrix}$ with $\mathbf{A} = \text{diag}(A_1,\ldots,A_n)$ and $\mathbf{C} = \text{diag}(C_1,\ldots,C_n)$, see Appendix F.2. The Chebyshev acceleration for this coupled-and-local constraint matrix is implemented by Algorithm 5, corresponding to the shorthand $\mathbf{C} \to P_C(\mathbf{C}^\top\mathbf{C})$, $\mathbf{W} \to P_W(\mathbf{W})$, and $\mathbf{B} \to P_B(\mathbf{B}^\top\mathbf{B})$.

Now we introduce natural complexity parameters for this setup.

**Definition 4.2.** Consider two sets of matrices $(B_1,\ldots,B_n)$ and $(D_1,\ldots,D_n)$. Let $\mathbf{B}' = (B_1,\ldots,B_n)$ and $\mathbf{D} = \text{diag}(D_1,\ldots,D_n)$. We define

$$\widetilde{\mu}_{BD} := \frac{1}{n}\sigma^2_{\min^+}(\mathbf{B}'\mathbf{P}_{\ker\mathbf{D}}),$$

$$\widetilde{\kappa}_{BD} := \begin{cases} \dfrac{\max_{i=1,\ldots,n}\lambda_{\max}(B_iB_i^\top)}{\widetilde{\mu}_{BD}}, & \text{if } \widetilde{\mu}_{BD} > 0, \\ 1, & \text{if } \widetilde{\mu}_{BD} = 0. \end{cases}$$

The definition of $\widetilde{\kappa}_{BD} = 1$ for the case $\widetilde{\mu}_{BD} = 0$ is introduced for convenience, since in that case $\widetilde{\kappa}_{BD}$ does not affect the complexity (see Lemma 4.4 and Theorem 4.5).

*Remark* 4.3. Note that if $\widetilde{\mu}_{BD} > 0$ then we have $\widetilde{\kappa}_{BD} \leq \widehat{\kappa}_B$. Indeed, for any matrix $M$ and linear subspace $\mathcal{L}$ of compatible dimensions it holds $\ker^\perp M\mathbf{P}_{\mathcal{L}} = \ker^\perp M \cap \mathcal{L}$, and thus $\sigma_{\min^+}(M\mathbf{P}_{\mathcal{L}}) = \min_{h\in\ker^\perp M, h\in\mathcal{L}}\|Mh\|/\|h\| \geq \min_{h\in\ker^\perp M}\|Mh\|/\|h\| = \sigma_{\min^+}(M)$. The equality is reached if a singular vector corresponding to $\sigma_{\min^+}(\mathbf{B}')$ belongs to $\ker\mathbf{D}$. In particular, we have $\widetilde{\kappa}_{BD} = \widehat{\kappa}_B$ when $\mathbf{D} = 0$.

Following lemma gives upper bound for $\kappa_B$ which is essential for complexity upper bounds. It also characterizes how interconnection between distributed affine constraints and communication network affects optimization performance.

**Lemma 4.4** (**new,** Appendix H)**.** *There exist constants* $\alpha > 0$ *and* $\beta > 0$ *such that the condition number* $\kappa_{\mathbf{B}}$ *satisfies*

$$\kappa_{\mathbf{B}} = \begin{cases} O\left(\widetilde{\kappa}_{AC}\kappa_W + \widetilde{\kappa}_{AC}\kappa_C\right), & \widetilde{\mu}_{AC} > 0, \\ O\left(\kappa_W + \kappa_C\right), & \widetilde{\mu}_{AC} = 0. \end{cases} \quad (13)$$

**Theorem 4.5** (**new,** Appendix I)**.** *Applying Algorithm 1 to a reformulation of problem* (C&L) *with* $\mathbf{C} \to P_C(\mathbf{C}^\top\mathbf{C})$, $\mathbf{W} \to P_W(\mathbf{W})$ *and then* $\mathbf{B} \to P_B(\mathbf{B}^\top\mathbf{B})$ *we obtain a method with complexity specified in Table 1. These complexity bounds are optimal due to corresponding lower bounds.*

When there are no local constraints ($\mathbf{C} = 0$) we have $\widetilde{\kappa}_{AC} = \widehat{\kappa}_A$ (see Remark 4.3). In this case, the complexity bounds obtained in this subsection are identical to the bounds in Theorem F.4, providing a strong generalization of the main results from (Yarmoshik et al., 2024b).

We provide numerical experiments in Appendix N to illustrate the dependence of $\kappa_{\mathbf{B}}$ on the network and local constraint conditioning and to verify the scaling predicted by Lemma 4.4.

### 4.3. Mixed Constraints

We introduce $\mathbf{K} = \mathrm{diag}(\mathbf{B}, \widetilde{\mathbf{B}})$, $\mathbf{z} = \mathrm{col}(\mathbf{x}, \mathbf{y}, \tilde{\mathbf{x}})$, $\mathbf{v} = \mathrm{col}(\mathbf{b}, \mathbf{c}, \tilde{\mathbf{c}}, \mathbf{0})$ and define the feasible set $\mathcal{Z} = \mathbb{R}^d \times (\mathbb{R}^m)^n \times (\mathbb{R}^{\tilde{d}})^n$. Then, problem (P) can be reformulated as

$$\min_{\mathbf{z} \in \mathcal{Z}} \ G(\mathbf{z}) \coloneqq \sum_{i=1}^{n} f_i(x_i, \tilde{x}_i) \quad \text{s.t.} \quad \mathbf{Kz} = \mathbf{v}. \quad (14)$$

We now gather all our previous results and illustrate how complexity bounds for decentralized optimization with shared variable constraints (Section 4.1) and with coupled and local constraints (Section 4.2) are combined in the general mixed setting formulated in (P). Theorem 4.6 gives upper bounds for problem (P).

The Chebyshev acceleration for the mixed constraint matrix is implemented blockwise by Algorithm 6, using the routines $\mathsf{Ch}_{\mathbf{B},(\mathbf{b},\mathbf{c})}$ and $\mathsf{Ch}_{\widetilde{\mathbf{B}},\tilde{\mathbf{c}}}$ for the two diagonal blocks of $\mathbf{K}$.

**Theorem 4.6** (**new,** Appendix J)**.** *Applying Algorithm 1 to problem* (14)*, we obtain a method with complexity specified in Table 1. In the case of identical local constraints when matrices* $\widetilde{C}_i$ *are equal, complexities* $N_{\widetilde{C}}$ *and* $N_W$ *change to*

$$N_{\widetilde{C}} = O\left(\sqrt{\kappa_f}\sqrt{\kappa_{\widetilde{C}}}\log\left(\frac{1}{\varepsilon}\right)\right)$$

$$N_W = O\left(\sqrt{\kappa_f}\sqrt{\widetilde{\kappa}_{AC}}\sqrt{\kappa_W}\log\left(\frac{1}{\varepsilon}\right)\right).$$

Thus, in the presence of both coupled and local constraints acting on different sets of variables, our unified framework

yields tight complexity bounds. These results interpolate between the purely coupled and purely local cases, showing that the overall difficulty is governed by the joint conditioning of $\mathbf{A}$, $\mathbf{C}$, and the network topology $W$. These bounds are optimal due to combination of lower bounds in Theorems F.2 and F.4 in case of identical local constraints and Theorems 4.1 and F.4 in the general case.

## 5. Extension to Non-Strongly Convex and Nonsmooth Optimization

### 5.1. Smooth and Convex Optimization

Using the Penalty Similar Triangles Method (PSTM) (Dvinskikh & Gasnikov, 2021b, Section 3), we derive the upper bounds on the iteration complexity for problem (P) in smooth, non-strongly convex regime, as stated in Theorem 5.1.

**Theorem 5.1** (**new,** Appendix K)**.** *Let Assumptions 3.1 and 3.2 hold with* $L_f > \mu_f = 0$. *Consider applying the Penalty Similar Triangles Method (PSTM) (Dvinskikh & Gasnikov, 2021b, Section 3) to problem* (14)*, which is a reformulation of problem* (P)*. Then the resulting method has complexity specified in Table 6.*

From this Theorem, we directly obtain the results for other types of constraints, they are summarized in Table 6. See Appendix K for explanations. Note that these upper bounds are suboptimal in the communication and matrix-multiplication complexity due to the logarithmic factor.

### 5.2. Nonsmooth and Convex Optimization

Analogically, we derived the results for Nonsmooth and Convex case (Table 7).

**Theorem 5.2** (**new,** Appendix L)**.** *Let Assumptions 3.3 hold with* $\mu_f = 0$ *and* $M_f > 0$. *Applying Gradient Sliding to problem* (14) *with Chebyshev Acceleration we obtain a method with complexity specified in Table 7.*

### 5.3. Nonsmooth and Strongly Convex Optimization

In the strongly convex and non-smooth setting, we consider the problem (P) on a bounded set $X_1 \times \cdots \times X_n \times \widetilde{X}$, otherwise Assumption 3.3 with $\mu_f > 0$ cannot be held. Denote $\mathcal{X} = X_1 \times \cdots \times X_n$, $\widetilde{\mathcal{X}} = \left(\widetilde{X}\right)^n$. Lemma 5.3 ensures that the penalized reformulation of $G$ is strongly convex.

**Lemma 5.3** (**new,** Appendix M.1)**.** *Suppose that Assumption 2.1 holds with* $\mu_f > 0$. *Then there exist a set of constants, such that a penalized objective constructed for problem* (14)*, is* $\frac{\mu_f}{2}$-*strongly convex on* $\mathcal{X} \times \mathcal{L}_m^\perp \times \widetilde{\mathcal{X}}$.

By integrating the results from Section 2.3, Section 3 and

Lemma 5.3, we establish upper bounds on the iteration complexity of decentralized optimization with mixed affine constraints. The summarization of results is presented in Table 8.

**Theorem 5.4** (**new**, Appendix M.2). *Let Assumptions 3.3 hold with $\mu_f > 0$ and $M_f > 0$. Applying Restarted Gradient Sliding with Chebyshev Acceleration we obtain a method with complexity specified in Table 8.*

## 6. Conclusion

We unify a series of results in decentralized optimization under a problem statement of mixed affine constraints. The generality of our formulation reduces to horizontal and vertical federated learning, control of distributed systems and other topics in distributed machine learning. Our aim is to analyze the problem systematically. To do this, we prove lower complexity bounds and provide corresponding optimal methods for different variants of our problem statement.

The logical continuation of the paper is the discussion of how different problem classes reduce to each other (see i.e. results in Table 1). Can the complexity of one optimization problem be reduced if the problem is rewritten in a different form?

A different future direction is to consider constraints $\sum_{i=1}^{n}(A_i x_i + \tilde{A}_i \tilde{x}_i - b_i) = 0$. We hypothesize that such constraints can cover new machine learning formulations, i.e. mixture of global and local models in federated learning and distributed self-supervised learning.

## Acknowledgements

The study was supported by the Ministry of Economic Development of the Russian Federation (agreement No. 139-15-2025-013, dated June 20, 2025, IGK 000000C313925P4B0002).

## Impact Statement

This paper presents work whose goal is to advance the fields of Optimization Theory and Machine Learning. As the work is primarily complexity analysis, there are no potential societal consequences of our work that we feel must be specifically highlighted here.

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

## A. Additional notation

By $\mathcal{L}_m$ we denote the so-called consensus space, which is given as $\mathcal{L}_m = \{(y_1, \ldots, y_n) \in (\mathbb{R}^m)^n : y_1, \ldots, y_n \in \mathbb{R}^m$ and $y_1 = \cdots = y_n\}$, and $\mathcal{L}_m^{\perp}$ denotes the orthogonal complement to $\mathcal{L}_m$, which is given as

$$\mathcal{L}_m^{\perp} = \{(y_1, \ldots, y_n) \in (\mathbb{R}^m)^n : y_1, \ldots, y_n \in \mathbb{R}^m \text{ and } y_1 + \cdots + y_n = 0\}. \tag{15}$$

If not otherwise specified, for any matrix $A$ we denote by $L_A$ and $\mu_A$ some upper and lower bound on its maximal and minimal positive squared singular values respectively:

$$\lambda_{\max}(A^{\top}A) = \sigma_{\max}^2(A) \leq L_A, \qquad \mu_A \leq \sigma_{\min^+}^2(A) = \lambda_{\min^+}(A^{\top}A). \tag{16}$$

When referring to a group of matrices $A = (A_1 \ldots A_n)$, we interpret these parameters as defined for the block-diagonal matrix, e.g. $L_A = \max_i \sigma_{\max}^2(A_i)$, $\mu_A = \min_i \sigma_{\min^+}^2(A_i)$.

| Symbol | Description |
|---|---|
| $n$ | Number of agents (nodes) in the network |
| $\mathcal{G} = (\mathcal{V}, \mathcal{E})$ | Communication network graph |
| $x_i \in \mathbb{R}^{d_i}$ | Local decision variable of agent $i$ |
| $\tilde{x} \in \mathbb{R}^{\tilde{d}}$ | Shared (global) variable common to all agents |
| $f_i$ | Local convex objective function of agent $i$ |
| $A_i \in \mathbb{R}^{m \times d_i}$ | Coupled constraint matrix of agent $i$ |
| $C_i \in \mathbb{R}^{p_i \times d_i}$ | Local constraint matrix of agent $i$ |
| $\widetilde{C}_i \in \mathbb{R}^{\tilde{p}_i \times \tilde{d}}$ | Shared variable constraint matrix for agent $i$ |

*Table 2.* General Notation.

| Symbol | Description |
|---|---|
| $W \in \mathbb{R}^{n \times n}$ | Gossip matrix |
| $\mathbf{W} = W \otimes I$ | Consensus matrix (identity dimension determined by context) |
| $\mathbf{A} = \operatorname{diag}(A_1, \ldots, A_n)$ | Block-diagonal coupled constraint matrix |
| $\mathbf{A}' = (A_1 \cdots A_n)$ | Horizontally concatenated coupled constraints |
| $\mathbf{C} = \operatorname{diag}(C_1, \ldots, C_n)$ | Block-diagonal local constraint matrix |
| $\widetilde{\mathbf{C}} = \operatorname{diag}(\widetilde{C}_1, \ldots, \widetilde{C}_n)$ | Block-diagonal shared variable constraint matrix |

*Table 3.* Gossip, Block and Aggregate Matrices.

| Problem | Constraint matrix |
|---------|-------------------|
| Consensus (4) | $\mathbf{W}$ |
| Coupled (C) | $\begin{pmatrix} \mathbf{A} & \beta\mathbf{W} \end{pmatrix}$ |
| Shared variable (S) | $\widetilde{\mathbf{B}}^{\top} = \begin{pmatrix} \widetilde{\mathbf{C}}^{\top} & \gamma\mathbf{W} \end{pmatrix}$ |
| Coupled & Local (C&L) | $\mathbf{B} = \begin{pmatrix} \mathbf{A} & \alpha\mathbf{W} \\ \beta\mathbf{C} & 0 \end{pmatrix}$ |
| Mixed (P) | $\mathbf{K} = \mathrm{diag}(\mathbf{B}, \widetilde{\mathbf{B}})$ |

*Table 4.* Constraint matrix structures by problem type.

| Symbol | Definition | Description |
|--------|-----------|-------------|
| $\kappa_f$ | $L_f/\mu_f$ | Condition number of objective functions |
| $\kappa_W$ | $L_W/\mu_W$ | Condition number of gossip matrix |
| $\kappa_C$ | $\dfrac{\max\limits_{i} \sigma^2_{\max}(C_i)}{\min\limits_{i} \sigma^2_{\min^+}(C_i)}$ | Condition number of local constraint matrix |
| $S_B$ | $\dfrac{1}{n}\sum\limits_{i=1}^{n} B_i B_i^{\top}$ | Interaction matrix |
| $\widehat{\kappa}_A$ | $\dfrac{\max\limits_{i} \sigma^2_{\max}(A_i)}{\lambda_{\min^+}(S_A)}$ | Condition number of coupled constraint matrix |
| $\widehat{\kappa}_{\widetilde{C}^{\top}}$ | $\dfrac{\max\limits_{i} \sigma^2_{\max}(\widetilde{C}_i)}{\lambda_{\min^+}(S_{\widetilde{C}^{\top}})}$ | Condition number of shared variable constraint matrix |
| $\widetilde{\mu}_{AC}$ | $\dfrac{1}{n}\sigma^2_{\min^+}(\mathbf{A}'\,\mathbf{P}_{\ker}\mathbf{C})$ | Mixed coupled-local spectral parameter |
| $\widetilde{\kappa}_{AC}$ | $\dfrac{\max\limits_{i} \sigma^2_{\max}(A_i)}{\widetilde{\mu}_{AC}}$  if $\widetilde{\mu}_{AC} > 0$ | Mixed coupled-local condition number |

*Table 5.* Condition numbers used in the analysis.

## B. Problem Formulations for Mixed Constraints

**Mixture of Local and Global Models.** The difference with consensus optimization is that locally held model weights may change from one node to another.

$$\min_{z,x_1,\ldots,x_n} \sum_{i=1}^{n} f_i(x_i) + \frac{\lambda}{2} \|x_i - z\|^2$$
$$\text{s.t. } x_1 + \cdots + x_n = nz$$

The variable $z$ is stored at the first node, and we come to a problem with coupled constraints.

**Distributed Multi-Task Learning (MTL).** In (Wang et al., 2016), the authors propose a distributed MTL. Every node trains its own model, but joint (non-separable) regularizers are enforced in a per-coordinate fashion.

$$\min_{x_1,\ldots,x_n} \sum_{i=1}^{n} f_i(x_i) + \lambda \sum_{j=1}^{d} r(x^{(j)}),$$

where $x^{(j)}$ is a vector in $\mathbb{R}^n$ that consists of the $j$-th coordinates of $x_1, \ldots, x_n$. The regularizer is chosen as $r(x) = \|x\|_2$ or $r(x) = \|x\|_\infty$. This problem enables rewriting in a coupled constraints form. Introduce matrices $Q_{ij}$ of size $n \times d$ that have all zero entries but one unit entry at position $(i, j)$. Multiplication $y = Q_{ij}x$ returns a vector $y \in \mathbb{R}^n$ with $i$-th entry equal to the $j$-th entry of $x$ and all other components equal to zero.

$$\min_{\substack{x_1,\ldots,x_n \\ y_1,\cdots y_d}} \sum_{i=1}^{n} f_i(x_i) + \lambda \sum_{j=1}^{d} r(y_j)$$
$$\text{s.t. } y_j = \sum_{i=1}^{n} Q_{ij}x_i.$$

Here $y_j = x^{(j)}$ are vectors that collect $j$-th components of locally held $x_i$.

**Federated Self-Supervised Learning (SSL).**

In distributed SSL a group of agents seek to learn representations of the locally held data while synchronizing with others if the corresponding data has common features or samples. The parties share a common representation alignment dataset to align their representations (Makhija et al., 2022). The corresponding problem can be formulated as optimization with consensus constraints. Let each node hold a model with weights $x_i \in \mathbb{R}^{d_i}$ (local models possibly have different architectures). For simplicity we assume that models are linear and their loss functions write as $\|C_i x_i - c_i\|^2$. The consensus is enforced on inner representations, which we assume to have form $A_i x_i$. The problem of federated SSL writes as

$$\min_{\substack{x_1,\ldots,x_n \\ z_1,\ldots,z_n}} \sum_{i=1}^{n} \|C_i x_i - c_i\|^2 + \frac{\mu}{2} \|A_i x_i - z_i\|^2$$
$$\text{s.t. } z_1 = \ldots = z_n.$$

We see that this problem comes down to consensus optimization.

# C. Chebyshev Iteration

This appendix lists the Chebyshev-accelerated residual oracles used in the main text. Each routine replaces the corresponding affine residual evaluation in the base algorithm; the outer optimization algorithm itself is unchanged.

---

**Algorithm 3** $\mathsf{Ch}_{B,b}(u)$: Generic Chebyshev residual oracle (Gutknecht & Röllin, 2002)

---

1: **Input:** $u, B, b$
2: $T := \left\lceil \sqrt{\frac{\sigma_{\max}^2(B)}{\sigma_{\min^+}^2(B)}} \right\rceil$
3: $\rho := \left(\sigma_{\max}^2(B) - \sigma_{\min^+}^2(B)\right)^2 / 16, \quad \nu := \left(\sigma_{\max}^2(B) + \sigma_{\min^+}^2(B)\right)/2$
4: $\delta^0 := -\nu/2$
5: $p^0 := -B^\top(Bu - b)/\nu$
6: $u^1 := u + p^0$
7: **for** $i = 1, \dots, T-1$ **do**
8: $\quad \beta^{i-1} := \rho/\delta^{i-1}$
9: $\quad \delta^i := -(\nu + \beta^{i-1})$
10: $\quad p^i := \left(B^\top(Bu^i - b) + \beta^{i-1}p^{i-1}\right)/\delta^i$
11: $\quad u^{i+1} := u^i + p^i$
12: **end for**
13: **Output:** $u - u^T$

---

**Algorithm 4** $\mathsf{Ch}_{\widetilde{\mathbf{B}}, \tilde{\mathbf{c}}}(\tilde{\mathbf{x}})$: Chebyshev residual oracle for Problem (S)

---

1: **Input:** $\tilde{\mathbf{x}}$, $\tilde{\mathbf{c}}$
2: $\widehat{L}_{\widetilde{\mathbf{B}}} = \sigma_{\max}^2(\widetilde{\mathbf{C}}) + \left[\sigma_{\max}^2(\widetilde{\mathbf{C}}) + \lambda_{\min^+}(S_{\widetilde{C}^\top})\right]$
3: $\widehat{\mu}_{\widetilde{\mathbf{B}}} = \frac{1}{2}\lambda_{\min^+}(S_{\widetilde{C}^\top})$
4: $\rho := \left(\widehat{L}_{\widetilde{\mathbf{B}}} - \widehat{\mu}_{\widetilde{\mathbf{B}}}\right)^2 / 16, \quad \nu := \left(\widehat{L}_{\widetilde{\mathbf{B}}} + \widehat{\mu}_{\widetilde{\mathbf{B}}}\right)/2$
5: $T := \lceil\sqrt{\widehat{L}_{\widetilde{\mathbf{B}}}/\widehat{\mu}_{\widetilde{\mathbf{B}}}}\rceil, \quad \delta^0 := -\nu/2$
6: $\mathbf{p}^0 = -\left[\widetilde{\mathbf{C}}^\top(\widetilde{\mathbf{C}}\tilde{\mathbf{x}} - \tilde{\mathbf{c}}) + \gamma^2 \mathsf{Ch}_{\mathbf{W},\mathbf{0}}(\tilde{\mathbf{x}})\right]/\nu$
7: $\tilde{\mathbf{x}}^1 := \tilde{\mathbf{x}} + \mathbf{p}^0$
8: **for** $i = 1, \dots, T-1$ **do**
9: $\quad \beta^{i-1} := \rho/\delta^{i-1}$
10: $\quad \delta^i := -(\nu + \beta^{i-1})$
11: $\quad \mathbf{p}^i := \left(\widetilde{\mathbf{C}}^\top(\widetilde{\mathbf{C}}\tilde{\mathbf{x}}^i - \tilde{\mathbf{c}}) + \gamma^2 \mathsf{Ch}_{\mathbf{W},\mathbf{0}}(\tilde{\mathbf{x}}^i) + \beta^{i-1}\mathbf{p}^{i-1}\right)/\delta^i$
12: $\quad \tilde{\mathbf{x}}^{i+1} := \tilde{\mathbf{x}}^i + \mathbf{p}^i$
13: **end for**
14: **Output:** $\tilde{\mathbf{x}} - \tilde{\mathbf{x}}^T$

---

---

**Algorithm 5** $\mathsf{Ch}_{\mathbf{B},(\mathbf{b},\mathbf{c})}\left(\left(\begin{smallmatrix}\mathbf{x}\\\mathbf{y}\end{smallmatrix}\right)\right)$: Chebyshev residual oracle for Problem (C&L)

---

1: **Input:** $\begin{pmatrix}\mathbf{x}\\\mathbf{y}\end{pmatrix}, \begin{pmatrix}\mathbf{b}\\\mathbf{c}\end{pmatrix}$

2: $\rho := \left(L_A - \widetilde{\mu}_{AC}\right)^2/16, \quad \nu := \left(L_A + \widetilde{\mu}_{AC}\right)/2$

3: $T := \lceil\sqrt{L_A/\widetilde{\mu}_{AC}}\rceil, \quad \delta^0 := -\nu/2$

4: $\widehat{\mathbf{x}} = \mathsf{Ch}_{\mathbf{C},\mathbf{c}}(\mathbf{x}), \ \widehat{\mathbf{y}} = \mathsf{Ch}_{\mathbf{W},\mathbf{0}}(\mathbf{y})$

5: $\mathbf{p}^0 = -\left[\mathbf{A}^\top(\mathbf{A}\mathbf{x} + \alpha\widehat{\mathbf{y}} - \mathbf{b}) + \beta^2\widehat{\mathbf{x}}\right]/\nu$

6: $\mathbf{q}^0 = -\alpha\mathsf{Ch}_{\mathbf{W},\mathbf{0}}(\mathbf{A}\mathbf{x} + \alpha\widehat{\mathbf{y}} - \mathbf{b})/\nu$

7: $\mathbf{x}^1 := \mathbf{x} + \mathbf{p}^0$

8: $\mathbf{y}^1 := \mathbf{y} + \mathbf{q}^0$

9: **for** $i = 1, \ldots, T-1$ **do**

10: $\quad \omega^{i-1} := \rho/\delta^{i-1}$

11: $\quad \delta^i := -(\nu + \omega^{i-1})$

12: $\quad \widehat{\mathbf{x}}^i = \mathsf{Ch}_{\mathbf{C},\mathbf{c}}(\mathbf{x}^i), \ \widehat{\mathbf{y}}^i = \mathsf{Ch}_{\mathbf{W},\mathbf{0}}(\mathbf{y}^i)$

13: $\quad \mathbf{p}^i := \left(\mathbf{A}^\top(\mathbf{A}\mathbf{x}^i + \alpha\widehat{\mathbf{y}}^i - \mathbf{b}) + \beta^2\widehat{\mathbf{x}}^i + \omega^{i-1}\mathbf{p}^{i-1}\right)/\delta^i$

14: $\quad \mathbf{q}^i = \left(\alpha\mathsf{Ch}_{\mathbf{W},\mathbf{0}}(\mathbf{A}\mathbf{x}^i + \alpha\widehat{\mathbf{y}}^i - \mathbf{b}) + \omega^{i-1}\mathbf{q}^{i-1}\right)/\delta^i$

15: $\quad \mathbf{x}^{i+1} := \mathbf{x}^i + \mathbf{p}^i$

16: $\quad \mathbf{y}^{i+1} := \mathbf{y}^i + \mathbf{q}^i$

17: **end for**

18: **Output:** $\begin{pmatrix}\mathbf{x}\\\mathbf{y}\end{pmatrix} - \begin{pmatrix}\mathbf{x}^T\\\mathbf{y}^T\end{pmatrix}$

---

---

**Algorithm 6** $\mathsf{Ch}_{\mathbf{K},\mathbf{v}}\left(\left(\begin{smallmatrix}\mathbf{x}\\\mathbf{y}\\\tilde{\mathbf{x}}\end{smallmatrix}\right)\right)$: Chebyshev residual oracle for Problem (P)

---

1: **Input:** $\begin{pmatrix}\mathbf{x}\\\mathbf{y}\\\tilde{\mathbf{x}}\end{pmatrix}, \begin{pmatrix}\mathbf{b}\\\mathbf{c}\\\tilde{\mathbf{c}}\end{pmatrix}$

2: $\left(\begin{smallmatrix}\widehat{\mathbf{x}}\\\widehat{\mathbf{y}}\end{smallmatrix}\right) := \mathsf{Ch}_{\mathbf{B},(\mathbf{b},\mathbf{c})}\left(\left(\begin{smallmatrix}\mathbf{x}\\\mathbf{y}\end{smallmatrix}\right)\right)$

3: $\widehat{\tilde{\mathbf{x}}} := \mathsf{Ch}_{\widetilde{\mathbf{B}},\tilde{\mathbf{c}}}(\tilde{\mathbf{x}})$

4: **Output:** $\mathsf{Ch}_{\mathbf{K},\mathbf{v}}\left(\left(\begin{smallmatrix}\mathbf{x}\\\mathbf{y}\\\tilde{\mathbf{x}}\end{smallmatrix}\right)\right) := \begin{pmatrix}\widehat{\mathbf{x}}\\\widehat{\mathbf{y}}\\\widehat{\tilde{\mathbf{x}}}\end{pmatrix}$

---

## D. Proof of Theorem 2.4

### D.1. Strongly convex case ($\mu > 0$)

To prove this case, we just recall a theorem on APAPC convergence from (Salim et al., 2022).

**Theorem D.1.** *((Salim et al., 2022)). Let Assumptions 2.1 and 2.2 hold. There exists a set of parameters for Algorithm 1 such that to yield $u^N$ satisfying $\left\| u^N - u^* \right\|_2^2 \leq \varepsilon$ it requires $N = O(\kappa_B \sqrt{L/\mu} \log(1/\varepsilon))$ iterations.*

### D.2. Convex case ($\mu = 0$)

Regularization can be used to adapt methods designed for strongly convex objectives to non-strongly convex problems. Let us define the regularized function as follows:

$$G^\nu(u) = G(u) + \frac{\nu}{2} \|u^0 - u\|^2. \tag{17}$$

The following lemma describes the accuracy needed for solution of the regularized problem.

**Lemma D.2 (new).** *Let $G : \mathbb{R}^d \to \mathbb{R}$ be convex and $L$-smooth function and suppose that there exists solution $u^* \in \underset{u:Bu=b}{\operatorname{Arg\,min}} G(u)$ and $u_\nu^* = \underset{u:Bu=b}{\operatorname{arg\,min}} G^\nu(u)$. Define $D = G(u^*) - \underset{u}{\min} G(u)$. Assume that $\|u^0 - u^*\|^2 \leq R^2$. Recall $\nu = \varepsilon/R^2$ from Theorem 2.4 and set*

$$\delta = \frac{\varepsilon^2}{32 \left( D + \frac{\varepsilon}{2} \right) \left( L + \frac{\varepsilon}{R^2} \right)}. \tag{18}$$

*If we have $\|u - u_\nu^*\|^2 \leq \delta$, then*

$$G(u) - G(u^*) \leq \varepsilon, \ \|Bu - b\|^2 \leq \delta \sigma_{\max}^2(B). \tag{19}$$

*In other words, it is sufficient to solve the regularized problem with accuracy $\delta = O(\varepsilon^2)$ in terms of convergence in argument to get an $\varepsilon$-solution of the initial problem.*

*Proof.* We have

$$
\begin{aligned}
G(u) - G(u^*) &\overset{(a)}{\leq} G^\nu(u) - G^\nu(u_\nu^*) + \frac{\nu}{2} \|u^0 - u^*\|^2 \\
&\overset{(b)}{\leq} \langle \nabla G^\nu(u_\nu^*), u - u_\nu^* \rangle + \frac{L+\nu}{2} \|u - u_\nu^*\|^2 + \frac{\nu}{2} \|u^0 - u^*\|^2 \\
&\overset{(c)}{\leq} \|\nabla G^\nu(u_\nu^*)\| \cdot \|u - u_\nu^*\| + \frac{L+\nu}{2} \|u - u_\nu^*\|^2 + \frac{\nu}{2} \|u^0 - u^*\|^2 \\
&\overset{(d)}{\leq} \sqrt{2(L+\nu) \left( G^\nu(u_\nu^*) - \underset{u}{\min} G^\nu(u) \right) \|u - u_\nu^*\|^2} + \frac{L+\nu}{2} \|u - u_\nu^*\|^2 + \frac{\nu}{2} \|u^0 - u^*\|^2 \\
&\overset{(e)}{\leq} \sqrt{2(L+\nu) \left( G^\nu(u^*) - \underset{u}{\min} G(u) \right) \|u - u_\nu^*\|^2} + \frac{L+\nu}{2} \|u - u_\nu^*\|^2 + \frac{\nu}{2} \|u^0 - u^*\|^2 \\
&\overset{(f)}{=} \sqrt{2(L+\nu) \left( G(u^*) - \underset{u}{\min} G(u) + \frac{\nu}{2} \|u^0 - u^*\|^2 \right) \|u - u_\nu^*\|^2} + \frac{L+\nu}{2} \|u - u_\nu^*\|^2 + \frac{\nu}{2} \|u^0 - u^*\|^2 \\
&\overset{(g)}{=} \sqrt{2(L+\nu) \left( D + \frac{\nu R^2}{2} \right) \delta} + \frac{(L+\nu)\delta}{2} + \frac{\nu R^2}{2} \\
&\overset{(h)}{\leq} \sqrt{2(L+\nu) \left( D + \frac{\varepsilon}{2} \right) \delta} + \frac{(L+\nu)\delta}{2} + \frac{\nu R^2}{2} \\
&\overset{(i)}{\leq} \frac{\varepsilon}{64} + \frac{\varepsilon}{4} + \frac{\varepsilon}{2} \\
&< \varepsilon,
\end{aligned}
$$

where (a) follows the definition of the regularized function $G^\nu$; (b) uses the $(L+\nu)$-smoothness of $G^\nu$; (c) uses the Cauchy–Schwarz inequality; (d) uses the convexity and smoothness of $G^\nu$; (e) uses the fact that $G(u) \leq G^\nu(u)$ for all

$u$; (f) uses the definition of $G^\nu$; (g) uses the definitions of $R$, $D$ and the assumption that $\|u - u_\nu^*\| \leq \delta$; (h) is due to the definition of $\nu$; (i) uses the definition of $\delta$ in (18).

Moreover,

$$\|Bu - b\|^2 = \|Bu - Bu_\nu^*\|^2 \leq \sigma_{\max}^2(B) \cdot \|u - u_\nu^*\|^2 \leq \sigma_{\max}^2(B) \cdot \delta = O(\sigma_{\max}^2(B)\varepsilon^2).$$

$\square$

From Proposition 1 in (Salim et al., 2022) and Lemma D.2, to achieve $G(u^k) - G(u^*) \leq \varepsilon$, APAPC (Algorithm 1) requires $\mathcal{O}\left(\kappa_B \sqrt{\frac{L+\nu}{\nu}} \log\left(\frac{1}{\delta}\right)\right)$ iterations.

From the definitions of $\delta$ in (18) and $\nu$ in Lemma D.2, we have:

$$\frac{1}{\delta} = \mathcal{O}\left(\frac{\left(M + \frac{\varepsilon}{2}\right)\left(L + \frac{\varepsilon}{R^2}\right)}{\varepsilon^2}\right), \quad \frac{L+\nu}{\nu} = 1 + \frac{L}{\nu} = 1 + \frac{LR^2}{\varepsilon} = \mathcal{O}\left(\frac{LR^2}{\varepsilon}\right)$$

Hence, the resulting complexity is $\mathcal{O}\left(\kappa_B \sqrt{\frac{LR^2}{\varepsilon}} \log\left(\frac{\left(M + \frac{\varepsilon}{2}\right)\left(L + \frac{\varepsilon}{R^2}\right)}{\varepsilon^2}\right)\right)$ iterations.

# E. Discussion on Gradient Sliding Method

## E.1. Preliminary: Gradient Sliding

Let us discuss the deterministic gradient sliding (GS) method (Lan, 2020, Algorithm 8.1); (Lan et al., 2020) that is used for optimization problems consisting of two summands to split the complexities. Consider problem

$$\min_{u \in U} K(u) := K_s(u) + K_n(u). \tag{20}$$

**Theorem E.1** (Lan (2020)). *Let $K_s$ satisfy Assumption 2.1 with strong convexity parameter $\mu = 0$ and Assumption 2.2, $K_n(u)$ satisfy Assumption 2.1 with $\mu = 0$ and Assumption 2.3 and $\left\|u^0 - u^*\right\|^2 \leq R^2$. Gradient sliding algorithm applied to problem (20) requires $N_s = O\left(\sqrt{\frac{LR^2}{\varepsilon}}\right)$ calls to gradient of $K_s$ and $N_n = O\left(\frac{M^2 R^2}{\varepsilon^2} + N_s\right)$ calls to subgradient of $K_n$ to yield $\widehat{u}$ such that $K(\widehat{u}) - K^* \leq \varepsilon$.*

By the method of Lagrange multipliers, problem (5) can be equivalently written as the following saddle point problem:

$$\min_{u \in U} \max_{v \in \mathbb{R}^p} \left[G(u) + \langle v, Bu - b\rangle\right] \tag{21}$$

**Lemma E.2** (Lan et al. (2020)). *Let $u^*$ be an optimal solution of (5). Then there exists an optimal dual multiplier $v^*$ for (21) such that*

$$\|v^*\| \leq R_{\text{dual}} := \frac{M}{\sigma_{\min+}(B)}. \tag{22}$$

**Lemma E.3** (Gorbunov et al. (2019)). *Let $\varepsilon > 0$ and $r = \frac{2R_{\text{dual}}^2}{\varepsilon}$. If $\widehat{u}$ is an $\varepsilon$-solution of (7), i.e.*

$$H_r(\widehat{u}) - \min_{u \in U} H_r(u) \leq \varepsilon,$$

*then*

$$G(\widehat{u}) - \min_{Bu-b=0} G(u) \leq \varepsilon, \quad \|B\widehat{u} - b\| \leq \frac{2\varepsilon}{R_{\text{dual}}}.$$

Although (Lan et al., 2020) and (Gorbunov et al., 2019) considered only consensus optimization ($B = \mathbf{W}$, $b = 0$), proofs of Lemmas E.2 and E.3 in referenced sources work without changes for any $B$ and $b \in \text{Im } B$.

## E.2. Proof of Theorem 2.5

*Proof.* Denote $K_n(u) = G(u)$ and $K_s(u) = \frac{R_{\text{dual}}^2}{\varepsilon}\|Bu\|_2^2$, implying $L = \frac{2R_{\text{dual}}^2 \sigma_{\max}^2(B)}{\varepsilon} \leq \frac{2M^2 \sigma_{\max}^2(B)}{\varepsilon \sigma_{\min+}^2(B)} = \frac{2M^2}{\varepsilon}\kappa_B$, where the inequality follows from Lemma E.2. Applying Theorem E.1 gives the desired result.

When $\mu > 0$, we simply substitute $L = \frac{2M^2}{\varepsilon}\kappa_B$ into the complexity of R-Sliding (Lan, 2020, Theorem 8.3) $N_s = O\left(\sqrt{\frac{L}{\mu}}\log\frac{1}{\varepsilon}\right)$, $N_n = O(\frac{M^2}{\mu\varepsilon} + N_s)$. Note, that for some reason (Lan, 2020, Section 8.1.3.1) only considers the case then the smooth component is strongly convex. But as it can be seen from the proof of (Lan, 2020, Theorem 8.3), it only uses strong convexity of the sum $K(u)$ of the smooth and the nonsmooth terms, thus we can apply it in our case where $K_n(u)$ is strongly convex, as was also done in (Dvinskikh & Gasnikov, 2021a; Uribe et al., 2020). $\square$

# F. Auxiliary Theorems and Lemmas for Section 3

## F.1. Identical Local Constraints

Let the constraint matrices $C_i$ be equal. Then the block-diagonal matrix of affine constraints has form $\mathbf{C} = I_n \otimes C$. The spectral properties of $\mathbf{B}^\top = [\mathbf{C}^\top \ \gamma\mathbf{W}]$ can be revisited in comparison with Lemma F.1.

**Lemma F.1** ((Rogozin et al., 2022, Lemma 1))**.**

$$\sigma^2_{\max}(\mathbf{B}) = \sigma^2_{\max}(C) + \gamma^2 \sigma^2_{\max}(W),$$
$$\sigma^2_{\min+}(\mathbf{B}) = \min\left\{\sigma^2_{\min+}(C), \gamma^2 \sigma^2_{\min+}(W)\right\}.$$

As a result, we obtain an enhanced bound on the number of communications w.r.t. Theorem 4.1.

**Theorem F.2** ((Rogozin et al., 2022))**.** *Applying Algorithm 1 to problem* (11) *with* $C_1 = \ldots = C_n = C$, *after applying* $\mathbf{W} \to P_W(\mathbf{W})$ *and* $\mathbf{C} \to P_C(\mathbf{C}^\top \mathbf{C})$, *we obtain a method that requires*

$$N_{\nabla f} = O\left(\sqrt{\kappa_f} \log\left(\frac{1}{\varepsilon}\right)\right) \text{ gradient calls,}$$
$$N_C = O\left(\sqrt{\kappa_f}\sqrt{\kappa_C} \log\left(\frac{1}{\varepsilon}\right)\right) \text{ mul. by } \mathbf{C} \text{ and } \mathbf{C}^\top,$$
$$N_W = O\left(\sqrt{\kappa_f}\sqrt{\kappa_W} \log\left(\frac{1}{\varepsilon}\right)\right) \text{ communications.}$$

*This upper bounds are optimal due to corresponding lower bounds.*

## F.2. Coupled Constraints

Let $A_i \in \mathbb{R}^{m \times d_i}$. A decentralized-friendly reformulation of coupled constraints is $\mathbf{Ax} + \mathbf{Wy} = \mathbf{b}$, where $\mathbf{y} \in \mathbb{R}^{mn}$. Then the problem writes as

$$\min_{\mathbf{x} \in \mathbb{R}^d} F(\mathbf{x}) := \sum_{i=1}^n f_i(x_i) \quad \text{s.t.} \quad \mathbf{Ax} + \mathbf{Wy} = \mathbf{b}. \tag{C}$$

Indeed, since range of $\mathbf{W}$ is $\{\mathbf{x} : x_1 + x_2 + \ldots x_n = 0\}$ we have $\mathbf{Ax} - \mathbf{b} \in \text{range } \mathbf{W} \Leftrightarrow \sum_{i=1}^n (A_i x_i - b_i) = 0$. The variable $\mathbf{y}$ is introduced to parametrize range $\mathbf{W}$. The cost of this, however, is that after addition of $\mathbf{y}$, the objective $F(\mathbf{x}, \mathbf{y})$ is no longer strongly convex. Thus, the essential technique to utilize strong convexity is to add augmented-Lagrangian-type penalization term in the objective with proper scaling (see, e.g. Lemma I.1).

The following definition is needed to describe convergence rates of decentralized algorithms for problems with coupled constraints.

**Lemma F.3** ((Yarmoshik et al., 2024b, Lemma 2))**.** *Denote* $\mathbf{B} = \begin{pmatrix} \mathbf{A} & \beta\mathbf{W} \end{pmatrix}$. *Setting* $\beta^2 = \frac{\lambda_{\min+}(S_A) + \sigma^2_{\max}(\mathbf{A})}{\sigma^2_{\min+}(\mathbf{W})}$, *we get*

$$\sigma^2_{\max}(\mathbf{B}) \leq \sigma^2_{\max}(\mathbf{A}) + (\sigma^2_{\max}(\mathbf{A}) + \lambda_{\min+}(S_A))\kappa^2_W,$$
$$\sigma^2_{\min+}(\mathbf{B}) \geq \frac{\lambda_{\min+}(S_A)}{2}.$$

**Theorem F.4** ((Yarmoshik et al., 2024b), Theorems 1 and 2)**.** *After penalizing* (C) *and applying Chebyshev accelerations* $\mathbf{W} \to P_W(\mathbf{W})$ *and then* $\mathbf{B} \to P_B(\mathbf{B}^\top \mathbf{B})$ *and applying Algorithm 1, we obtain a method that requires*

$$N_{\nabla f} = O\left(\sqrt{\kappa_f} \log\left(\frac{1}{\varepsilon}\right)\right) \text{ gradient calls,}$$
$$N_A = O\left(\sqrt{\kappa_f}\sqrt{\widehat{\kappa}_A} \log\left(\frac{1}{\varepsilon}\right)\right) \text{ mul. by } \mathbf{A} \text{ and } \mathbf{A}^\top,$$
$$N_W = O\left(\sqrt{\kappa_f}\sqrt{\widehat{\kappa}_A}\sqrt{\kappa_W} \log\left(\frac{1}{\varepsilon}\right)\right) \text{ communications.}$$

*This bound is optimal in a naturally defined class of decentralized first-order algorithms for problems with coupled constraints.*

# G. Proof of the Complexity Bounds for Shared Variable Constraints (Theorem 4.1)

## G.1. The upper bound

As announced in the theorem's statement, we first apply Chebyshev's preconditioning (Section 2.4) to matrix $\mathbf{W}$ and obtain matrix $\mathbf{W}' = P_W(\mathbf{W})$ with $\kappa_{\mathbf{W}'} = O(1)$. Then, by Lemma F.3, choosing $\gamma$ according to its statement, we obtain for $\widetilde{\mathbf{B}}^\top = \begin{pmatrix} \widetilde{\mathbf{C}} & \gamma \mathbf{W}' \end{pmatrix}$

$$\kappa_{\widetilde{\mathbf{B}}} = \frac{\sigma_{\max}^2(\widetilde{\mathbf{B}})}{\sigma_{\min^+}^2(\widetilde{\mathbf{B}})} = \frac{\sigma_{\max}^2(\widetilde{\mathbf{B}}^\top)}{\sigma_{\min^+}^2(\widetilde{\mathbf{B}}^\top)} \leq \frac{\sigma_{\max}^2(\widetilde{\mathbf{C}}^\top) + \left[\sigma_{\max}^2(\widetilde{\mathbf{C}}^\top) + \lambda_{\min^+}(S_{\widetilde{C}^\top})\right]\kappa_{\mathbf{W}'}^2}{\frac{1}{2}\lambda_{\min^+}(S_{\widetilde{C}^\top})} = O(\kappa_{\mathbf{W}'}^2, \widehat{\kappa}_{\widetilde{\mathbf{C}}}) = O(\widehat{\kappa}_{\widetilde{\mathbf{C}}}). \quad (23)$$

Finally, due to Theorem D.1, iteration complexity of Algorithm 1 applied to the problem

$$\min_{\mathbf{x} \in (\mathbb{R}^d)^n} F(\mathbf{x}) := \sum_{i=1}^n f_i(x_i) \text{ s.t. } \widetilde{\mathbf{B}}'x = \tilde{b}', \quad (24)$$

is $N = O(\sqrt{\kappa_f} \log \frac{1}{\varepsilon})$, where $\widetilde{\mathbf{B}}' = P_{\widetilde{B}}(\widetilde{\mathbf{B}}^\top \widetilde{\mathbf{B}})$, $\kappa_{\widetilde{\mathbf{B}}'} = O(1)$ and $\tilde{b}' = \frac{P_{\widetilde{B}}(\widetilde{\mathbf{B}}^\top \widetilde{\mathbf{B}})}{\widetilde{\mathbf{B}}^\top \widetilde{\mathbf{B}}} \widetilde{\mathbf{B}}^\top \begin{pmatrix} \tilde{c} \\ 0 \end{pmatrix}$.

Each iteration of Algorithm 1 in this case requires 1 computation of $\nabla f$, $O(\deg P_{\widetilde{\mathbf{B}}}) = O(\sqrt{\kappa_{\widetilde{\mathbf{B}}}}) = O(\sqrt{\widehat{\kappa}_{\widetilde{\mathbf{C}}}})$ multiplications by $\widetilde{\mathbf{B}}, \widetilde{\mathbf{B}}^\top$, each requiring $O(1)$ multiplication by $\widetilde{\mathbf{C}}, \widetilde{\mathbf{C}}^\top$ and $O(\deg P_W) = O(\sqrt{\kappa_W})$ multiplications by $\mathbf{W}$, i.e., communication rounds, which gives the first part of the theorem.

## G.2. The lower bound

This proof is a modification of the proof of (Yarmoshik et al., 2024b, Theorem 2). The main difference is in how we split a constraint matrix to obtain a splitting of the Nesterov's bad function $h(z) = \frac{1}{2}z^\top \mathbf{M}z + \alpha \|z\|_2^2 - z_1$ between nodes in Section G.2.4. In the original case of coupled constraints, matrix $\mathbf{M}$ is split into two summands $\mathbf{M} = \mathbf{E}_1^\top \mathbf{E}_2 + \mathbf{E}_2^\top \mathbf{E}_2$. Matrices $\mathbf{E}_1, \mathbf{E}_2$ are used to form matrices of the coupled constraint. Here we take a non-symmetric root matrix $\mathbf{E}$ such that $\mathbf{M} = \mathbf{E}^\top \mathbf{E}$ and split it into matrices $\mathbf{E}_1, \mathbf{E}_2$ by rows to form matrices of local constraints. For completeness, we describe here the whole construction of the lower bound, since its other parts, such as objective functions defined in Section G.2.3, also required changes (more subtle and technical, though) to be applied in this setup.

### G.2.1. DUAL PROBLEM

The proof relies on obtaining Nesterov's function as the objective of the *dual* problem. The primal and dual variables in our construction have similar component structure, what allows to derive the upper bound on accuracy of an approximate solution to the original problem from the explicit expression for the exact solution of the dual problem and an upper bound on the number of nonzero components in the approximate solution.

Let us derive the dual problem. Consider the primal problem with zero right-hand side in the constraints

$$\begin{aligned} \min_{x_1,\ldots,x_n \in \ell_2} & \sum_{i=1}^n f_i(x_i) \\ \text{s.t. } & \mathbf{C}_i x_i = 0 \quad \forall i = 1, \ldots, n. \\ & x_1 = \ldots = x_n. \end{aligned} \quad (25)$$

Simplifying the consensus constraint and combining the affine constraints to a single constraint $\tilde{\mathbf{C}}x = 0$ (e.g., by vertically stacking matrices $\mathbf{C}_i$), we rewrite the problem as

$$\begin{aligned} \min_{x \in \ell_2} & \sum_{i=1}^n f_i(x) \\ \text{s.t. } & \tilde{\mathbf{C}}x = 0. \end{aligned} \quad (26)$$

The dual problem has the form

$$\max_z \min_{x \in \ell_2} \left[ \sum_{i=1}^n f_i(x) - \left\langle z, \tilde{\mathbf{C}}x \right\rangle \right] = -\min_z F^*(\tilde{\mathbf{C}}^\top z), \tag{27}$$

where $F(x) = \sum_{i=1}^n f_i(x)$.

### G.2.2. EXAMPLE GRAPH

The graph construction is the same as in the lower bound for coupled constraints.

We follow the principle of lower bounds construction introduced in (Kovalev et al., 2021a) and take the example graph from (Scaman et al., 2017). Let the functions held by the nodes be organized into a path graph with $n$ vertices, where $n$ is divisible by 3. The nodes of graph $\mathcal{G} = (\mathcal{V}, \mathcal{E})$ are divided into three groups $\mathcal{V}_1 = \{1, \dots, n/3\}, \mathcal{V}_2 = \{n/3 + 1, \dots, 2n/3\}, \mathcal{V}_3 = \{2n/3 + 1, \dots, n\}$ of $n/3$ vertices each.

Now we recall the construction from (Scaman et al., 2017). Maximum and minimum eigenvalues of a path graph have form $\lambda_{\max}(W) = 2\left(1 + \cos\frac{\pi}{n}\right)$, $\lambda_{\min+}(W) = 2\left(1 - \cos\frac{\pi}{n}\right)$. Let $\beta_n = \frac{1+\cos\left(\frac{\pi}{n}\right)}{1-\cos\left(\frac{\pi}{n}\right)}$. Since $\beta_n \overset{n\to\infty}{\to} +\infty$, there exists $n = 3m \geq 3$ such that $\beta_n \leq \kappa_W < \beta_{n+3}$. For this $n$, introduce edge weights $w_{i,i+1} = 1 - a\mathbb{I}\{i = 1\}$, take the corresponding weighted Laplacian $W_a$ and denote its condition number $\kappa(W_a)$. If $a = 1$, the network is disconnected and therefore $\kappa(W_a) = \infty$. If $a = 0$, we have $\kappa(W_a) = \beta_n$. By continuity of Laplacian spectra we obtain that for some $a \in [0, 1)$ it holds $\kappa(W_a) = \kappa_W$. Note that $\pi/(n+3) \in [0, \pi/3]$, and for $x \in [0, \pi/3]$ we have $1 - \cos x \geq x^2/4$. We have

$$\kappa_W \leq \beta_{n+3} = \frac{1 + \cos\frac{\pi}{n+3}}{1 - \cos\frac{\pi}{n+3}} \leq \frac{72(n+3)^2}{\pi^2} \leq \frac{288n^2}{\pi^2} \leq 32n^2 \quad \Rightarrow \quad \sqrt{\kappa_W} \leq 4\sqrt{2}n = O(n). \tag{28}$$

### G.2.3. EXAMPLE FUNCTIONS

We let $e_1 = (1\ 0\ \dots\ 0)^\top$ denote the first coordinate vector, let $x$ be composed from two variable blocks $x = \begin{pmatrix} p \\ t \end{pmatrix}$. We set functions $f_i$ to be the same for all nodes, and define them as

$$f_i(p, t) = \frac{\mu_f}{2} \|p\|^2 + L_f \left\| t + \frac{L'_{\mathbf{C}}}{\mu_f} e_1 \right\|_2^2.$$

Correspondingly,

$$f_i^*(u, v) = \frac{1}{2\mu_f} \|u\|_2^2 + \frac{1}{L_f} \|v\|_2^2 - \frac{L'_{\mathbf{C}}}{\mu_f} v_1. \tag{29}$$

Let

$$\mathbf{E} = \begin{pmatrix} 1 & 0 & 0 & 0 & \cdots \\ -1 & 1 & 0 & 0 & \cdots \\ 0 & -1 & 1 & 0 & \cdots \\ 0 & 0 & -1 & 1 & \cdots \\ \vdots & \vdots & \vdots & \vdots & \ddots \end{pmatrix}, \quad \tilde{\mathbf{C}} = \left( -\sqrt{L'_{\mathbf{C}}}\mathbf{E}^\top \quad \sqrt{\mu'_{\mathbf{C}}}\mathbf{I} \right) \tag{30}$$

From (27) and (29), the dual problem is

$$\begin{aligned}
&\min_z \frac{L'_{\mathbf{C}}}{2\mu_f} \|\mathbf{E}z\|_2^2 + \frac{\mu'_{\mathbf{C}}}{L_f} \|z\|_2^2 - \frac{L'_{\mathbf{C}}}{\mu_f} z_1 \\
&= \min_z \frac{L'_{\mathbf{C}}}{2\mu_f} \langle z, \mathbf{M}z \rangle + \frac{\mu'_{\mathbf{C}}}{L_f} \|z\|_2^2 - \frac{L'_{\mathbf{C}}}{\mu_f} z_1 \\
&= \min_z \frac{L'_{\mathbf{C}}}{\mu_f} \left( \frac{1}{2}\langle z, \mathbf{M}z \rangle + \frac{\mu'_{\mathbf{C}}\mu_f}{L'_{\mathbf{C}}L_f} \|z\|_2^2 - z_1 \right),
\end{aligned} \tag{31}$$

where

$$\mathbf{M} = \mathbf{E}^\top \mathbf{E} = \begin{pmatrix} 2 & -1 & 0 & 0 & 0 & \cdots \\ -1 & 2 & -1 & 0 & 0 & \cdots \\ 0 & -1 & 2 & -1 & 0 & \cdots \\ \vdots & \vdots & \vdots & \vdots & \vdots & \ddots \end{pmatrix}.$$

Problem (31) is exactly the Nesterov's worst problem for smooth strongly convex minimization by first-order methods.

**Lemma G.1** ((Yarmoshik et al., 2024b, Lemma 6)). *The solution of Problem (31) is $z^* = \left\{\rho^k\right\}_{k=1}^{\infty}$, where*

$$\rho = \frac{\sqrt{\frac{2}{3}\frac{L_{\mathbf{C}}L_f}{\mu_{\mathbf{C}}\mu_f}+1}-1}{\sqrt{\frac{2}{3}\frac{L_{\mathbf{C}}L_f}{\mu_{\mathbf{C}}\mu_f}+1}+1}.$$

### G.2.4. EXAMPLE MATRICES

We split matrix $\tilde{\mathbf{C}}$ into two matrices as follows: even rows of $\tilde{\mathbf{C}}$ go to the first matrix, odd rows are filled with zeros; the second matrix is constructed in the same way from odd rows of $\tilde{\mathbf{C}}$. Formally, let $L_{\mathbf{C}}' = \frac{1}{2}L_{\mathbf{C}} - \frac{3}{2}\mu_{\mathbf{C}}$, $\mu_{\mathbf{C}}' = 3\mu_{\mathbf{C}}$, where $L_{\mathbf{C}} > 0$ and $\mu_{\mathbf{C}} > 0$ are any parameters such that $\widehat{\kappa}_{C^\top} = \frac{L_{\mathbf{C}}}{\mu_{\mathbf{C}}}$, and introduce

$$\mathbf{C}_i = \begin{cases} \left(\sqrt{L_{\mathbf{C}}'}\mathbf{E}_1^\top \quad \sqrt{\mu_{\mathbf{C}}'}\mathbf{I}_1\right), & i \in \mathcal{V}_1 \\ \left(\mathbf{0} \qquad\qquad \mathbf{0}\right), & i \in \mathcal{V}_2 , \\ \left(\sqrt{L_{\mathbf{C}}'}\mathbf{E}_2^\top \quad \sqrt{\mu_{\mathbf{C}}'}\mathbf{I}_2\right), & i \in \mathcal{V}_3 \end{cases}$$

$$\mathbf{E}_1^\top = \begin{pmatrix} 1 & -1 & 0 & 0 & 0 & \cdots \\ 0 & 0 & 0 & 0 & 0 & \cdots \\ 0 & 0 & 1 & -1 & 0 & \cdots \\ 0 & 0 & 0 & 0 & 0 & \cdots \\ \vdots & \vdots & \vdots & \vdots & \vdots & \ddots \end{pmatrix}, \quad \mathbf{I}_1 = \begin{pmatrix} 1 & 0 & 0 & 0 & \cdots \\ 0 & 0 & 0 & 0 & \cdots \\ 0 & 0 & 1 & 0 & \cdots \\ 0 & 0 & 0 & 0 & \cdots \\ \vdots & \vdots & \vdots & \vdots & \ddots \end{pmatrix}, \tag{32}$$

$$\mathbf{E}_2^\top = \begin{pmatrix} 0 & 0 & 0 & 0 & 0 & \cdots \\ 0 & 1 & -1 & 0 & 0 & \cdots \\ 0 & 0 & 0 & 0 & 0 & \cdots \\ 0 & 0 & 0 & 1 & -1 & \cdots \\ \vdots & \vdots & \vdots & \vdots & \vdots & \ddots \end{pmatrix}, \quad \mathbf{I}_2 = \begin{pmatrix} 0 & 0 & 0 & 0 & \cdots \\ 0 & 1 & 0 & 0 & \cdots \\ 0 & 0 & 0 & 0 & \cdots \\ 0 & 0 & 0 & 1 & \cdots \\ \vdots & \vdots & \vdots & \vdots & \ddots \end{pmatrix}. \tag{33}$$

It is clear, that so-defined local constraints are equivalent to $\tilde{\mathbf{C}}x = 0$.

Let us make sure that this choice of $\mathbf{C}_i$ indeed guarantees that $\frac{\max\limits_{i=1,\dots,n}\lambda_{\max}(C_i^\top C_i)}{\lambda_{\min+}(S_{C^\top})} \leq \widehat{\kappa}_{C^\top}$ as required by the statement of the theorem and Definition 3.4.

For the numerator we have

$$\max_i \lambda_{\max}(\mathbf{C}_i^\top \mathbf{C}_i) = \max_i \lambda_{\max}(\mathbf{C}_i\mathbf{C}_i^\top) = \lambda_{\max}\left(L_{\mathbf{C}}'\mathbf{E}_1^\top \mathbf{E}_1 + \mu_{\mathbf{C}}'\mathbf{I}\right) = 2L_{\mathbf{C}}' + \mu_{\mathbf{C}}' = L_{\mathbf{C}}. \tag{34}$$

For the denominator direct calculation yields

$$
\mathbf{C}_1^\top \mathbf{C}_1 = \begin{pmatrix}
1 & -1 & 0 & 0 & \cdots & 1 & 0 & 0 & 0 & \cdots \\
-1 & 1 & 0 & 0 & \cdots & -1 & 0 & 0 & 0 & \cdots \\
0 & 0 & 1 & -1 & \cdots & 0 & 0 & 1 & 0 & \cdots \\
0 & 0 & -1 & 1 & \cdots & 0 & 0 & -1 & 0 & \cdots \\
\vdots & \vdots & \vdots & \vdots & \cdots & \vdots & \vdots & \vdots & \vdots & \cdots \\
1 & -1 & 0 & 0 & \cdots & 1 & 0 & 0 & 0 & \cdots \\
0 & 0 & 0 & 0 & \cdots & 0 & 0 & 0 & 0 & \cdots \\
0 & 0 & 1 & -1 & \cdots & 0 & 0 & 1 & 0 & \cdots \\
0 & 0 & 0 & 0 & \cdots & 0 & 0 & 0 & 0 & \cdots \\
\vdots & \vdots & \vdots & \vdots & \cdots & \vdots & \vdots & \vdots & \vdots & \ddots
\end{pmatrix}, \quad
\mathbf{C}_2^\top \mathbf{C}_2 = \begin{pmatrix}
0 & 0 & 0 & 0 & \cdots & 0 & 0 & 0 & 0 & \cdots \\
0 & 1 & -1 & 0 & \cdots & 0 & 1 & 0 & 0 & \cdots \\
0 & -1 & 1 & 0 & \cdots & 0 & -1 & 0 & 0 & \cdots \\
0 & 0 & 0 & 1 & \cdots & 0 & 0 & 0 & 1 & \cdots \\
\vdots & \vdots & \vdots & \vdots & \cdots & \vdots & \vdots & \vdots & \vdots & \cdots \\
0 & 0 & 0 & 0 & \cdots & 0 & 0 & 0 & 0 & \cdots \\
0 & 1 & -1 & 0 & \cdots & 0 & 1 & 0 & 0 & \cdots \\
0 & 0 & 0 & 0 & \cdots & 0 & 0 & 0 & 0 & \cdots \\
0 & 0 & 0 & 1 & \cdots & 0 & 0 & 0 & 1 & \cdots \\
\vdots & \vdots & \vdots & \vdots & \cdots & \vdots & \vdots & \vdots & \vdots & \ddots
\end{pmatrix}.
$$
$$(35)$$

Using this, we get

$$
\lambda_{\min^+}\left(\frac{1}{n}\sum_{i=1}^n \mathbf{C}_i^\top \mathbf{C}_i\right) = \frac{1}{3}\lambda_{\min^+}\left(\begin{matrix} L_\mathbf{C}'(\mathbf{E}_1\mathbf{E}_1^\top + \mathbf{E}_2\mathbf{E}_2^\top) & \sqrt{L_\mathbf{C}'\mu_\mathbf{C}'}(\mathbf{E}_1\mathbf{I}_1 + \mathbf{E}_2\mathbf{I}_2) \\ \sqrt{L_\mathbf{C}'\mu_\mathbf{C}'}(\mathbf{I}_1^\top\mathbf{E}_1 + \mathbf{I}_2^\top\mathbf{E}_2) & \mu_\mathbf{C}'(\mathbf{I}_1^\top\mathbf{I}_1 + \mathbf{I}_2^\top\mathbf{I}_2) \end{matrix}\right)
$$

$$
= \frac{1}{3}\lambda_{\min^+}\left(\begin{matrix} L_\mathbf{C}'\mathbf{L}_\text{path} & \sqrt{L_\mathbf{C}'\mu_\mathbf{C}'}\mathbf{E} \\ \sqrt{L_\mathbf{C}'\mu_\mathbf{C}'}\mathbf{E}^\top & \mu_\mathbf{C}'\mathbf{I} \end{matrix}\right)
$$

$$
= \frac{1}{3}\lambda_{\min^+}\left(\begin{pmatrix} \sqrt{L_\mathbf{C}'}\mathbf{E} \\ \sqrt{\mu_\mathbf{C}'}\mathbf{I} \end{pmatrix}\begin{pmatrix} \sqrt{L_\mathbf{C}'}\mathbf{E}^\top & \sqrt{\mu_\mathbf{C}'}\mathbf{I} \end{pmatrix}\right)
$$

$$
= \frac{1}{3}\lambda_{\min^+}\left(\begin{pmatrix} \sqrt{L_\mathbf{C}'}\mathbf{E}^\top & \sqrt{\mu_\mathbf{C}'}\mathbf{I} \end{pmatrix}\begin{pmatrix} \sqrt{L_\mathbf{C}'}\mathbf{E} \\ \sqrt{\mu_\mathbf{C}'}\mathbf{I} \end{pmatrix}\right)
$$

$$
= \frac{1}{3}\lambda_{\min^+}\left(L_\mathbf{C}'\mathbf{M} + \mu_\mathbf{C}'\mathbf{I}\right) = \frac{\mu_\mathbf{C}'}{3} = \mu_\mathbf{C},
$$

where $\mathbf{L}_\text{path} = \mathbf{E}\mathbf{E}^\top$ is the Laplacian of infinite (in one direction) path graph, which differs from $\mathbf{M}$ only in the first diagonal component $\mathbf{L}_\text{path}[1,1] = 1$.

### G.2.5. BOUNDING ACCURACY

We consider the class of first-order decentralized algorithms for problems with local affine constraints defined as follows

**Definition G.2** ((Yarmoshik et al., 2024a, Definition 1)). Denote $\mathcal{M}_i(k)$, where $\mathcal{M}_i(0) = \{x_i^0\}$, as the local memory of the $i$-th node at step $k$. The set of allowed actions of a first order decentralized algorithm at step $k$ is restricted to the three options

1. Local computation: $\mathcal{M}_i(k) = \text{span}\left(\{x, \nabla f_i(x), \nabla f_i^*(x) : x \in \mathcal{M}_i(k)\}\right)$;

2. Decentralized communication with immediate neighbours: $\mathcal{M}_i(k) = \text{span}\left(\{\mathcal{M}_j(k) : \text{edge } (i,j) \in E\}\right)$.

3. Matrix multiplication: $\mathcal{M}_i(k) = \text{span}\left(\{b_i, \mathbf{C}_i^\top\mathbf{C}_i x : x \in \mathcal{M}_i(k)\}\right)$.

After each step $k$, an algorithm must provide a current approximate solution $x_i^k \in \mathcal{M}_i(k)$ and set $\mathcal{M}_i(k+1) = \mathcal{M}_i(k)$.

Without loss of generality, we can assume $x_i^0 = 0$. Recall that we split $x$ in two variable blocks $p$ and $t$. From the structure of matrix $\mathbf{C}_1^\top\mathbf{C}_1$ it is clear, that a node $i \in \mathcal{V}_1$ on $k$-th step can only increase the number of nonzero components in $p_i^k$ or $t_i^k$ by one if the index of the last nonzero component in $p_i^k$ or $t_i^k$ is odd. Similarly, by structure of $\mathbf{C}_2^\top\mathbf{C}_2$, nodes from $\mathcal{V}_2$ can only "unlock" next zero component if it is odd. Thus, to increase the number of nonzero components in $p_i^k$ or $t_i^k$ by two, the information must be transmitted from $\mathcal{V}_1$ to $\mathcal{V}_2$ and back (or vice versa), what requires to perform $2n/3 = \Omega(\sqrt{\kappa_W})$ decentralized communication rounds and two matrix multiplications.

Due to the strong duality, the solution of problem (25) can be obtained from the solution of its dual (27) as

$$
x^*(z^*) = \begin{pmatrix} p^* \\ t^* \end{pmatrix} (\mathbf{C}^\top z^*) = \begin{pmatrix} \dfrac{\sqrt{L_\mathbf{C}'}}{\mu_f}\mathbf{E}z^* \\ \dfrac{\sqrt{\widehat{\mu}_\mathbf{A}}}{2L_f}\left(z^* - \dfrac{L_\mathbf{C}'}{\mu_f}e_1\right) \end{pmatrix}.
$$

Therefore $t^*$ is just a scaled version of $z^*$ up to the first component. Lemma G.1 and the standard calculation (see (Yarmoshik et al., 2024b, Appendix C.4)) then leads to the following bound on the number $q$ of nonzero components in $t_i^k$ required to reach the accuracy $\left\| x_i^k - x_i^* \right\|_2^2 \leq \varepsilon$:

$$q \geq \Omega \left( \sqrt{\frac{L_{\mathbf{C}} L_f}{\mu_{\mathbf{C}} \mu_f}} \log \left( \frac{1}{\varepsilon} \right) \right), \tag{36}$$

what translates to the required number of matrix multiplications

$$N_{\mathbf{C}} \geq \Omega \left( \sqrt{\frac{L_{\mathbf{C}} L_f}{\mu_{\mathbf{C}} \mu_f}} \log \left( \frac{1}{\varepsilon} \right) \right), \tag{37}$$

and decentralized communication rounds

$$N_{\mathbf{W}} \geq \Omega \left( \sqrt{\kappa_W} \sqrt{\frac{L_{\mathbf{C}} L_f}{\mu_{\mathbf{C}} \mu_f}} \log \left( \frac{1}{\varepsilon} \right) \right). \tag{38}$$

The lower bound on the number of gradient computations is obtained using the same sum-trick as in (Yarmoshik et al., 2024b): to each $f_i(x_i)$ we add an independent copy of Nestrov's worst function $h_i(w_i)$ with appropriate parameters (variables $w_i$ are new independent variables without any coupling between different nodes).

# H. Proof of Lemma 4.4

First, the squared maximum singular value of a block matrix is upper bounded by the sum of the squared maximum singular values of its blocks, therefore

$$\sigma_{\max}^2(\mathbf{B}) \leq \sigma_{\max}^2(\mathbf{A}) + \gamma^2 \sigma_{\max}^2(\mathbf{W}) + \beta^2 \sigma_{\max}^2(\mathbf{C}).$$

We set coefficients $\alpha$ and $\beta$ to

$$\alpha^2 = \frac{1}{\mu_W} \begin{cases} L_A + \frac{1}{4}\widetilde{\mu}_{AC}, & \widetilde{\mu}_{AC} > 0 \\ 2L_A, & \widetilde{\mu}_{AC} = 0 \end{cases}, \quad \beta^2 = \frac{1}{\mu_C} \begin{cases} L_S + \frac{1}{2}\widetilde{\mu}_{AC}, & \widetilde{\mu}_{AC} > 0 \\ L_S + 2L_A, & \widetilde{\mu}_{AC} = 0 \end{cases}, \tag{39}$$

where $L_S = \frac{1}{n}\sigma_{\max}^2(\mathbf{A}')$.

We are going to prove the following bound on minimal positive singular value of $\mathbf{B}$:

$$\sigma_{\min+}^2(\mathbf{B}) \geq \begin{cases} \frac{1}{4}\widetilde{\mu}_{AC}, & \widetilde{\mu}_{AC} > 0, \\ L_A, & \widetilde{\mu}_{AC} = 0. \end{cases} \tag{40}$$

**Proof of the lower bound on $\sigma_{\min+}^2(\mathbf{B})$**

Since $\sigma_{\min+}^2(\mathbf{B}) = \sigma_{\min+}^2(\mathbf{B}^\top)$ we will bound the latter. image We have $\ker^\perp \mathbf{B}^\top = \operatorname{Im}\mathbf{B} = \operatorname{Im}\begin{pmatrix}\mathbf{A}\\\mathbf{C}\end{pmatrix} + \begin{pmatrix}\mathcal{L}_m^\perp\\0\end{pmatrix}$. Consider arbitrary $z \in \ker^\perp \mathbf{B}^\top$. Using $\operatorname{Im}\mathbf{W} = \mathcal{L}_m^\perp$, we can represent $z$ as $z = \begin{pmatrix}\mathbf{A}\\\mathbf{C}\end{pmatrix}\xi + \begin{pmatrix}\mathbf{P}_{\mathcal{L}_m^\perp}\\0\end{pmatrix}\eta$ for some $\xi, \eta$.

The key technique of the proof is to decompose $z$ into three orthogonal components $z = \begin{pmatrix}u\\0\end{pmatrix} + \begin{pmatrix}v\\0\end{pmatrix} + \begin{pmatrix}0\\w\end{pmatrix}$, where $u = \mathbf{P}_{\mathcal{L}_m^\perp}(\mathbf{A}\xi + \mathbf{P}_{\mathcal{L}_m^\perp}\eta)$, $v = \mathbf{P}_{\mathcal{L}_m}(\mathbf{A}\xi + \mathbf{P}_{\mathcal{L}_m^\perp}\eta) = \mathbf{P}_{\mathcal{L}_m}\mathbf{A}\xi$ and $w = \mathbf{C}\xi$.

The following relations trivially follow from the definition of the decomposition:

$$u \in \mathcal{L}_m^\perp \tag{41a}$$
$$v \in \mathcal{L}_m \tag{41b}$$
$$u + v \in \operatorname{Im}\mathbf{A} + \mathcal{L}_m^\perp \tag{41c}$$
$$w \in \operatorname{Im}\mathbf{C} \tag{41d}$$
$$\begin{pmatrix}v\\w\end{pmatrix} \in \operatorname{Im}\begin{pmatrix}\mathbf{P}_{\mathcal{L}_m}\mathbf{A}\\\mathbf{C}\end{pmatrix} = \ker^\perp\begin{pmatrix}\mathbf{A}^\top\mathbf{P}_{\mathcal{L}_m} & \mathbf{C}^\top\end{pmatrix} \tag{41e}$$

Following (Yarmoshik et al., 2024b, Lemma 2) and using the relations above we bound $\left\|\mathbf{B}^\top z\right\|_2^2$ as

$$\left\|\mathbf{B}^\top z\right\|_2^2 \overset{(a)}{=} \left\|\begin{pmatrix}\mathbf{A}^\top(u+v) + \mathbf{C}^\top w\\\mathbf{W}u\end{pmatrix}\right\|_2^2 \overset{(b)}{\geq} -L_A\|u\|_2^2 + \frac{1}{2}\left\|\mathbf{A}^\top v + \mathbf{C}^\top w\right\|_2^2 + \mu_W\|u\|_2^2, \tag{42}$$

where (a) is due to (41b); (b) is due to Young's inequality, (41a) and definitions of $L_A$, $\mu_W$ (16).

Now consider the second term in the rhs. By (41b) we have $\mathbf{A}^\top v = \mathbf{A}^\top \mathbf{P}_{\mathcal{L}_m} v$. Then, denoting $\mathbf{J} = \begin{pmatrix}\mathbf{P}_{\mathcal{L}_m}\mathbf{A}\\\mathbf{C}\end{pmatrix}$ and using (41e) we obtain

$$\left\|\mathbf{A}^\top v + \mathbf{C}^\top w\right\|_2^2 = \left\|\mathbf{A}^\top \mathbf{P}_{\mathcal{L}_m} v + \mathbf{C}^\top w\right\|_2^2 \geq \mu_{\mathbf{J}}\left\|\begin{pmatrix}v\\w\end{pmatrix}\right\|_2^2. \tag{43}$$

To estimate $\mu_{\mathbf{J}}$, we consider a vector $t \in \ker^\perp \mathbf{J} = \operatorname{Im}\mathbf{A}^\top\mathbf{P}_{\mathcal{L}_m} + \operatorname{Im}\mathbf{C}^\top$ and decompose it into orthogonal components as $t = y + q$, where $y = \mathbf{P}_{\ker^\perp \mathbf{C}} t$ and $q = \mathbf{P}_{\ker \mathbf{C}} t$. Then we apply Young's inequality

$$\|\mathbf{J}t\|_2^2 = \|\mathbf{P}_{\mathcal{L}_m}\mathbf{A}(y+q)\|_2^2 + \|\mathbf{C}t\|_2^2 \geq -\sigma_{\max}^2(\mathbf{P}_{\mathcal{L}_m}\mathbf{A})\|y\|_2^2 + \frac{1}{2}\|\mathbf{P}_{\mathcal{L}_m}\mathbf{A}q\|_2^2 + \mu_C\|y\|_2^2. \tag{44}$$

Let us bound the first and the second terms in the rhs. First, $\|\mathbf{P}_{\mathcal{L}_m}\mathbf{A}y\|_2^2 = \left\|\frac{1}{n}\mathbf{1}_n \otimes \mathbf{A}'y\right\|_2^2 = \frac{1}{n}\|\mathbf{A}'y\|_2^2 \leq \frac{1}{n}\sigma_{\max}^2(\mathbf{A}')\|y\|_2^2$ for any $y$. Since $\frac{1}{n}\sigma_{\max}^2(\mathbf{A}') = \sigma_{\max}\left(\frac{1}{n}\mathbf{1}_n\sum_{i=1}^n \mathbf{A}_i\mathbf{A}_i^\top\right) = \sigma_{\max}(\mathbf{S})$, we denote the coefficient by $L_S$. Second, by definitions of $q$ and $t$ we have $q = \mathbf{P}_{\ker \mathbf{C}}\, t \in \operatorname{Im}\mathbf{P}_{\ker \mathbf{C}}\,\mathbf{A}^\top\mathbf{P}_{\mathcal{L}_m} = \operatorname{Im}\mathbf{P}_{\ker \mathbf{C}}\,\mathbf{A}'^\top = \ker^\perp \mathbf{A}'\mathbf{P}_{\ker \mathbf{C}}$. Therefore, $\|\mathbf{P}_{\mathcal{L}_m}\mathbf{A}q\|_2^2 = \left\|\frac{1}{n}\mathbf{1}_n \otimes \mathbf{A}'q\right\|_2^2 = \left\|\frac{1}{n}\mathbf{1}_n \otimes \mathbf{A}'\mathbf{P}_{\ker \mathbf{C}}\, q\right\|_2^2 \geq \frac{1}{n}\sigma_{\min+}^2(\mathbf{A}'\mathbf{P}_{\ker \mathbf{C}})\|q\|_2^2 = \widetilde{\mu}_{AC}\|q\|_2^2$. Here we allow $\widetilde{\mu}_{AC}$ to be equal to zero if $\mathbf{A}'\mathbf{P}_{\ker \mathbf{C}}$ is the zero matrix. Summarizing, we get

$$\|\mathbf{J}t\|_2^2 \geq -L_S\|y\|_2^2 + \frac{1}{2}\widetilde{\mu}_{AC}\|q\|_2^2 + \mu_C\|y\|_2^2. \tag{45}$$

Now let us utilize the properties of scaling coefficients (39). If $\widetilde{\mu}_{AC} = 0$, we have $t = y$ and $\|\mathbf{J}t\|_2^2 \geq (\mu_C - L_S)\|t\|_2^2$, thus ensuring $\mu_C \geq L_S + 2L_A$ we get $\mu_J \geq 2L_A$. Otherwise, if $\widetilde{\mu}_{AC} > 0$, for $\mu_C \geq L_S + \frac{1}{2}\widetilde{\mu}_{AC}$ we obtain $\mu_J = \frac{1}{2}\widetilde{\mu}_{AC}$, which cannot be further increased by scaling $\mathbf{C}$ in this case.

Plugging this into (42) yields

$$\sigma_{\min+}^2(\mathbf{B}) \geq \begin{cases} L_A, & \widetilde{\mu}_{AC} = 0, \\ \frac{1}{4}\widetilde{\mu}_{AC}, & \widetilde{\mu}_{AC} > 0, \end{cases} \tag{46}$$

where we assume $\mu_W \geq L_A + \begin{cases} L_A, & \widetilde{\mu}_{AC} = 0 \\ \frac{1}{4}\widetilde{\mu}_{AC}, & \widetilde{\mu}_{AC} > 0 \end{cases}$.

**Bounds for $\kappa_B$**

Finally, to simplify the expression for the condition number of $\mathbf{B}$ we use the following bounds:

$$\begin{aligned}
\widetilde{\mu}_{AC} &= \frac{1}{n}\sigma_{\min+}^2(\mathbf{A}'\mathbf{P}_{\ker \mathbf{C}}) \leq \frac{1}{n}\sigma_{\max}^2(\mathbf{A}'\mathbf{P}_{\ker \mathbf{C}}) \leq \frac{1}{n}\sigma_{\max}^2(\mathbf{A}')\sigma_{\max}^2(\mathbf{P}_{\ker \mathbf{C}}) \\
&= \frac{1}{n}\sigma_{\max}^2(\mathbf{A}') \leq \frac{1}{n}\sum_{i=1}^n \sigma_{\max}^2(\mathbf{A}_i) \leq \sigma_{\max}^2(\mathbf{A}) = L_A,
\end{aligned} \tag{47}$$

and

$$L_S = \frac{1}{n}\sigma_{\max}^2(\mathbf{A}') \leq L_A. \tag{48}$$

For $\widetilde{\mu}_{AC} > 0$ this gives

$$\kappa_B = \frac{L_B}{\mu_B} \leq \frac{4L_A}{\widetilde{\mu}_{AC}} + \frac{4L_A + \widetilde{\mu}_{AC}}{\widetilde{\mu}_{AC}}\frac{L_W}{\mu_W} + \frac{4L_S + 2\widetilde{\mu}_{AC}}{\widetilde{\mu}_{AC}}\frac{L_C}{\mu_C} = O\left(\widetilde{\kappa}_{AC}\kappa_W + \widetilde{\kappa}_{AC}\kappa_C\right), \tag{49}$$

and for $\widetilde{\mu}_{AC} = 0$

$$\kappa_B = \frac{L_B}{\mu_B} \leq \frac{L_A}{L_A} + \frac{L_A + L_A}{L_A}\frac{L_W}{\mu_W} + \frac{L_S + 2L_A}{L_A}\frac{L_C}{\mu_C} = O\left(\kappa_W + \kappa_C\right). \tag{50}$$

# I. Proof of the Complexity Bounds for Coupled and Local Constraints (Theorem 4.5)

## I.1. The upper bound

Since coupled constraints require introducing auxiliary variable $y$ on which the objective function $F(\mathbf{x})$ does not depend, and therefore is not strongly convex with respect to the whole set of variables what does not allow to apply Algorithm 1. Therefore we introduce a regularized/penalized (in the augmented Lagrangian fashion) objective

$$G(\mathbf{x}, \mathbf{y}) = \sum_{i=1}^{n} f_i(x_i) + \frac{r}{2} \|\mathbf{A}\mathbf{x} + \alpha \mathbf{W}\mathbf{y} - \mathbf{b}\|_2^2, \quad r = \frac{\mu_f}{2L_A}. \tag{51}$$

The next lemma shows that $G(\mathbf{x}, \mathbf{y})$ fixes the strong convexity problem.

**Lemma I.1** (essentially (Yarmoshik et al., 2024b, Lemma 1))**.** $G(\mathbf{x}, \mathbf{y})$ *is* ($\mu_G = \mu_f/4$)*-strongly convex on* $\mathbb{R}^d \times \mathcal{Y}$ *and* ($L_G = 2L_f \kappa_W^2$)*-smooth, where* $\mathcal{Y}$ *is the subspace of all* $\mathbf{y} \in R^{mn}$ *(recall that* $A_i \in \mathbb{R}^{m \times d_i}$*) such that* $\langle \mathbf{y}, \mathbf{1}_{mn} \rangle = 0$.

*Proof.* Let $D_G(\mathbf{x}', \mathbf{y}'; \mathbf{x}, \mathbf{y})$ denote the Bregman divergence of $G$:

$$D_G(\mathbf{x}', \mathbf{y}'; \mathbf{x}, \mathbf{y}) = G(\mathbf{x}', \mathbf{y}') - G(\mathbf{x}, \mathbf{y}) - \langle \nabla_x G(\mathbf{x}, \mathbf{y}), \mathbf{x}' - \mathbf{x} \rangle - \langle \nabla_y G(\mathbf{x}, \mathbf{y}), \mathbf{y}' - \mathbf{y} \rangle. \tag{52}$$

The value of $\mu_G$ can be obtained as follows:

$$
\begin{aligned}
D_G(x', y'; x, y) &= D_F(x'; x) + \frac{r}{2} \|\mathbf{A}(x' - x) + \alpha \mathbf{W}(y' - y)\|_2^2 \\
&\overset{(a)}{\geq} \frac{\mu_f}{2} \|x' - x\|_2^2 + \frac{r}{4} \|\alpha \mathbf{W}(y' - y)\|_2^2 - \frac{r}{2} \|\mathbf{A}(x' - x)\|_2^2 \\
&\overset{(b)}{\geq} \frac{\mu_f}{2} \|x' - x\|_2^2 + \frac{r\alpha^2 \mu_{\mathbf{W}}}{4} \|y' - y\|_2^2 - \frac{rL_{\mathbf{A}}}{2} \|x' - x\|_2^2 \\
&\geq \frac{\mu_f}{4} \|x' - x\|_2^2 + \frac{\mu_f \alpha^2 \mu_{\mathbf{W}}}{8L_{\mathbf{A}}} \|y' - y\|_2^2, \\
&\overset{(c)}{\geq} \frac{\mu_f}{8} \left\| \begin{pmatrix} x' - x \\ y' - y \end{pmatrix} \right\|^2,
\end{aligned}
$$

where (a) is due to Young's inequality; (b) is due to $y' - y \in \mathcal{Y}$ and $\ker^{\perp} \mathbf{W} = \mathcal{Y}$; (c) is because $\alpha^2 \geq \frac{L_A}{\mu_W}$ by its definition in (39).

The value of $L_G$ can be obtained as follows:

$$
\begin{aligned}
D_G(x', y'; x, y) &= D_F(x'; x) + \frac{r}{2} \|\mathbf{A}(x' - x) + \alpha \mathbf{W}(y' - y)\|_2^2 \\
&\overset{(a)}{\leq} \frac{L_f}{2} \|x' - x\|_2^2 + r \|\alpha \mathbf{W}(y' - y)\|_2^2 + r \|\mathbf{A}(x' - x)\|_2^2 \\
&\leq \frac{L_f}{2} \|x' - x\|_2^2 + r\alpha^2 L_{\mathbf{W}} \|y' - y\|_2^2 + rL_{\mathbf{A}} \|x' - x\|_2^2 \\
&\leq \frac{L_f + \mu_f}{2} \|x' - x\|_2^2 + \frac{\mu_f \alpha^2 L_{\mathbf{W}}}{2L_{\mathbf{A}}} \|y' - y\|_2^2, \\
&\overset{(b)}{\leq} L_f \frac{L_{\mathbf{W}}}{\mu_{\mathbf{W}}} \left\| \begin{pmatrix} x' - x \\ y' - y \end{pmatrix} \right\|^2,
\end{aligned}
$$

where (a) is due to Young's inequality; (b) is because $\alpha^2 \leq 2\frac{L_A}{\mu_W}$ by its definition in (39) and $\mu_f \leq L_f$. $\qquad \square$

Thus we replace $\sum_{i=1}^{n} f_i(x_i)$ with $G(\mathbf{x}, \mathbf{y})$ in the decentralized-friendly reformulation of problem (C&L)

$$
\min_{\mathbf{x}, \mathbf{y}} \sum_{i=1}^{n} f_i(x_i)
$$

$$
\text{s.t. } \mathbf{B} \begin{pmatrix} \mathbf{x} \\ \mathbf{y} \end{pmatrix} = \begin{pmatrix} \mathbf{b} \\ \beta \mathbf{c} \end{pmatrix}, \quad \mathbf{B} = \begin{pmatrix} \mathbf{A} & \alpha \mathbf{W} \\ \beta \mathbf{C} & 0 \end{pmatrix}.
$$

Note, that this does not change the minimizer of the problem.

Next we apply Chebyshev's preconditioning (Section 2.4) and replace matrices $\mathbf{W}$ and $\mathbf{C}$ with $\mathbf{W}' = P_W(\mathbf{W})$, $\kappa_{W'} = O(1)$ and $\mathbf{C}' = P_C(\mathbf{C}^\top \mathbf{C})$, $\kappa_{C'} = O(1)$ correspondingly so that $\mathbf{B} = \begin{pmatrix} \mathbf{A} & \alpha \mathbf{W}' \\ \beta \mathbf{C}' & 0 \end{pmatrix}$. Then, by Lemma 4.4, choosing $\alpha$ and $\beta$ as defined in (39), we obtain $\kappa_B = O(\widetilde{\kappa}_{AC}\kappa_{W'} + \widetilde{\kappa}_{AC}\kappa_{C'}) = O(\widetilde{\kappa}_{AC})$, and by Lemma I.1 $\kappa_G = \frac{8L_f \kappa_{W'}^2}{\mu_f} = O(\kappa_f)$.

Finally, due to Theorem D.1, iteration complexity of Algorithm 1 applied to the following equivalent reformulation of the problem (C&L)

$$\min_{\mathbf{x} \in \mathbb{R}^d, \mathbf{y} \in \mathbb{R}^{mn}} G(\mathbf{x}, \mathbf{y}) \text{ s.t. } \mathbf{B}' \begin{pmatrix} \mathbf{x} \\ \mathbf{y} \end{pmatrix} = \mathbf{b}', \tag{53}$$

where $\mathbf{B}' = P_B(\mathbf{B}^\top \mathbf{B})$, $\kappa_{\mathbf{B}'} = O(1)$ and $\mathbf{b}' = \frac{P_B(\mathbf{B}^\top \mathbf{B})}{\mathbf{B}^\top \mathbf{B}} \mathbf{B}^\top \begin{pmatrix} \mathbf{b} \\ \mathbf{c}' \end{pmatrix}$, $\mathbf{c}' = \beta \frac{P_C(\mathbf{C}^\top \mathbf{C})}{\mathbf{C}^\top \mathbf{C}} \mathbf{C}^\top \mathbf{c}$ is $N = O(\sqrt{\kappa_G} \log \frac{1}{\varepsilon}) = O(\sqrt{\kappa_f} \log \frac{1}{\varepsilon})$.

Each iteration of Algorithm 1 in this case requires 1 computation of $\nabla f$, $O(\deg P_B) = O(\sqrt{\kappa_B}) = O(\sqrt{\widetilde{\kappa}_{AC}})$ multiplications by $\mathbf{B}, \mathbf{B}^\top$, each requiring $O(1)$ multiplication by $\mathbf{A}, \mathbf{A}^\top$, $O(\deg P_C) = O(\sqrt{\kappa_C})$ multiplication by $\mathbf{C}, \mathbf{C}^\top$ and $O(\deg P_W) = O(\sqrt{\kappa_W})$ multiplications by $\mathbf{W}$, i.e., communication rounds, what gives the first part of the theorem.

Note, that we correctly applied Lemma I.1 here despite the condition $\mathbf{y} \in \mathcal{Y}$ because when being started from $\mathbf{y} = 0$ Algorithm 1 will keep all iterates $\mathbf{y}^k$ within the subspace $\mathcal{Y}$ due to $\operatorname{Im} \mathbf{W} = \operatorname{Im} \mathbf{W}' = \mathcal{Y}$.

### I.2. The lower bound

Our worst problem example falls in the case $\widetilde{\mu}_{AC} > 0$.

We use the proof of the lower bound for coupled constraints from (Yarmoshik et al., 2024b, Theorem 2) combined with the idea of the two-level communication graph used in the proof of the lower bound for local constraints (Yarmoshik et al., 2024a, Theorem 2).

Let $W \in \mathbb{R}^{n \times n}$ be defined as in Appendix G.2.2. Let $W_C$ be also a Laplacian matrix with $\frac{\lambda_{\max}(W_C)}{\lambda_{\min+}(W_C)} = \kappa_C$ constructed as in Appendix G.2.2, and denote the number of vertices in the corresponding path graph as $l$, so that $W_C \in \mathbb{R}^{l \times l}$.

Consider the following objective function

$$f_i(p_i, t_i) = \frac{\mu_f}{2} \frac{1}{l} \sum_{j=1}^{l} \left\| p_{ij} + \frac{\sqrt{L_A'}}{2\mu_f} e_1 \right\|_2^2 + \frac{L_f}{2} \frac{1}{l} \sum_{j=1}^{l} \|t_{ij}\|_2^2, \tag{54}$$

where $x_i = (p_i, t_i)$, $p_i$ and $t_i$ are $\in \ell_2^l$, and $e_1 = (1\ 0\ \ldots\ 0)^\top$ denote the first coordinate vector. We immediately note that each $f_i$ is $L_f/l$-smooth and $\mu_f/l$-strongly convex, thus the condition number of $\sum_{i=1}^{n} f_i(x_i)$ is indeed $\kappa_f = \frac{L_f}{\mu_f}$.

Define $C_i = \begin{pmatrix} \sqrt{W_C \otimes I_{\ell_2}} & 0 \\ 0 & \sqrt{W_C \otimes I_{\ell_2}} \end{pmatrix}$ and $b_i = 0$ for all $i \in \{1, \ldots, n\}$. This gives the desired condition number:

$\frac{\sigma_{\max}^2(C)}{\sigma_{\min+}^2(C)} = \frac{\lambda_{\max}(W_C)}{\lambda_{\min+}(W_C)} = \kappa_C$. The local constraint $C_i \begin{pmatrix} p_i \\ t_i \end{pmatrix} = 0$ expresses two consensus constraints: indeed, the first block is equivalent to $(W_C \otimes I_{\ell_2})p_i = 0$ and, in turn, to $p_{i,1} = \ldots = p_{i,l}$, since $\ker W_C = \{\alpha 1_l \mid \alpha \in \mathbb{R}\}$; same, the second block gives $t_{i,1} = \ldots = t_{i,l}$.

Recall, that, by construction, $n$ and $l$ are divisible by 3. We define index sets $\mathcal{V}_1 = \{1, \ldots, n/3\}, \mathcal{V}_2 = \{n/3 + 1, \ldots, 2n/3\}, \mathcal{V}_3 = \{2n/3 + 1, \ldots, n\}$, and, similarly, $\mathcal{U}_1 = \{1, \ldots, l/3\}, \mathcal{U}_2 = \{l/3 + 1, \ldots, 2l/3\}, \mathcal{U}_3 =$

$\{2l/3 + 1, \ldots, l\}$. Let

$$
E_1 = \begin{pmatrix} 1 & 0 & 0 & 0 & 0 & \ldots \\ 0 & 1 & -1 & 0 & 0 & \ldots \\ 0 & 0 & 0 & 0 & 0 & \ldots \\ 0 & 0 & 0 & 1 & -1 & \ldots \\ \vdots & \vdots & \vdots & \vdots & \vdots & \ddots \end{pmatrix}, \quad E_2 = \begin{pmatrix} 1 & -1 & 0 & 0 & 0 & \ldots \\ 0 & 0 & 0 & 0 & 0 & \ldots \\ 0 & 0 & 1 & -1 & 0 & \ldots \\ 0 & 0 & 0 & 0 & 0 & \ldots \\ \vdots & \vdots & \vdots & \vdots & \vdots & \ddots \end{pmatrix},
$$

$$
A_i = \begin{cases} \left( \mathrm{diag}(\delta_{\mathcal{U}_1}) \otimes \sqrt{L'_A} E_1^\top \quad \mathrm{diag}(\delta_{\mathcal{U}_1}) \otimes \sqrt{\mu'_A} I_{\ell_2^l} \right), & i \in \mathcal{V}_1 \\ 0, & i \in \mathcal{V}_2 \\ \left( \mathrm{diag}(\delta_{\mathcal{U}_3}) \otimes \sqrt{L'_A} E_2^\top \quad \mathrm{diag}(\delta_{\mathcal{U}_3}) \otimes \sqrt{\mu'_A} I_{\ell_2^l} \right), & i \in \mathcal{V}_3 \end{cases},
$$

where $\delta_{\mathcal{U}_k} \in \mathbb{R}^l$ and $[\delta_{\mathcal{U}_k}]_j = \begin{cases} 1, & j \in \mathcal{U}_k \\ 0, & \text{otherwise} \end{cases}$.

Note, that if we presolve the local consensus constraints, we obtain exactly the same problem as in (Yarmoshik et al., 2024b, Appendix C.3). Therefore, the solution to the problem considered here is the solution of the problem in (Yarmoshik et al., 2024b, Appendix C) copied $l$ times at each vertex. By Definition 4.2

$$
\widetilde{\kappa}_{AC} = \frac{\max\limits_{i=1,\ldots,n} \lambda_{\max}(A_i A_i^\top)}{\frac{1}{n} \sigma^2_{\min+}\left(\mathbf{A}' \mathbf{P}_{\ker \mathbf{C}}\right)} \tag{55}
$$

To estimate $\max\limits_{i=1,\ldots,n} \lambda_{\max}(A_i A_i^\top)$ we rearrange columns of $A_i$ so that it becomes a block-diagonal matrix

$$
A_i = \begin{cases} \mathrm{diag}(\delta_{\mathcal{U}_1}) \otimes \left( \sqrt{L'_A} E_1^\top \quad \sqrt{\mu'_A} I_{\ell_2^l} \right), & i \in \mathcal{V}_1 \\ 0, & i \in \mathcal{V}_2 \\ \mathrm{diag}(\delta_{\mathcal{U}_3}) \otimes \left( \sqrt{L'_A} E_2^\top \quad \sqrt{\mu'_A} I_{\ell_2^l} \right), & i \in \mathcal{V}_3 \end{cases} =: \begin{cases} \mathrm{diag}(\delta_{\mathcal{U}_1}) \otimes \bar{A}_1, & i \in \mathcal{V}_1 \\ 0, & i \in \mathcal{V}_2 \\ \mathrm{diag}(\delta_{\mathcal{U}_3}) \otimes \bar{A}_2, & i \in \mathcal{V}_3 \end{cases} \tag{56}
$$

and by the proof of (Yarmoshik et al., 2024b, Theorem 2), maximal squared singular values of its blocks are upper bounded by $\sigma^2_{\max}(\bar{A}_k) \le 2L'_A + \mu'_A = L_A$, $k \in \{1, 2\}$ as needed, if we set $L'_A = \frac{1}{2}L_A - \frac{9}{2}\mu_A$, $\mu'_A = 9\widetilde{\mu}_{AC}$.

For the denominator of (55) we have $\ker C_i = \{(p_i, t_i) : p_{i1} = \ldots = p_{il}, \; t_{i1} = \ldots = t_{il}\}$, $\mathbf{P}_{\ker C_i} = \begin{pmatrix} 1 & 0 \\ 0 & 1 \end{pmatrix} \otimes \left( \frac{1}{l} 1_l 1_l^\top \otimes I_{\ell_2} \right)$, thus, calculating $\mathbf{A}' \mathbf{P}_{\ker \mathbf{C}}$ and rearranging its columns as in (56) we obtain

$$
\mathbf{A}' \mathbf{P}_{\ker \mathbf{C}} = \begin{pmatrix} 1_{|\mathcal{V}_1|}^\top \otimes \frac{1}{l} 1_{|\mathcal{U}_1|} 1_l^\top \otimes \bar{A}_1 & 0 \\ 0 & 1_{|\mathcal{V}_2|}^\top \otimes \frac{1}{l} 1_{|\mathcal{U}_2|} 1_l^\top \otimes \bar{A}_2 \end{pmatrix}. \tag{57}
$$

Consider, for example, the first nonzero diagonal block

$$
\frac{1}{n} \sigma^2_{\min+}\left( 1_{|\mathcal{V}_1|}^\top \otimes \frac{1}{l} 1_{|\mathcal{U}_1|} 1_l^\top \otimes \bar{A}_1 \right) = \frac{|\mathcal{V}_1||\mathcal{U}_1|l}{nl^2} \sigma^2_{\min+}(\bar{A}_1) = \frac{1}{9} \sigma^2_{\min+}(\bar{A}_1) \ge \frac{1}{9} \mu'_A = \widetilde{\mu}_{AC}. \tag{58}
$$

The same holds for the block with $\bar{A}_2$, thus setting $\widetilde{\mu}_{AC} = 1$ and $L_A = \widetilde{\kappa}_{AC}$ the condition number of $A$ is indeed $\widetilde{\kappa}_{AC}$.

With this we verified that the considered problem instance satisfies the assumption on the values of objective function and constraint matrices condition numbers. Now let us obtain the lower bounds for the oracle complexities of solving this problem instances by any first-order method.

The solution of the problem has the property that the number of nonzero components $\nu = \max\{k : \exists i, j : [p_{i,j}]_k > 0\}$ in an approximate solution $x$ is lower bounded by the squared distance $\varepsilon$ between $x$ and the exact solution $x^*$ as $\nu \ge \Omega\left(\sqrt{\kappa_A \kappa_f} \log\left(\frac{1}{\varepsilon}\right)\right)$ (Yarmoshik et al., 2024b, Theorem 2). Assuming without loss of generality that the starting point for the algorithm is zero, it is clear from the structure of the problem (and, especially, from the structure of $E_1$) that nodes in $\mathcal{V}_1$ cannot increase the maximum number of nonzero components in any of $x_{i,j}$ by themselves for more than 1 by performing

any operation allowed for a first order method, i.e. primal/dual gradient computation, multiplications by $C_i, C_i^\top, A_i, A_i^\top$ and decentralized communication (as well as nodes in $\mathcal{V}_3$; nodes in $\mathcal{V}_2$ cannot increase $\nu$ by local operations). Thus the information is needed to be transferred from $\mathcal{V}_1$ to $\mathcal{V}_3$ (or vice versa) in order to increase the number of nonzero components by $O(1)$ what costs $n/3 \geq \Omega\left(\sqrt{\kappa_W}\right)$ communication rounds. This part of the proof coincides with the case of coupled constraints.

Now, the difference introduced with local constraints is that for nodes in $\mathcal{V}_1$ multiplication by matrix $E_1$ (which is the only way for them to increase $\nu$) is performed only for $j \in \mathcal{U}_1$, while $E_2$ applies to $j \in \mathcal{U}_3$. Therefore, any algorithm is required to transfer the information between $\mathcal{U}_1$ and $\mathcal{U}_3$ to increase $\nu$ by 1, what costs $l/3 \geq \Omega\left(\sqrt{\kappa_C}\right)$ multiplications by $C_i, C_i^\top$.

Bringing this all together, we obtain

$$N_A \geq \Omega\left(\sqrt{\widetilde{\kappa}_{AC}\kappa_f}\log\left(\frac{1}{\varepsilon}\right)\right) \tag{59}$$

$$N_W \geq \Omega\left(\sqrt{\kappa_W\widetilde{\kappa}_{AC}\kappa_f}\log\left(\frac{1}{\varepsilon}\right)\right) \tag{60}$$

$$N_C \geq \Omega\left(\sqrt{\kappa_C\widetilde{\kappa}_{AC}\kappa_f}\log\left(\frac{1}{\varepsilon}\right)\right) \tag{61}$$

The bound on $N_{\nabla f}$ is obtained by taking the sum of the constructed objectives $f_i(x_i)$ with independent copies of Nestrov's worst functions $h_i(w_i)$ with appropriate parameters on each node (variables $w_i$ are new independent variables without any coupling between different nodes).

## J. Proof of the Upper Bounds for Mixed Constraints (Theorem 4.6)

Define $\mathbf{K} = \mathrm{diag}(\mathbf{B}, \widetilde{\mathbf{B}})$, where $\mathbf{B}$ is the matrix of coupled and local constraints $\mathbf{B} = \begin{pmatrix} \mathbf{A} & \alpha W \otimes I_m \\ \beta \mathbf{C} & 0 \end{pmatrix}$, $\mathbf{A} = \mathrm{diag}(A_1, \ldots, A_n)$, and $\widetilde{\mathbf{B}}$ is the matrix of consensus and local constraints $\widetilde{\mathbf{B}} = \begin{pmatrix} \widetilde{\mathbf{C}} \\ \gamma W \otimes I_d \end{pmatrix}$.

Since $\mathbf{K}$ is block-diagonal, we obtain its preconditioning by separately preconditioning $\mathbf{B}_1$ as described in Section I.1, and $\mathbf{B}_2$ as described in Appendix G.1.

### J.1. Non-identical local constraints

*Proof.* We apply Chebyshev's preconditioning (Section 2.4) and replace matrices $W$ and $\mathbf{C}$ with $W' = P_W(W)$, $\kappa_{W'} = O(1)$ and $\mathbf{C}' = P_C(\mathbf{C}^\top \mathbf{C})$, $\kappa_{C'} = O(1)$ respectively, so that $\mathbf{B}_1 = \begin{pmatrix} \mathbf{A} & \alpha W' \otimes I_m \\ \beta \mathbf{C}' & 0 \end{pmatrix}$ and $\mathbf{B}_2^\top = \begin{pmatrix} \widetilde{\mathbf{C}}^\top & \gamma W' \otimes I_d \end{pmatrix}$.

Then we replace $\mathbf{B}_1$ with $\mathbf{B}_1' = P_{\mathbf{B}_1}(\mathbf{B}_1^\top \mathbf{B}_1)$, $\kappa_{\mathbf{B}_1'} = O(1)$, and $\mathbf{B}_2$ with $\mathbf{B}_2' = P_{\mathbf{B}_2}(\mathbf{B}_2^\top \mathbf{B}_2)$, $\kappa_{\mathbf{B}_2'} = O(1)$. This preconditioning gives us $\sigma_{\min^+}(\mathbf{B}_1'), \sigma_{\min^+}(\mathbf{B}_2') \geq 11/15$ and $\sigma_{\max}(\mathbf{B}_1'), \sigma_{\max}(\mathbf{B}_2') \leq 19/15$ (Salim et al., 2022, Section 6.3.2) thus for $\mathbf{K} = \mathrm{diag}(\mathbf{B}_1', \mathbf{B}_2')$ we get $\kappa_K = O(1)$.

To fix the strong convexity issues with coupled constraints (see Section I.1), we penalize/regularize the objective: $G(\mathbf{x}, \tilde{\mathbf{x}}, \mathbf{y}) = \sum_{i=1}^n f_i(x_i, \tilde{x}_i) + \frac{r}{2} \|\mathbf{A}\mathbf{x} + \alpha \mathbf{W}'\mathbf{y} - \mathbf{b}\|_2^2$. By (Yarmoshik et al., 2024b, Lemma 1) for $r = \frac{\mu_f}{2L_A}$ we have $\kappa_G = O(\kappa_f)$.

Now, due to Theorem D.1, iteration complexity of Algorithm 1 applied to the following equivalent reformulation of problem (P)

$$\min_{\mathbf{x}, \tilde{\mathbf{x}}, \mathbf{y}} G(\mathbf{x}, \tilde{\mathbf{x}}, \mathbf{y}) \text{ s.t. } \mathbf{K} \begin{pmatrix} \mathbf{x} \\ \mathbf{y} \\ \tilde{\mathbf{x}} \end{pmatrix} = \begin{pmatrix} \mathbf{b}' \\ \mathbf{c}' \end{pmatrix}, \tag{62}$$

where $\mathbf{b}' = \frac{P_{\mathbf{B}_1}(\mathbf{B}_1^\top \mathbf{B}_1)}{\mathbf{B}_1^\top \mathbf{B}_1} \mathbf{B}_1^\top \begin{pmatrix} \mathbf{b} \\ \mathbf{c} \end{pmatrix}$, $\mathbf{c}' = \frac{P_{\mathbf{B}_2}(\mathbf{B}_2^\top \mathbf{B}_2)}{\mathbf{B}_2^\top \mathbf{B}_2} \mathbf{B}_2^\top \begin{pmatrix} \tilde{\mathbf{c}} \\ 0 \end{pmatrix}$ is $N = O(\sqrt{\kappa_G} \log \frac{1}{\varepsilon}) = O(\sqrt{\kappa_f} \log \frac{1}{\varepsilon})$. Each iteration of Algorithm 1 in this case requires

- 1 computation of $\nabla f$;

- $O(1)$ multiplications by $\mathbf{B}_1', \mathbf{B}_1'^\top$ equivalent to $O(\deg P_{\mathbf{B}_1}) = O(\sqrt{\kappa_{\mathbf{B}_1}}) = O(\sqrt{\kappa_{AC}})$ multiplications by $\mathbf{B}_1, \mathbf{B}_1^\top$, each requiring $O(1)$ multiplications by $\mathbf{A}, \mathbf{A}^\top$, $O(\deg P_W) = O(\sqrt{\kappa_W})$ multiplications by $\mathbf{W}$ (communications) and $O(\deg P_C) = O(\sqrt{\kappa_C})$ multiplications by $\mathbf{C}, \mathbf{C}^\top$;

- $O(1)$ multiplications by $\mathbf{B}_2, \mathbf{B}_2'^\top$ equivalent to $O(\deg P_{\mathbf{B}_2}) = O(\sqrt{\kappa_{\widetilde{\mathbf{C}}^\top}})$ multiplications by $\mathbf{B}_2, \mathbf{B}_2^\top$ each requiring $O(1)$ multiplications by $\widetilde{\mathbf{C}}, \widetilde{\mathbf{C}}^\top$ and $O(\deg P_W) = O(\sqrt{\kappa_W})$ communications.

Counting total number of matrix multiplications concludes the proof. □

### J.2. Identical local constraints

In this case we have $\widetilde{\mathbf{C}} = I_n \otimes \tilde{C}$.

*Proof.* The difference is in the application of Chebyshev's preconditioning: instead of taking polynomial of $\mathbf{B}_2$ we simply replace matrix $\widetilde{\mathbf{C}}$ with $\widetilde{\mathbf{C}}' = P_C(\mathbf{C})$, $\kappa_{\widetilde{C}'} = O(1)$, what by Lemma F.1 with $\gamma = 1$ also gives that $\mathbf{B}_2^\top = \begin{pmatrix} I_n \otimes C'^\top & \gamma W' \otimes I_d \end{pmatrix}$ has $\sigma_{\min^+}(\mathbf{B}_2), \sigma_{\max}(\mathbf{B}_2)$ both bounded as $\Theta(1)$. $\mathbf{B}_1, W$ and objective function are treated the same way as in the case of non-identical constraints.

Each iteration of Algorithm 1 in this case requires

- (same as in the non-identical case) 1 computation of $\nabla f$;

- (same as in the non-identical case) $O(1)$ multiplications by $\mathbf{B}'_1, \mathbf{B}'^\top_1$ equivalent to $O(\deg P_{\mathbf{B}_1}) = O(\sqrt{\kappa_{\mathbf{B}_1}}) = O(\sqrt{\widehat{\kappa}_A})$ multiplications by $\mathbf{B}_1, \mathbf{B}^\top_1$, each requiring $O(1)$ multiplications by $\mathbf{A}, \mathbf{A}^\top$, $O(\deg P_W) = O(\sqrt{\kappa_W})$ multiplications by $\mathbf{W}$ (communications) and $O(\deg P_C) = O(\sqrt{\kappa_C})$ multiplications by $\mathbf{C}, \mathbf{C}^\top$;

- (reduced complexity by $W$) $O(1)$ multiplications by $\mathbf{B}_2, \mathbf{B}'^\top_2$ each requiring $O(\deg P_{\widetilde{\mathbf{C}}}) = O(\sqrt{\widehat{\kappa}_C})$ multiplications by $\mathbf{C}, \mathbf{C}^\top$ and $O(\deg P_W) = O(\sqrt{\kappa_W})$ communications.

$\square$

For the case when $\mathbf{C} = 0$ we also give the following useful lemma, which allows to precisely estimate the condition number of matrix $\mathbf{B}$ without applying Chebyshev's preconditioning, but only by scaling matrices with scalars.

**Lemma J.1.** *Let* $\widetilde{C}_1 = \ldots = \widetilde{C}_n = \widetilde{C}$. *Denote* $\mathbf{B} = \operatorname{diag}(\alpha \mathbf{B}_1, \mathbf{B}_2)$, *where* $\mathbf{B}_1 = \begin{pmatrix} \mathbf{A} & \beta W \otimes I_m \end{pmatrix}$, $\mathbf{A} = \operatorname{diag}(A_1, \ldots, A_n)$ *and* $\mathbf{B}^\top_2 = \begin{pmatrix} I_n \otimes \widetilde{C}^\top & \gamma W \otimes I_d \end{pmatrix}$. *Also recall definitions of* $S_A$ *and* $\widehat{\kappa}_A$ *from Definition 3.4. Setting* $\alpha^2 = \frac{2\sigma^2_{\min+}(\widetilde{C})}{\lambda_{\min+}(S_A)}$, $\beta^2 = \frac{\lambda_{\min+}(S_A) + \sigma^2_{\max}(\mathbf{A})}{\sigma^2_{\min+}(W)}$ *and* $\gamma^2 = \frac{\sigma^2_{\min+}(\widetilde{C})}{\sigma^2_{\min+}(W)}$, *we obtain*

$$\sigma^2_{\max}(\mathbf{B}) \leq O\left(\sigma^2_{\max}(\widetilde{C}) + \sigma^2_{\min+}(\widetilde{C}) \cdot \kappa^2_W \widehat{\kappa}_A\right),$$
$$\sigma^2_{\min+}(\mathbf{B}) \geq \sigma^2_{\min+}(\widetilde{C}),$$

*thus* $\kappa_B = O(\kappa_{\widetilde{C}} + \kappa^2_W \widehat{\kappa}_A)$.

*Proof.* Let us separately estimate the spectrum of $\mathbf{B}_1$ and $\mathbf{B}_2$. Set $\beta^2 = \frac{\lambda_{\min+}(S_A) + \sigma^2_{\max}(\mathbf{A})}{\sigma^2_{\min+}(W)}$. According to Lemma F.3, we have

$$\sigma^2_{\max}(\mathbf{B}_1) \leq \sigma^2_{\max}(\mathbf{A}) + (\sigma^2_{\max}(\mathbf{A}) + \lambda_{\min+}(S_A))\kappa^2_W,$$
$$\sigma^2_{\min+}(\mathbf{B}_1) \geq \frac{\lambda_{\min+}(S_A)}{2}.$$

By Lemma F.1 we obtain

$$\sigma^2_{\max}(\mathbf{B}_2) = \sigma^2_{\max}(\widetilde{C}) + \gamma^2 \sigma^2_{\max}(W),$$
$$\sigma^2_{\min+}(\mathbf{B}_2) = \min(\sigma^2_{\min+}(\widetilde{C}), \gamma^2 \sigma^2_{\min+}(W)).$$

Recalling the values of $\alpha^2$ and $\gamma^2$ from the statement of the lemma, we obtain the estimate on $\sigma^2_{\max}(\mathbf{B})$:

$$
\begin{aligned}
\sigma^2_{\max}(\mathbf{B}) &\leq \max(\alpha^2 \sigma^2_{\max}(\mathbf{B}_1), \sigma^2_{\max}(\mathbf{B}_2)) \\
&= \max\left(2\sigma^2_{\min+}(\widetilde{C})\left(\widehat{\kappa}_A + (\widehat{\kappa}_A + 1)\kappa^2_W\right), \sigma^2_{\max}(\widetilde{C}) + \sigma^2_{\min+}(\widetilde{C})\kappa^2_W, \right) \\
&= O\left(\sigma^2_{\max}(\widetilde{C}) + \sigma^2_{\min+}(\widetilde{C})\kappa^2_W \widehat{\kappa}_A\right).
\end{aligned}
$$

Analogously we get a bound on $\sigma^2_{\min+}(\mathbf{B})$:

$$
\begin{aligned}
\sigma^2_{\min+}(\mathbf{B}) &\geq \min(\alpha^2 \sigma^2_{\min+}(\mathbf{B}_1), \sigma^2_{\min+}(\mathbf{B}_2)) \\
&= \min\left(\frac{\alpha^2 \lambda_{\min+}(S_A)}{2}, \sigma^2_{\min+}(\widetilde{C}), \gamma^2 \sigma^2_{\min+}(W)\right) \\
&= \sigma^2_{\min+}(\widetilde{C}).
\end{aligned}
$$

$\square$

## K. Missing Proofs From Section 5.1

| PROB. | ORACLE | COMPL. |
|---|---|---|
| COUPLED CONSTR. (C) | GRAD. | $\sqrt{\frac{L_f R^2}{\varepsilon}}$ |
| | MAT. | $\sqrt{\widehat{\kappa}_A}\sqrt{\frac{L_f R^2}{\varepsilon}}\log\left(\frac{1}{\varepsilon}\right)$ |
| | COMM. | $\sqrt{\kappa_W}\sqrt{\widehat{\kappa}_A}\sqrt{\frac{L_f R^2}{\varepsilon}}\log\left(\frac{1}{\varepsilon}\right)$ |
| | PAPER | THIS PAPER, TH. 5.1 |
| SHARED VAR. CONSTR. (S) | GRAD. | $\sqrt{\frac{L_f R^2}{\varepsilon}}$ |
| | MAT. $\widetilde{C}$ | $\sqrt{\widehat{\kappa}_{\widetilde{C}^\top}}\sqrt{\frac{L_f R^2}{\varepsilon}}\log\left(\frac{1}{\varepsilon}\right)$ |
| | COMM. | $\sqrt{\widehat{\kappa}_{\widetilde{C}^\top}}\sqrt{\kappa_W}\sqrt{\frac{L_f R^2}{\varepsilon}}\log\left(\frac{1}{\varepsilon}\right)$ |
| | PAPER | THIS PAPER, TH. 5.1 |
| LOCAL VAR. CONSTR. (C&L) | GRAD. | $\sqrt{\frac{L_f R^2}{\varepsilon}}$ |
| | MAT. $A$ | $\sqrt{\widetilde{\kappa}_{AC}}\sqrt{\frac{L_f R^2}{\varepsilon}}\log\left(\frac{1}{\varepsilon}\right)$ |
| | MAT. $C$ | $\sqrt{\widetilde{\kappa}_{AC}}\sqrt{\kappa_C}\sqrt{\frac{L_f R^2}{\varepsilon}}\log\left(\frac{1}{\varepsilon}\right)$ |
| | COMM. | $\sqrt{\widetilde{\kappa}_{AC}}\sqrt{\kappa_W}\sqrt{\frac{L_f R^2}{\varepsilon}}\log\left(\frac{1}{\varepsilon}\right)$ |
| | PAPER | THIS PAPER, TH. 5.1 |
| GENERAL MIXED CONSTR. (P) | GRAD. | $\sqrt{\frac{L_f R^2}{\varepsilon}}$ |
| | MAT. $A$ | $\sqrt{\widetilde{\kappa}_{AC}}\sqrt{\frac{L_f R^2}{\varepsilon}}\log\left(\frac{1}{\varepsilon}\right)$ |
| | MAT. $C$ | $\sqrt{\widetilde{\kappa}_{AC}}\sqrt{\kappa_C}\sqrt{\frac{L_f R^2}{\varepsilon}}\log\left(\frac{1}{\varepsilon}\right)$ |
| | MAT. $\widetilde{C}$ | $\sqrt{\widehat{\kappa}_{\widetilde{C}^\top}}\sqrt{\frac{L_f R^2}{\varepsilon}}\log\left(\frac{1}{\varepsilon}\right)$ |
| | COMM. | $\left(\sqrt{\widetilde{\kappa}_{AC}}+\sqrt{\widehat{\kappa}_{\widetilde{C}^\top}}\right)\sqrt{\kappa_W}\sqrt{\frac{L_f R^2}{\varepsilon}}\log\left(\frac{1}{\varepsilon}\right)$ |
| | PAPER | THIS PAPER, TH. 5.1 |

*Table 6.* Convergence rates for decentralized smooth convex optimization with constraints. Constants $\widehat{\kappa}_A, \widetilde{\kappa}_{AC}, \widehat{\kappa}_{C^\top}, \kappa_C$ have the same meaning as in Table 1.

### K.1. Proof of Theorem 5.1

By (Dvinskikh & Gasnikov, 2021b, Section 3) the Penalty Similar Triangles Method (PSTM) produces an $\varepsilon$-solution using $N_{\nabla f} = \mathcal{O}\left(\sqrt{\frac{LR^2}{\varepsilon}}\right)$ gradient computations and $N_{\mathbf{B}} = \mathcal{O}\left(\sqrt{\frac{LR^2}{\varepsilon}}\sqrt{\kappa_{\mathbf{B}}}\log\left(\frac{1}{\varepsilon}\right)\right)$ matrix multiplications with $\mathbf{B}$ and $\mathbf{B}^\top$. Alternatively, from Theorem 2.4, we can use Algorithm 1 to obtain an $\varepsilon$-solution, which requires $N_{\nabla f} = \mathcal{O}\left(\sqrt{\frac{LR^2}{\varepsilon}}\log\left(\frac{1}{\varepsilon}\right)\right)$ gradient computations and $N_{\mathbf{B}} = \mathcal{O}\left(\sqrt{\frac{LR^2}{\varepsilon}}\sqrt{\kappa_{\mathbf{B}}}\log\left(\frac{1}{\varepsilon}\right)\right)$ matrix multiplications with $\mathbf{B}$ and $\mathbf{B}^\top$. Therefore, using Algorithm 1 will give the same complexity for matrix multiplications and communications, but the gradient complexity will have an extra $\log\left(\frac{1}{\varepsilon}\right)$ factor.

Follow the Lemma 4.4 with $\alpha$, $\beta$ are chosen as in (39), we obtain

$$\kappa_{\mathbf{B}} = O\left(\widetilde{\kappa}_{AC}\kappa_W + \widetilde{\kappa}_{AC}\kappa_C\right).$$

The Chebyshev preconditioning for $\mathbf{B}$ requires $O(\sqrt{\widetilde{\kappa}_{AC}})$ multiplications by $\mathbf{A}$, $\mathbf{A}^\top$, $O(\sqrt{\widetilde{\kappa}_{AC}}\sqrt{\kappa_C})$ multiplications by $\mathbf{C}$, $\mathbf{C}^\top$, and $O(\sqrt{\widetilde{\kappa}_{AC}}\sqrt{\kappa_W})$ communication rounds.

Applying Lemma F.3 to the matrix $\widetilde{\mathbf{B}}^\top = (\widetilde{\mathbf{C}}^\top \ \gamma\mathbf{W})$ with $\gamma^2 = \frac{\lambda_{\min+}(\mathbf{S}_{\widetilde{C}^\top}) + \sigma^2_{\max}(\widetilde{\mathbf{C}}^\top)}{\sigma^2_{\min+}(\mathbf{W})}$, we obtain

$$\kappa_{\widetilde{\mathbf{B}}} = \kappa_{\widetilde{\mathbf{B}}^\top} \leq 2\widehat{\kappa}_{\widetilde{C}^\top} + 2\left(\widehat{\kappa}_{\widetilde{C}^\top} + 1\right)\kappa_W^2.$$

Then when we apply Chebyshev acceleration for matrix $\widetilde{\mathbf{B}}$, it requires to perform $O\left(\sqrt{\widehat{\kappa}_{\widetilde{C}^\top}}\right)$ multiplications by $\widetilde{\mathbf{C}}, \widetilde{\mathbf{C}}^\top$ and $O\left(\sqrt{\kappa_W}\sqrt{\widehat{\kappa}_{\widetilde{C}^\top}}\right)$ multiplications by $\mathbf{W}$.

Therefore, in total, when the Chebyshev preconditioning for $\mathbf{K}$ requires $O(\sqrt{\widetilde{\kappa}_{AC}})$ multiplications by $\mathbf{A}, \mathbf{A}^\top$, $O(\sqrt{\widetilde{\kappa}_{AC}}\sqrt{\kappa_C})$ multiplications by $\mathbf{C}, \mathbf{C}^\top$, $O\left(\sqrt{\widehat{\kappa}_{\widetilde{C}^\top}}\right)$ multiplications by $\widetilde{\mathbf{C}}, \widetilde{\mathbf{C}}^\top$ and $O\left((\sqrt{\widetilde{\kappa}_{AC}} + \sqrt{\widehat{\kappa}_{\widetilde{C}^\top}})\sqrt{\kappa_W}\right)$ communication rounds. From this we derived the results in the Table 6.

When local matrices $\widetilde{C}_i$ are equal, for the block $\widetilde{\mathbf{B}}$, preconditioning requires $O(\sqrt{\kappa_C})$ multiplications by $\mathbf{C}, \mathbf{C}^\top$ and $O(\sqrt{\widetilde{\kappa}_{AC}}\sqrt{\kappa_W})$ communication rounds. In that case, the complexities $N_C$ and $N_W$ change to

$$N_{\widetilde{C}} = O\left(\sqrt{\kappa_{\widetilde{C}}}\sqrt{\frac{L_f R^2}{\varepsilon}}\log\left(\frac{1}{\varepsilon}\right)\right)$$

$$N_W = O\left(\sqrt{\widetilde{\kappa}_{AC}}\sqrt{\kappa_W}\sqrt{\frac{L_f R^2}{\varepsilon}}\log\left(\frac{1}{\varepsilon}\right)\right).$$

# L. Missing Proofs From Section 5.2

| Prob. | Oracle | Compl. |
|-------|--------|--------|
| **Coupled Constr. (C)** | Grad. | $\dfrac{M_f^2 R^2}{\varepsilon^2}$ |
| | Mat. | $\sqrt{\widehat{\kappa}_A}\dfrac{M_f R}{\varepsilon}$ |
| | Comm. | $\sqrt{\kappa_W}\sqrt{\widehat{\kappa}_A}\dfrac{M_f R}{\varepsilon}$ |
| | Paper | This paper, Th. 5.2 |
| **Shared Var. Constr. (S)** | Grad. | $\dfrac{M_f^2 R^2}{\varepsilon^2}$ |
| | Mat. $\widetilde{C}$ | $\sqrt{\widehat{\kappa}_{\widetilde{C}^\top}}\dfrac{M_f R}{\varepsilon}$ |
| | Comm. | $\sqrt{\widehat{\kappa}_{\widetilde{C}^\top}}\sqrt{\kappa_W}\dfrac{M_f R}{\varepsilon}$ |
| | Paper | This paper, Th. 5.2 |
| **Local Var. Constr. (C&L)** | Grad. | $\dfrac{M_f^2 R^2}{\varepsilon^2}$ |
| | Mat. $A$ | $\sqrt{\widetilde{\kappa}_{AC}}\dfrac{M_f R}{\varepsilon}$ |
| | Mat. $C$ | $\sqrt{\widetilde{\kappa}_{AC}}\sqrt{\kappa_C}\dfrac{M_f R}{\varepsilon}$ |
| | Comm. | $\sqrt{\widetilde{\kappa}_{AC}}\sqrt{\kappa_W}\dfrac{M_f R}{\varepsilon}$ |
| | Paper | This paper, Th. 5.2 |
| **Mixed Constr. (P)** | Grad. | $\dfrac{M_f^2 R^2}{\varepsilon^2}$ |
| | Mat. $A$ | $\sqrt{\widetilde{\kappa}_{AC}}\dfrac{M_f R}{\varepsilon}$ |
| | Mat. $C$ | $\sqrt{\widetilde{\kappa}_{AC}}\sqrt{\kappa_C}\dfrac{M_f R}{\varepsilon}$ |
| | Mat. $\widetilde{C}$ | $\sqrt{\widehat{\kappa}_{\widetilde{C}^\top}}\dfrac{M_f R}{\varepsilon}$ |
| | Comm. | $\left(\sqrt{\widetilde{\kappa}_{AC}} + \sqrt{\widehat{\kappa}_{\widetilde{C}^\top}}\right)\sqrt{\kappa_W}\dfrac{M_f R}{\varepsilon}$ |
| | Paper | This paper, Th. 5.2 |

*Table 7.* Convergence rates for decentralized nonsmooth and (non-strongly) convex case.

## L.1. Proof of Theorem 5.2

Follow the Theorem 2.5, the penalized reformulation of problem (14) can be solved by gradient sliding with $N_\mathbf{K} = O\left(\dfrac{M_f R}{\varepsilon}\sqrt{\kappa_\mathbf{K}}\right)$ multiplications by $\mathbf{K}, \mathbf{K}^\top$ and $N_{\nabla F} = O\left(\dfrac{M_f^2 R^2}{\varepsilon^2} + N_\mathbf{K}\right)$ calls of subgradient of $F$.

As shown in Appendix K, the Chebyshev preconditioning for $\mathbf{K}$ requires $O(\sqrt{\widetilde{\kappa}_{AC}})$ multiplications by $\mathbf{A}$, $\mathbf{A}^\top$, $O(\sqrt{\widetilde{\kappa}_{AC}}\sqrt{\kappa_C})$ multiplications by $\mathbf{C}, \mathbf{C}^\top$, $O\left(\sqrt{\widehat{\kappa}_{\widetilde{C}^\top}}\right)$ multiplications by $\widetilde{\mathbf{C}}, \widetilde{\mathbf{C}}^\top$ and $O\left(\left(\sqrt{\widetilde{\kappa}_{AC}} + \sqrt{\widehat{\kappa}_{\widetilde{C}^\top}}\right)\sqrt{\kappa_W}\right)$ communication rounds. From this we derived the results in the Table 7.

## M. Missing Proofs From Section 5.3

| Prob. | Oracle | Compl. |
|---|---|---|
| **Coupled Constr. (C)** | Grad. | $\frac{M_f^2}{\mu_f \varepsilon}$ |
| | Mat. $A$ | $\sqrt{\widehat{\kappa}_A} \frac{M_f}{\sqrt{\mu_f \varepsilon}}$ |
| | Comm. | $\sqrt{\kappa_W} \sqrt{\widehat{\kappa}_A} \frac{M_f}{\sqrt{\mu_f \varepsilon}}$ |
| | Paper | This paper, Th. 5.4 |
| **Shared Var. Constr. (S)** | Grad. | $\frac{M_f^2}{\mu_f \varepsilon}$ |
| | Mat. $\widetilde{C}$ | $\sqrt{\widehat{\kappa}_{\widetilde{C}^\top}} \frac{M_f}{\sqrt{\mu_f \varepsilon}}$ |
| | Comm. | $\sqrt{\widehat{\kappa}_{\widetilde{C}^\top}} \sqrt{\kappa_W} \frac{M_f}{\sqrt{\mu_f \varepsilon}}$ |
| | Paper | This paper, Th. 5.4 |
| **Local Var. Constr. (C&L)** | Grad. | $\frac{M_f^2}{\mu_f \varepsilon}$ |
| | Mat. $A$ | $\sqrt{\widetilde{\kappa}_{AC}} \frac{M_f}{\sqrt{\mu_f \varepsilon}}$ |
| | Mat. $C$ | $\sqrt{\widetilde{\kappa}_{AC}} \sqrt{\kappa_C} \frac{M_f}{\sqrt{\mu_f \varepsilon}}$ |
| | Comm. | $\sqrt{\widetilde{\kappa}_{AC}} \sqrt{\kappa_W} \frac{M_f}{\sqrt{\mu_f \varepsilon}}$ |
| | Paper | This paper, Th. 5.4 |
| **Mixed Constr. (P)** | Grad. | $\frac{M_f^2}{\mu_f \varepsilon}$ |
| | Mat. $A$ | $\sqrt{\widehat{\kappa}_A} \frac{M_f}{\sqrt{\mu_f \varepsilon}}$ |
| | Mat. $C$ | $\sqrt{\widetilde{\kappa}_{AC}} \sqrt{\kappa_C} \frac{M_f}{\sqrt{\mu_f \varepsilon}}$ |
| | Mat. $\widetilde{C}$ | $\sqrt{\widehat{\kappa}_{\widetilde{C}^\top}} \frac{M_f}{\sqrt{\mu_f \varepsilon}}$ |
| | Comm. | $\left(\sqrt{\widetilde{\kappa}_{AC}} + \sqrt{\widehat{\kappa}_{\widetilde{C}^\top}}\right) \sqrt{\kappa_W} \frac{M_f}{\sqrt{\mu_f \varepsilon}}$ |
| | Paper | This paper, Th. 5.4 |

*Table 8.* Convergence rates for decentralized nonsmooth and strongly convex optimization with different types of affine constraints.

In the strongly convex and non-smooth setting, we consider the problem (9) on a bounded set $\mathcal{X} \times \widetilde{\mathcal{X}}$, where $\mathcal{X} = X_1 \times \cdots \times X_n$, otherwise Assumption 2.3 and Assumption 2.1 with $\mu > 0$ cannot be held simultaneously. If a first-order algorithm use $\mathbf{y}^0 = \mathbf{0}$ as a starting point, then its iterates $\mathbf{y}^k$ will belong to the linear subspace $\mathcal{L}_m^\perp$. Hence we can restrain the search space on this subspace and rewrite the penalized formulation of the problem (14) as follows

$$\min_{\mathbf{x} \in \mathcal{X}, \mathbf{y} \in \mathcal{L}_m^\perp, \tilde{\mathbf{x}} \in \widetilde{\mathcal{X}}} G(\mathbf{x}, \mathbf{y}, \tilde{\mathbf{x}}) = F(\mathbf{x}, \tilde{\mathbf{x}}) + \frac{r^2}{\varepsilon} \left\| \mathbf{K} \begin{pmatrix} \mathbf{x} \\ \mathbf{y} \\ \tilde{\mathbf{x}} \end{pmatrix} - \mathbf{v} \right\|_2^2. \tag{63}$$

### M.1. Proof of Lemma 5.3

Suppose that Assumption 3.3 holds with $\mu_f \geq 0$. Let $\alpha$ and $\varepsilon$ satisfy following conditions:

$$\alpha^2 = \frac{\mu_{\mathbf{A}} + L_{\mathbf{A}}}{\mu_{\mathbf{W}}}, \quad \varepsilon \leq \frac{4r^2 \mu_{\mathbf{A}}}{\mu_f}. \tag{64}$$

Let

$$\delta = \frac{1}{1 + \frac{\mu_f \varepsilon}{4r^2 L_{\mathbf{A}}}} \in (0, 1). \tag{65}$$

For any $\mathbf{z} = \mathrm{col}(\mathbf{x}, \mathbf{y}, \tilde{\mathbf{x}})$ and $\mathbf{z}' = \mathrm{col}(\mathbf{x}', \mathbf{y}', \tilde{\mathbf{x}}')$, where $\mathbf{x}, \mathbf{x}' \in \mathcal{X}$, $\mathbf{y}, \mathbf{y}' \in \mathcal{L}_m^\perp$ and $\tilde{\mathbf{x}}, \tilde{\mathbf{x}}' \in \widetilde{\mathcal{X}}$, we have

$$
\begin{aligned}
D_G(\mathbf{x}', \mathbf{y}', \tilde{\mathbf{x}}'; \mathbf{x}, \mathbf{y}, \tilde{\mathbf{x}}) &= D_F(\mathbf{x}', \tilde{\mathbf{x}}'; \mathbf{x}, \tilde{\mathbf{x}}) + \frac{r^2}{\varepsilon} \|\mathbf{K}(\tilde{\mathbf{z}}' - \mathbf{z})\|^2 \\
&\stackrel{(a)}{\geq} \frac{\mu_f}{2} \|\mathbf{x}' - \mathbf{x}\|^2 + \frac{\mu_f}{2} \|\tilde{\mathbf{x}}' - \tilde{\mathbf{x}}\|^2 + \frac{r^2}{\varepsilon} \|\mathbf{A}(\mathbf{x}' - \mathbf{x}) + \gamma \mathbf{W}(\mathbf{y}' - \mathbf{y})\|^2 \\
&\stackrel{(b)}{\geq} \frac{\mu_f}{2} \|\mathbf{x}' - \mathbf{x}\|^2 + \frac{\mu_f}{2} \|\tilde{\mathbf{x}}' - \tilde{\mathbf{x}}\|^2 + \frac{r^2(1-\delta)}{\varepsilon} \|\gamma \mathbf{W}(\mathbf{y}' - \mathbf{y})\|^2 - \frac{r^2}{\varepsilon}\left(\frac{1}{\delta} - 1\right) \|\mathbf{A}(\mathbf{x}' - \mathbf{x})\|^2 \\
&\stackrel{(c)}{\geq} \frac{\mu_f}{2} \|\mathbf{x}' - \mathbf{x}\|^2 + \frac{\mu_f}{2} \|\tilde{\mathbf{x}}' - \tilde{\mathbf{x}}\|^2 + \frac{r^2(1-\delta)\gamma^2 \mu_{\mathbf{W}}}{\varepsilon} \|\mathbf{y}' - \mathbf{y}\|^2 - \frac{r^2 L_{\mathbf{A}}}{\varepsilon}\left(\frac{1}{\delta} - 1\right) \|\mathbf{x}' - \mathbf{x}\|^2 \\
&\stackrel{(d)}{=} \frac{\mu_f}{4} \|\mathbf{x}' - \mathbf{x}\|^2 + \frac{\mu_f}{2} \|\tilde{\mathbf{x}}' - \tilde{\mathbf{x}}\|^2 + \frac{r^2 \mu_f(\mu_{\mathbf{A}} + L_{\mathbf{A}})}{4r^2 L_{\mathbf{A}} + \mu_f \varepsilon} \|\mathbf{y}' - \mathbf{y}\|^2 \\
&\stackrel{(e)}{\geq} \frac{\mu_f}{4} \|\mathbf{x}' - \mathbf{x}\|^2 + \frac{\mu_f}{2} \|\tilde{\mathbf{x}}' - \tilde{\mathbf{x}}\|^2 + \frac{r^2 \mu_f(\mu_{\mathbf{A}} + L_{\mathbf{A}})}{4r^2 L_{\mathbf{A}} + 4r^2 \mu_{\mathbf{A}}} \|\mathbf{y}' - \mathbf{y}\|^2 \\
&= \frac{\mu_f}{4} \left\| \begin{pmatrix} \mathbf{x}' - \mathbf{x} \\ \mathbf{y}' - \mathbf{y} \end{pmatrix} \right\|^2 + \frac{\mu_f}{2} \|\tilde{\mathbf{x}}' - \tilde{\mathbf{x}}\|^2 \\
&> \frac{\mu_f}{4} \|\mathbf{z}' - \mathbf{z}\|^2,
\end{aligned}
$$

where (a) is due to Assumption 2.1 and definition of $\mathbf{K}$; (b) is due to Young's inequality; (c) is due to $y' - y \in \mathcal{L}_m^\perp$; (d) is due substitution of $\delta$ in (65) and $\alpha^2$ in (64); (e) is due to the upper bound on $\varepsilon$ in (64).

### M.2. Proof of Theorem 5.4

Consider the problem (14) in the form (63), where $F$ satisfies Assumption 3.3 with $\mu_f > 0$, $\beta, \gamma$ are defined as in Section 4 and $r, \alpha, \varepsilon$ satisfy following conditions:

$$
r \leq \frac{M}{\sigma_{\min^+}(\mathbf{B})}, \quad \alpha^2 = \frac{\mu_{\mathbf{A}} + L_{\mathbf{A}}}{\mu_{\mathbf{W}}}, \quad \varepsilon \leq \frac{4r^2 \mu_{\mathbf{A}}}{\mu_f}. \tag{66}
$$

As shown in Lemma 5.3, $G(\mathbf{x}, \mathbf{y}, \tilde{\mathbf{x}})$ is $\frac{\mu_f}{2}$-strongly convex on $\mathcal{X} \times \mathcal{L}_m^\perp \times \widetilde{\mathcal{X}}$. Applying the GRADIENT SLIDING with restarting procedure to the problem (63) and using the Theorem 2.5, we obtain that an $\varepsilon$-solution can be found using $N_K = O\left(\frac{M_f}{\sqrt{\mu_f \varepsilon}} \sqrt{\kappa_{\mathbf{K}}}\right)$ multiplications by $\mathbf{K}$ and $\mathbf{K}^\top$, and $N_{\nabla F} = O\left(\frac{M_f^2}{\mu_f \varepsilon} + N_K\right)$ calls to a subgradient oracle of $F$. Chebyshev preconditioning for $\mathbf{K}$ requires $O(\sqrt{\tilde{\kappa}_{AC}})$ multiplications by $\mathbf{A}$, $\mathbf{A}^\top$, $O(\sqrt{\tilde{\kappa}_{AC}}\sqrt{\kappa_C})$ multiplications by $\mathbf{C}$, $\mathbf{C}^\top$, $O\left(\sqrt{\hat{\kappa}_{\widetilde{C}^\top}}\right)$ multiplications by $\widetilde{\mathbf{C}}, \widetilde{\mathbf{C}}^\top$ and $O\left((\sqrt{\tilde{\kappa}_{AC}} + \sqrt{\hat{\kappa}_{\widetilde{C}^\top}})\sqrt{\kappa_W}\right)$ communication rounds. Then we receive a matrix $\mathbf{K}'$, for which $\kappa_{\mathbf{K}'} = O(1)$. Hence $N_{\mathbf{K}'} = O\left(\frac{M}{\sqrt{\mu_f \varepsilon}}\right)$. From this we derived the results in the Table 8.

# N. Numerical Experiments

We present numerical experiments illustrating the dependence of the condition number $\kappa_{\mathbf{B}}$ of the coupled and local constraint matrix in (C&L) on the network and local constraint conditioning. The purpose of these experiments is to verify the scaling predicted by Lemma 4.4. We generate synthetic problem instances and vary one parameter at a time while keeping the remaining parameters fixed. First, we vary the condition number $\kappa_W$ of the gossip matrix and observe the predicted linear dependence $\kappa_{\mathbf{B}} = O(\kappa_W)$. Second, we vary the local-constraint condition number $\kappa_C$ and observe the corresponding linear scaling $\kappa_{\mathbf{B}} = O(\kappa_C)$. Finally, we examine how $\kappa_{\mathbf{B}}$ and its theoretical upper bound scale with the number of agents $n$.

Across all three experiments, the empirical behavior supports Lemma 4.4: the condition number of the block constraint matrix $\mathbf{B}$ is governed by the interaction between the coupled constraints, the local constraints, and the communication network. In particular, the theoretical upper bound tracks the observed condition number up to a constant factor, indicating that the dependence on $\kappa_W$ and $\kappa_C$ is tight at the level of scaling.

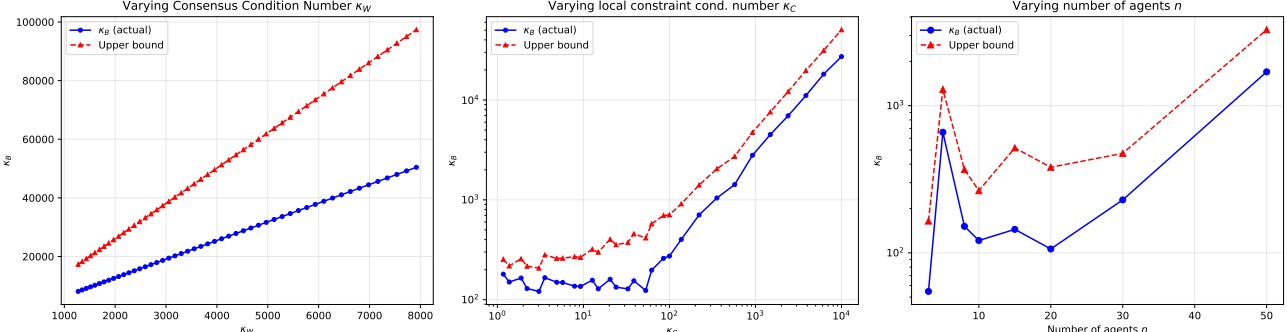

*Figure 1.* Upper bound on $\kappa_B$ for (C&L) problem.

We also conduct a comprehensive experiment comparing the theoretical spectral bounds with the actual condition numbers for different graph topologies. We consider path, cycle, Erdős–Rényi, and complete graphs, and report both the problem parameters and the resulting spectral quantities.

| # | $n$ | $m$ | $d_i$ | $p_i$ | Graph | $\kappa_W$ | $\kappa_C$ | $\widetilde{\kappa}_{AC}$ | $\widetilde{\mu}_{AC}$ | $\alpha$ | $\beta$ |
|---|-----|-----|-------|-------|-------|------------|------------|----------------------------|-------------------------|----------|---------|
| 1 | 10 | 2 | 8 | 3 | path | 1589.10 | 18.55 | 5.19 | 3.4215 | 44.075 | 3.005 |
| 2 | 10 | 2 | 8 | 3 | cycle | 109.67 | 46.85 | 5.14 | 3.6867 | 11.674 | 4.517 |
| 3 | 10 | 2 | 8 | 3 | ER(0.3) | 76.17 | 35.09 | 4.14 | 3.6159 | 5.186 | 3.637 |
| 4 | 10 | 2 | 8 | 3 | ER(0.7) | 7.39 | 17.04 | 4.41 | 3.8687 | 1.155 | 2.693 |
| 5 | 10 | 2 | 8 | 3 | complete | 1.00 | 37.50 | 4.17 | 4.0902 | 0.425 | 4.172 |
| 6 | 10 | 3 | 12 | 4 | cycle | 109.67 | 23.74 | 4.57 | 5.7992 | 13.846 | 3.347 |
| 7 | 10 | 3 | 12 | 4 | complete | 1.00 | 21.75 | 4.05 | 7.7964 | 0.579 | 3.707 |
| 8 | 20 | 2 | 8 | 3 | path | 26065.34 | 20.99 | 5.19 | 4.6701 | 204.701 | 2.851 |
| 9 | 20 | 2 | 8 | 3 | cycle | 1669.82 | 23.22 | 6.78 | 4.0069 | 54.235 | 3.033 |
| 10 | 20 | 2 | 8 | 3 | ER(0.3) | 24.25 | 21.92 | 5.50 | 3.9201 | 1.869 | 3.103 |
| 11 | 20 | 2 | 8 | 3 | complete | 1.00 | 33.63 | 5.44 | 4.1062 | 0.242 | 3.725 |
| 12 | 50 | 2 | 8 | 3 | path | 1025247.85 | 48.75 | 4.52 | 5.3803 | 1283.737 | 4.716 |
| 13 | 50 | 2 | 8 | 3 | cycle | 64331.50 | 40.71 | 4.16 | 4.5566 | 284.393 | 3.964 |
| 14 | 50 | 2 | 8 | 3 | ER(0.3) | 12.94 | 75.58 | 5.49 | 4.1811 | 0.702 | 5.736 |
| 15 | 50 | 2 | 8 | 3 | complete | 1.00 | 103.77 | 4.33 | 4.8067 | 0.094 | 5.941 |
| 16 | 100 | 2 | 8 | 3 | path | 16420168.39 | 105.26 | 7.82 | 4.7657 | 6283.433 | 6.099 |
| 17 | 100 | 2 | 8 | 3 | cycle | 1027273.94 | 65.43 | 5.84 | 4.7524 | 1363.388 | 5.009 |
| 18 | 100 | 2 | 8 | 3 | ER(0.3) | 8.81 | 70.31 | 5.25 | 4.2943 | 0.322 | 5.038 |
| 19 | 100 | 2 | 8 | 3 | complete | 1.00 | 167.56 | 5.01 | 4.9039 | 0.051 | 8.200 |

*Table 9.* Problem setup, graph topology, and component condition numbers.

Table 9 summarizes the generated problem instances. The graph condition number $\kappa_W$ is largest for path and cycle graphs and increases rapidly with the number of agents, while complete graphs have $\kappa_W = 1$. The values of $\kappa_C$, $\widetilde{\kappa}_{AC}$, and $\widetilde{\mu}_{AC}$ remain moderate across the experiments, allowing us to isolate the effect of the network topology on the conditioning of the full constraint matrix.

| # | Graph | $\sigma^2_{\max}(\mathbf{B})$ | bound | ratio | $\sigma^2_{\min+}(\mathbf{B})$ | bound | ratio |
|---|---|---|---|---|---|---|---|
| 1 | path | 29586.73 | 29779.74 | 0.9935 | 3.1626 | 0.8554 | 3.6974 |
| 2 | cycle | 2189.90 | 2717.76 | 0.8058 | 3.4035 | 0.9217 | 3.6928 |
| 3 | ER(0.3) | 1220.27 | 1564.20 | 0.7801 | 3.3833 | 0.9040 | 3.7427 |
| 4 | ER(0.7) | 180.29 | 329.11 | 0.5478 | 3.1714 | 0.9672 | 3.2790 |
| 5 | complete | 387.55 | 420.35 | 0.9220 | 3.4054 | 1.0226 | 3.3303 |
| 6 | cycle | 3080.21 | 3465.61 | 0.8888 | 5.3484 | 1.4498 | 3.6891 |
| 7 | complete | 413.07 | 477.38 | 0.8653 | 6.7898 | 1.9491 | 3.4836 |
| 8 | path | 662215.52 | 662471.04 | 0.9996 | 4.3197 | 1.1675 | 3.6999 |
| 9 | cycle | 47072.23 | 47334.63 | 0.9945 | 3.6883 | 1.0017 | 3.6820 |
| 10 | ER(0.3) | 553.65 | 797.58 | 0.6942 | 3.5162 | 0.9800 | 3.5878 |
| 11 | complete | 347.90 | 392.28 | 0.8869 | 3.3661 | 1.0265 | 3.2791 |
| 12 | path | 26315700.41 | 26316313.99 | 1.0000 | 5.2049 | 1.3451 | 3.8696 |
| 13 | cycle | 1294082.10 | 1294493.75 | 0.9997 | 4.3114 | 1.1392 | 3.7848 |
| 14 | ER(0.3) | 835.83 | 1166.74 | 0.7164 | 3.9442 | 1.0453 | 3.7733 |
| 15 | complete | 1176.93 | 1217.48 | 0.9667 | 4.1885 | 1.2017 | 3.4855 |
| 16 | path | 631392777.26 | 631393979.32 | 1.0000 | 4.6617 | 1.1914 | 3.9127 |
| 17 | cycle | 29741258.90 | 29741980.02 | 1.0000 | 4.5899 | 1.1881 | 3.8632 |
| 18 | ER(0.3) | 693.42 | 920.26 | 0.7535 | 3.9895 | 1.0736 | 3.7161 |
| 19 | complete | 1799.34 | 1849.43 | 0.9729 | 4.3764 | 1.2260 | 3.5697 |

*Table 10.* Actual extremal eigenvalues of $\mathbf{B}^\top\mathbf{B}$ compared with the theoretical spectral bounds.

Table 10 compares the actual extremal nonzero eigenvalues of $\mathbf{B}^\top\mathbf{B}$ with the corresponding theoretical bounds. The upper bound on $\sigma^2_{\max}(\mathbf{B})$ is especially tight for poorly conditioned graphs such as paths and cycles, where the network term dominates. The lower bound on $\sigma^2_{\min+}(\mathbf{B})$ is less tight, but it is still close to the true value up to a small constant factor in all experiments.

| # | $n$ | Graph | $\kappa_B$ | bound | ratio |
|---|---|---|---|---|---|
| 1 | 10 | path | 9355.07 | 34814.87 | 0.2687 |
| 2 | 10 | cycle | 643.43 | 2948.73 | 0.2182 |
| 3 | 10 | ER(0.3) | 360.68 | 1730.37 | 0.2084 |
| 4 | 10 | ER(0.7) | 56.85 | 340.28 | 0.1671 |
| 5 | 10 | complete | 113.81 | 411.08 | 0.2768 |
| 6 | 10 | cycle | 575.91 | 2390.43 | 0.2409 |
| 7 | 10 | complete | 60.84 | 244.92 | 0.2484 |
| 8 | 20 | path | 153302.07 | 567419.75 | 0.2702 |
| 9 | 20 | cycle | 12762.51 | 47253.16 | 0.2701 |
| 10 | 20 | ER(0.3) | 157.46 | 813.84 | 0.1935 |
| 11 | 20 | complete | 103.35 | 382.14 | 0.2705 |
| 12 | 50 | path | 5055908.27 | 19564797.67 | 0.2584 |
| 13 | 50 | cycle | 300150.19 | 1136366.07 | 0.2641 |
| 14 | 50 | ER(0.3) | 211.91 | 1116.20 | 0.1899 |
| 15 | 50 | complete | 280.99 | 1013.15 | 0.2773 |
| 16 | 100 | path | 135442504.85 | 529947315.92 | 0.2556 |
| 17 | 100 | cycle | 6479767.53 | 25033441.52 | 0.2588 |
| 18 | 100 | ER(0.3) | 173.81 | 857.20 | 0.2028 |
| 19 | 100 | complete | 411.14 | 1508.53 | 0.2725 |

*Table 11.* Actual condition number $\kappa_{\mathbf{B}}$ compared with the theoretical upper bound.

Table 11 reports the resulting condition number $\kappa_{\mathbf{B}}$ and its theoretical upper bound. The proposed bound overestimates the true condition number by only a constant factor.

