# OpenReview forum: "Complexity of Decentralized Optimization with Mixed Affine Constraints"
_ICML.cc/2026/Conference — ICML 2026 regular_

### Official Review · Reviewer_X7NV · 2026-03-10

**Soundness:** 4
**Presentation:** 3
**Significance:** 3
**Originality:** 4
**Overall Recommendation:** 5
**Confidence:** 4

**Summary:**

The paper studies decentralized optimization with a general class of mixed affine constraints. The class of problems (P) includes several well-known classes of decentralized optimization as special cases. The paper establishes theoretically tight complexity bounds of certain first-order algorithms (based on existing APAPC or gradient sliding) to solve this problem under several new cases (based on smoothness, convexity, affine constraint types), introducing novel condition numbers that explicitly capture the interplay between local constraint matrices, coupled constraint matrices, and network topology.

**Compliance With Llm Reviewing Policy:**

Affirmed.

**Final Justification:**

The strength of the paper lies in theoretical contribution with carefully introduced condition numbers and a sound derivation. The rebuttal addressed the readability concern by summarizing the notation separately. The question on improving the log-factor and possibility of increased complexity in computing extreme singular values when implementing Chebyshev iteration, but they do not undermine the  core contribution of the work. Given that mine and all other reviewers' concerns have been addressed, I believe the paper is in good standing for an acceptance. I will maintain my high score of 5.

**Key Questions For Authors:**

1. In Section 5.1, is the logarithmic factor a result of regularized APAPC or property of the mixed constraint problem? Are there known lower bounds without log-free rate?
2. How does the algorithm handles computing the largest and smallest singular values in each Chebyshev iteration for large dimensions? Does it increase the overall complexity?

**Limitations:**

The Impact Statement is missing. Although there is no direct negative impact as the work is primarily complexity analysis, the statement should be included according to ICML instructions.

**Strengths And Weaknesses:**

Strengths:
1. The primary strength of this paper is its theoretical comprehensiveness. The authors elegantly unify several classes of decentralized optimization problems. The classes of problems and special cases are highly relevant to modern ML.
2. The theoretical derivations are technically sound. Through the new mixed condition numbers, the results quantify exactly how the joint structure of different affine constraints affect convergence and complexity bounds.

Weaknesses:
1. A discussion on whether an optimal log-free bound is achievable in Section 5.1 would be appreciated. As the reason seems to be regularization, a direct primal-dual approach would do the job?
2. The paper is dense in notation, making the results difficult to comprehend on a quick read. A table summarizing the block matrices and the condition numbers with their interpretation would be helpful.

---

> ### Author Rebuttal · Authors · 2026-03-31
>
> Thank you for your feedback. We appreciate your recognition of the technical soundness of our paper. In response to your comments:
>
> 1. The paper is dense in notation, making the results difficult to comprehend on a quick read. A table summarizing the block matrices and the condition numbers with their interpretation would be helpful.
>
>    - Thank you for your recommendation. We have added the tables for notation to the manuscript. The tables can be found in https://ibb.co/8C3hMDP.
>
> 2. In Section 5.1, is the logarithmic factor a result of regularized APAPC or property of the mixed constraint problem? Are there known lower bounds without log-free rate? As the reason seems to be regularization, a direct primal-dual approach would do the job?
>    - Thank you for asking this. We believe that the log factor can be removed, but we are not aware of the existence of accelerated primal-dual gradient method suitable for this setup. We observed that the logarithmic factor might be removed from the gradient oracle complexity by using PSTM method from [1] instead of APAPC with regularization. Moreover, [2] presented a gradient-tracking method for consensus decentralized optimization without the logarithmic factor in the communication complexity. Transferring this result to the general affine constraints (similarly to the generalization APAPC for general affine constraints in strongly convex setup in [3]) is an interesting research direction.
>
>
> 3. How does the algorithm handles computing the largest and smallest singular values in each Chebyshev iteration for large dimensions? Does it increase the overall complexity?
>
>    - In principle, the largest and the smallest singular values can be efficiently estimated using matrix-free methods [4], though in our case this will require a few centralized communication rounds to compute a matrix-vector multiplication with an interaction matrix $S_B$. Also it is not clear theoretically, what is the complexity of estimating the smallest nonzero singular value comparing to the complexity of Chebyshev iterations. But conceptually, we treat these values as hyperparameters of our algorithm, similarly to other works on gradient methods which are not devoted to parameter-free methods.
>
> [1] Dvinskikh, Gasnikov - Decentralized and Parallel Primal and Dual Accelerated
> Methods for Stochastic Convex Programming Problem
>
> [2] Li, Lin -  Accelerated Gradient Tracking over Time-varying Graphs for Decentralized Optimization
>
> [3] Salim et al - An optimal algorithm for strongly convex minimization under affine constraints
>
> [4] Lehoucq et al - ARPACK Users' Guide: Solution of Large-Scale Eigenvalue Problems with Implicitly Restarted Arnoldi Methods"

---

> > ### Author Rebuttal · Reviewer_X7NV · 2026-04-03
> >
> > Given that the authors provided the notation table, clarified the technical limitations of the log-factor, and presented a reason of their algorithmic choices, the paper is in solid standing.

---

> > > ### Author Response · Authors · 2026-04-06
> > >
> > > Dear Reviewer,
> > >
> > > We are grateful to you for a positive and constructive review.

---

### Official Review · Reviewer_nU89 · 2026-03-11

**Soundness:** 3
**Presentation:** 2
**Significance:** 3
**Originality:** 2
**Overall Recommendation:** 4
**Confidence:** 3

**Summary:**

This paper studies convex optimization with affine constraints. An APAPC and an gradient sliding algorithm have been applied to the problem with smooth and non-smooth functions, respectively. The convergence bounds are established and the lower bounds are also derived, demonstrating the optimality in the strongly convex case.

**Compliance With Llm Reviewing Policy:**

Affirmed.

**Final Justification:**

The authors have addressed my comments and I have raised my score.

**Key Questions For Authors:**

- The problem is not new and has been studied by many existing works (which is fine). However, many results and discussions (sections 2 and 3) are devoted to existing results by others. The APAPC algorithm is proposed in a prior work. Is it merely applied to the problem here? Is there any novelty in the algorithm design itself? Many results/theorems seem to be review of existing results instead of new results.
- The structure of the paper is a bit hard to follow. Sometimes I find it difficult to identify which results are newly developed and which are existing results. The novelty of the algorithm, the convergence results, and analysis tools should be clearly highlighted.
- No experimental result is presented.

**Strengths And Weaknesses:**

Strength:
- The problem formulation is quite general and covers many practical problems as special cases.
- The theoretical study is comprehensive and contains both upper and lower bounds.

Weakness
- The problem is not new and has been studied by many existing works (which is fine). However, many results and discussions (sections 2 and 3) are devoted to existing results by others. The APAPC algorithm is proposed in a prior work. Is it merely applied to the problem here? Is there any novelty in the algorithm design itself? Many results/theorems seem to be review of existing results instead of new results.
- The structure of the paper is a bit hard to follow. Sometimes I find it difficult to identify which results are newly developed and which are existing results. The novelty of the algorithm, the convergence results, and analysis tools should be clearly highlighted.
- No experimental result is presented.

---

> ### Author Rebuttal · Authors · 2026-03-31
>
> Thank you for your detailed and constructive feedback. We address your main concerns below:
> 1. The problem is not new and has been studied by many existing works (which is fine). However, many results and discussions (sections 2 and 3) are devoted to existing results by others. The APAPC algorithm is proposed in a prior work. Is it merely applied to the problem here? Is there any novelty in the algorithm design itself? Many results/theorems seem to be review of existing results instead of new results.
>     - The novelty of the algorithms:
>         1. For smooth and strongly convex objectives the only novelty in algorithmic design is in the scheme of applying Chebyshev preconditioning to the blocks of the constraint's matrix, which gives optimal dependence on the condition numbers of matrices, as verified by our lower bounds. In addition to that, for other setups we also develop following techniques:
>         2. For smooth non-strongly convex objectives we use the regularization technique and prove that it works in the affine-constrained case (Theorem 2.4.2);
>         3. For non-smooth strongly convex setup, we use penalization to both handle affine constraints and simultaneously obtain strong convexity by the slack variable $y$, which is also new (Lemma 5.3).
>    We will explicitly highlight these points in the new version of the paper.
> 2. The structure of the paper is a bit hard to follow. Sometimes I find it difficult to identify which results are newly developed and which are existing results. The novelty of the algorithm, the convergence results, and analysis tools should be clearly highlighted.
>     - We included many existing results in the text with the intention of making it clear how we obtained our new results and how they fit in the global picture. Realizing that this can confuse the reader, we either marked each new result as **new** (along with a reference to the proof) or cite the source of each known result. Could you please point us to specific examples, where novelty of a result is not clearly stated?
>
>
> 3. No experimental result is presented.
>     - In the new version of the paper, we present experimental results for estimating an upper bound on $\kappa_{\mathbf{B}}$, the condition number of the constraint matrix associated with the coupled and local constraints (C&L) problem. First, we isolate the dependence of $\kappa_{\mathbf{B}}$ on the graph condition number $\kappa_W$ and verify the linear scaling $\kappa_{\mathbf{B}} = O(\kappa_W)$ predicted by Lemma 4.4. Next, we examine the dependence on $\kappa_C$, confirming the linear scaling $\kappa_{\mathbf{B}} = O(\kappa_C)$ while keeping all other parameters fixed. Finally, we study how both $\kappa_{\mathbf{B}}$ and its upper bound scale with the number of agents $n$. Across all three experiments, the results support Lemma 4.4: the condition number $\kappa_{\mathbf{B}}$ of the block constraint matrix $\mathbf{B}$ is controlled by $\widetilde{\kappa}_{AC}(\kappa_W + \kappa_C)$, and the theoretical upper bound in (49) is tight up to a constant factor. The corresponding plot can be found at https://ibb.co/x8dDPk85. We also conduct a comprehensive experiment to compare theoretical bounds and actual condition numbers under different type of graphs. The tables of results can be found in https://ibb.co/JYQrbv6.

---

> > ### Author Rebuttal · Reviewer_nU89 · 2026-04-01
> >
> > The authors have highlighted their novelty and promised to reorganize the paper to improve readability and to add some experiments.

---

> > > ### Author Response · Authors · 2026-04-06
> > >
> > > Thank you for your response. Let us summarize the discussion of each weakness/question.
> > >
> > > 1. (**Algorithmic novelty**).
> > > > The authors have highlighted their novelty.
> > >
> > > We thank you for admitting that we underlined the novelty of the paper.
> > >
> > > 2. (**Presentation and highlighting the novelty**).
> > > > The authors promised to reorganize the paper to improve readability.
> > >
> > > In response to your review, we will explicitly highlight novelty of the algorithms as reflected in the rebuttal. Also, as requested by other reviewers, we will add tables with notation and more explicit listings of algorithms. Note that these changes do not require a reorganization of the text. Note, that a clear indication of new vs known results is already present in the original version of the manuscript.
> > >
> > > 3. (**Numerical experiments**).
> > > > The authors promised to add some experiments.
> > >
> > > We conducted experiments on how condition number $\kappa_B$ depends on $\kappa_W$ and $\kappa_C$ and attached the results to the rebuttal.
> > >
> > > Please let us know if we correctly understood your concerns. If any issues remain we are glad to clarify them, as well. We kindly ask you to reconsider your score if all the questions and weaknesses were addressed.

---

### Official Review · Reviewer_k8LP · 2026-03-12

**Soundness:** 3
**Presentation:** 2
**Significance:** 3
**Originality:** 3
**Overall Recommendation:** 4
**Confidence:** 3

**Summary:**

This paper studies decentralized convex optimization with mixed affine constraints, including coupled, local, and shared-variable constraints, under a unified problem formulation. Building on tools such as APAPC, gradient sliding, and Chebyshev acceleration, the authors develop first-order methods for this setting. The paper also establishes lower complexity bounds and provides matching optimal methods for smooth and strongly convex objectives, with complexity characterized by mixed condition numbers.

**Compliance With Llm Reviewing Policy:**

Affirmed.

**Final Justification:**

Although the idea novelty is somewhat limited, the paper is overall well executed and complete. The technical development is solid, the presentation is generally clear, and the empirical study is reasonably thorough. Considering the completeness of the work, I will keep a positive score.

**Key Questions For Authors:**

1. Could the authors provide a minimal numerical simulation to empirically illustrate the predicted convergence behavior?
2. The nonsmooth strongly convex result is obtained by restricting the problem to a bounded set $X$ so that bounded subgradients are compatible with $\mu_f>0$. This seems quite restrictive. Could the authors comment on how realistic this assumption is for the motivating applications, and whether there is any plausible analytical route to avoid it without artificially bounding the domain?
3. The smooth non-strongly convex result incurs an extra logarithmic factor through the regularization used to adapt APAPC. Do the authors believe this is mainly a limitation of that regularization step, or does it reflect a deeper bottleneck in obtaining optimal rates for mixed affine constraints without strong convexity?

**Limitations:**

Yes

**Strengths And Weaknesses:**

Strengths.
1.The paper is theoretically strong and mostly complete. It studies a significant problem and provides a unified framework for decentralized optimization under mixed affine constraints, covering several important settings.
2. For the smooth and strongly convex case, it establishes matching lower and upper bounds, and the mixed condition numbers offer a useful way to characterize the joint effect of network topology and constraint structure.
3. The paper is also informative and positions itself clearly relative to prior work.

Weaknesses.
1. The paper is quite difficult to read due to its dense notation and analysis.
2. Its main novelty lies more in the unified framework and complexity analysis than in fundamentally new algorithmic ideas, since the methods are largely built from existing tools.
3. In addition, the smooth convex result remains suboptimal by a logarithmic factor, the nonsmooth strongly convex analysis relies on a bounded-set assumption, and the framework does not yet cover the most general coupling between local and shared variables.
4. Finally, the paper does not include empirical validation.

---

> ### Author Rebuttal · Authors · 2026-03-31
>
> Thank you for your review and recognition of our contributions. We address your concerns below:
>
> Weaknesses.
> 1. The paper is quite difficult to read due to its dense notation and analysis.
>     - We have added the tables for explanation the notation of matrices and condition numbers to the new version of the paper. The tables can be found in https://ibb.co/8C3hMDP. We hope they will help you understand the article better.
>
> 2. Its main novelty lies more in the unified framework and complexity analysis than in fundamentally new algorithmic ideas, since the methods are largely built from existing tools.
>   novelty in the algorithm design itself?
>     - For smooth and strongly convex objectives the only novelty in algorithmic design is in the scheme of applying Chebyshev preconditioning to the blocks of the constraint's matrix, which gives optimal dependence on the condition numbers of matrices, as verified by our lower bounds. In addition to that, for other setups we also develop following techniques:
>     - For smooth non-strongly convex objectives we use the regularization technique and prove that it works in the affine-constrained case (Theorem 2.4.2);
>     - For non-smooth strongly convex setup, we use penalization to both handle affine constraints and simultaneously obtain strong convexity by the slack variable $y$, which is also new (Lemma 5.3).
>
>
> 3. The smooth convex result remains suboptimal by a logarithmic factor, the nonsmooth strongly convex analysis relies on a bounded-set assumption, and the framework does not yet cover the most general coupling between local and shared variables.
>     - Please see the answers to questions 2 and 3
>
> 4. The paper does not include empirical validation.
>     - For now we have experiments on the new constants, that we introduced for complexity of mixed constraint problem.  We will conduct the experiments to illustrate convergence behavior of proposed algorithms, for example on synthetic problems like quadratic objective functions and add the results to the new version of the paper.
>
> Questions:
> 1. Could the authors provide a minimal numerical simulation to empirically illustrate the predicted convergence behavior?
>     - For now, we present experimental results for estimating an upper bound on $\kappa_{\mathbf{B}}$, the condition number of the constraint matrix associated with the coupled and local constraints (C&L) problem. First, we isolate the dependence of $\kappa_{\mathbf{B}}$ on the graph condition number $\kappa_W$ and verify the linear scaling $\kappa_{\mathbf{B}} = O(\kappa_W)$ predicted by Lemma 4.4. Next, we examine the dependence on $\kappa_C$, confirming the linear scaling $\kappa_{\mathbf{B}} = O(\kappa_C)$ while keeping all other parameters fixed. Finally, we study how both $\kappa_{\mathbf{B}}$ and its upper bound scale with the number of agents $n$. Across all three experiments, the results support Lemma 4.4: the condition number $\kappa_{\mathbf{B}}$ of the block constraint matrix $\mathbf{B}$ is controlled by $\widetilde{\kappa}_{AC}(\kappa_W + \kappa_C)$, and the theoretical upper bound in (49) is tight up to a constant factor. The corresponding plot can be found at https://ibb.co/x8dDPk85. We also conduct a comprehensive experiment to compare theoretical bounds and actual condition numbers under different type of graphs. The tables of results can be found in https://ibb.co/JYQrbv6.
>
> 2. The nonsmooth strongly convex result is obtained by restricting the problem to a bounded set $X$ so that bounded subgradients are compatible with $\mu_f>0$.
>     - This is a classical issue that $\mu$-strong convexity contradicts $M$-Lipschitz continuousness when the domain is unrestricted. The simplest workaround, which we used, is to assume restricted domain. Indeed, it is possible to get rid of this assumption by defining $M$ as the largest Lipschitz constant of the objective on the trajectory $\{\mathbf{x}^k\}_{k=1}^N$ of the optimization method.
> 3. The smooth non-strongly convex result incurs an extra logarithmic factor through the regularization used to adapt APAPC.
>     - Thank you for highlighting this property. We observed, that the logarithmic factor might be removed from the gradient oracle complexity by using PSTM method from [1] instead of APAPC with regularization. Moreover, [2] presented a gradient-tracking method for consensus decentralized optimization without the logarithmic factor in the communication complexity. Transferring this result to the general affine constraints (similarly to the generalization APAPC for general affine constraints in strongly convex setup in [3]) is an interesting research direction.
>
> [1] Dvinskikh, Gasnikov - Decentralized and Parallel Primal and Dual Accelerated
> Methods for Stochastic Convex Programming Problem
>
> [2] Li, Lin -  Accelerated Gradient Tracking over Time-varying Graphs for Decentralized Optimization
>
> [3] Salim et al - An optimal algorithm for strongly convex minimization under affine constraints

---

> > ### Author Rebuttal · Reviewer_k8LP · 2026-04-02
> >
> > My concerns have been addressed.

---

> > > ### Author Response · Authors · 2026-04-06
> > >
> > > Dear Reviewer,
> > >
> > > We thank you for your detailed comments and review. Let us summarize what we added in the rebuttal.
> > >
> > > 1. Table with notation https://ibb.co/8C3hMDP.
> > >
> > > 2. Numerical simulations for dependence of $\kappa_B$ on $\kappa_C$ and $\kappa_W$ https://ibb.co/x8dDPk85 and a comparison for different condition numbers against their theoretical bounds https://ibb.co/JYQrbv6.
> > >
> > > 3. Note that papers [1,2] address your question on complexity gap for non-strongly convex problems.
> > >
> > > We hope that you acknowledge our answers and reconsider your score.

---

### Official Review · Reviewer_7dkE · 2026-03-13

**Soundness:** 3
**Presentation:** 3
**Significance:** 3
**Originality:** 3
**Overall Recommendation:** 4
**Confidence:** 3

**Summary:**

This paper studies decentralized first-order methods for convex optimization problems with mixed affine constraints. The setting is fairly general and appears to unify several important special cases in decentralized optimization, including consensus-type, local, and coupled constraints. The main technical contribution is a complexity analysis under multiple problem classes: the paper claims optimal complexity for the smooth strongly convex case, and nearly optimal guarantees for the smooth convex and nonsmooth convex cases. Overall, this is a theory-oriented paper with meaningful technical contents.

**Compliance With Llm Reviewing Policy:**

Affirmed.

**Final Justification:**

I will keep my score since it is already positive.

**Key Questions For Authors:**

See weaknesses:

1. Could the authors provide a more explicit decentralized implementation for the proposed algorithms?

2. The complexity bounds are presented in terms of global block operators (e.g., reformulated matrices such as $A,C$.) Can the authors explain whether these results can be translated into more explicit per-node computational costs?

**Limitations:**

yes

**Strengths And Weaknesses:**

Strengths:
1. THis paper is well-written. It firstly introduces the APAPC algorithm, then splits different settings to fit this algorithm. Moreover, for different secenarios (mixed constriants, coupled\general case), the authors provide modifications to the algoirthms so that the complexity could be improved to be nearly optimal.

2. Beyond the strongly convex case, the paper also discusses smooth convex and nonsmooth convex settings, which makes the overall theory more comprehensive than a single-regime result.

Weaknesses:

1. While the complexity results are interesting, the paper does not appear to introduce a fundamentally new optimization algorithm. A substantial part of the contribution comes from reformulating the problem so that existing methods (and their analysis tools) can be applied. This is still a valid contribution, but the paper should be more explicit about what is genuinely new at the algorithmic level versus what is inherited from prior methods.
2. For specific cases, the practical implementations of the algorithm should be provided, in the appendix.
3. It is better to provide numerical experimenets, since this is a machine learning conference.

---

> ### Author Rebuttal · Authors · 2026-03-31
>
> Thank you for your detailed feedback. We respectfully address your concerns:
>
> Weaknesses:
>
> 1. While the complexity results are interesting, the paper does not appear to introduce a fundamentally new optimization algorithm. A substantial part of the contribution comes from reformulating the problem so that existing methods (and their analysis tools) can be applied. This is still a valid contribution, but the paper should be more explicit about what is genuinely new at the algorithmic level versus what is inherited from prior methods.
>     - For smooth and strongly convex objectives the only novelty in algorithmic design is in the scheme of applying Chebyshev preconditioning to the blocks of the constraint's matrix, which gives optimal dependence on the condition numbers of matrices, as verified by our lower bounds. In addition to that, for other setups we also develop following techniques:
>     - For smooth non-strongly convex objectives we use the regularization technique and prove that it works in the affine-constrained case (Theorem 2.4.2);
>     - For non-smooth strongly convex setup, we use penalization to both handle affine constraints and simultaneously obtain strong convexity by the slack variable $y$, which is also new (Lemma 5.3).
>
>
> 2. For specific cases, the practical implementations of the algorithm should be provided, in the appendix.
>     - We have added the implementations the Chebyshev acceleration procedure for each kind of problems. The listings can be found in https://ibb.co/CswPZfHr.
>
> 3. It is better to provide numerical experimenets, since this is a machine learning conference.
>     - We present experimental results for estimating an upper bound on $\kappa_{\mathbf{B}}$, the condition number of the constraint matrix associated with the coupled and local constraints (C&L) problem. First, we isolate the dependence of $\kappa_{\mathbf{B}}$ on the graph condition number $\kappa_W$ and verify the linear scaling $\kappa_{\mathbf{B}} = O(\kappa_W)$ predicted by Lemma 4.4. Next, we examine the dependence on $\kappa_C$, confirming the linear scaling $\kappa_{\mathbf{B}} = O(\kappa_C)$ while keeping all other parameters fixed. Finally, we study how both $\kappa_{\mathbf{B}}$ and its upper bound scale with the number of agents $n$. Across all three experiments, the results support Lemma 4.4: the condition number $\kappa_{\mathbf{B}}$ of the block constraint matrix $\mathbf{B}$ is controlled by $\widetilde{\kappa}_{AC}(\kappa_W + \kappa_C)$, and the theoretical upper bound in (49) is tight up to a constant factor. The corresponding plot can be found at https://ibb.co/x8dDPk85. We also conduct a comprehensive experiment to compare theoretical bounds and actual condition numbers under different type of graphs. The tables of results can be found in https://ibb.co/JYQrbv6.
>
> Questions:
>
> 1. Could the authors provide a more explicit decentralized implementation for the proposed algorithms?
>    - Multiplication $\mathbf W \mathbf x$ is equivalent to receiving local variables $x_j$ from each neighbour $j \in N(i)$ of node $i$ and computing weghted sum $\sum_{j \in N(i) \cup \{i\}}W_{ij} x_j$.
>
> 2. The complexity bounds are presented in terms of global block operators (e.g., reformulated matrices such as $A,C$.) Can the authors explain whether these results can be translated into more explicit per-node computational costs?
>     - All matrices in complexity bounds ($A$, $C$, $\tilde C$) are block-diagonal. Since $\mathbf x$ is the column-stacked vector of local variables $x_i$, multiplication $A \mathbf x$ is equivalent to performing local multiplications $A_i x_i$ at each node, same for matrices $C$ and $\tilde C$.

---

> > ### Author Rebuttal · Reviewer_7dkE · 2026-04-02
> >
> > Thank you for your response. My concerns have beed addressed. I will keep my score.

---

> > > ### Author Response · Authors · 2026-04-06
> > >
> > > Dear Reviewer,
> > >
> > > We thank you for your review and positive feedback.

---

### Decision · Program_Chairs · 2026-04-30

**Decision:**

Accept (regular)

**Comment:**

The reviewers find that the manuscript has solid contributions and is thorough in its technical development. The authors are encouraged to incorporate the reviewers' comments when preparing the final version.